# DC-DPM: A DIVIDE-AND-CONQUER APPROACH FOR DIFFUSION REVERSE PROCESS

## ABSTRACT

Diffusion models have achieved great success in generative tasks, with the quality of generated samples guaranteed by their convergence properties, typically derived within the context of stochastic differential equations(SDE) and often involving Kolmogorov equations for proofs. This paper introduces a novel method for proving the convergence of diffusion models, which relies on direct estimation of distributions without the need for SDE tools. This approach inspires a **D**ivide-and-**C**onquer strategy for approximating the reversed transition kernel of **D**iffusion **P**robabilistic **M**odels (DC-DPM), which is not derived from SDEs, making previous convergence methods inapplicable. However, our method can be easily extended to accommodate this. As our DC-DPM learns specific kernels for each partition , these kernels require merging. According to the proof of convergence, we design two merging strategies for these cluster-specific kernels along with corresponding training and sampling methods. Experimental results demonstrate the superior generation quality of our method compared to the traditional single Gaussian kernel. Furthermore, our DC-DPM can synergize with previous kernel optimization methods, enhancing their generation quality, especially with a small number of timesteps.

## 1 INTRODUCTION

Diffusion models have recently gained prominence in generating multi-modal content across various tasks, including image generation (Dhariwal & Nichol, 2021; Ho et al., 2020; Rombach et al., 2022; Saharia et al., 2022; Ramesh et al., 2022), image super-resolution (Li et al., 2022), video generation (Ho et al., 2022a;b), text-to-speech synthesis (Popov et al., 2021), 3D generation (Poole et al., 2022), and motion planning (Carvalho et al., 2023).

Diffusion models generate data by iteratively predicting noise and solving diffusion SDEs to denoise (Song et al., 2020b). Given the complexity of this process, achieving convergence towards the desired data distribution is not a straightforward task, particularly when using a score function approximated by a neural network.

Significant research advancements have been made to validate the convergence of diffusion models. Lee et al. (2022) were the first to provide polynomial convergence guarantees for diffusion models. The scope of this convergence was further expanded to a broader range of data distributions (Chen et al., 2022; Lee et al., 2023). De Bortoli (2022) demonstrated convergence when the data is only supported on a lower-dimensional manifold. Chen et al. (2024); Benton et al. (2023) focused on the development of convergence for deterministic sampling. Additionally, Li et al. (2023) were able to achieve a superior error bound by making additional assumptions on the Jacobian of the score functions. However, the aforementioned convergence relies heavily on tools from SDEs, such as the Kolmogorov equations (Lee et al., 2022) and the Girsanov theorem (Chen et al., 2022). This reliance makes it challenging to adapt these methods to cases where the reverse process is not derived from a SDE.

In this paper, we introduce a novel method to demonstrate the convergence of diffusion models. Initially, we establish a error bound for one step by $\Delta t^{\frac{3-\beta}{2}}$ and $\epsilon_y$, which mirrors the "local error" in numerical methods for ODEs. Here, $\beta$ is a positive number and $\epsilon_y$ denotes the $L^2$ error of the neural network approximation. Subsequently, we tackle the corner cases at $t = 0$ and $t = 1$. As a

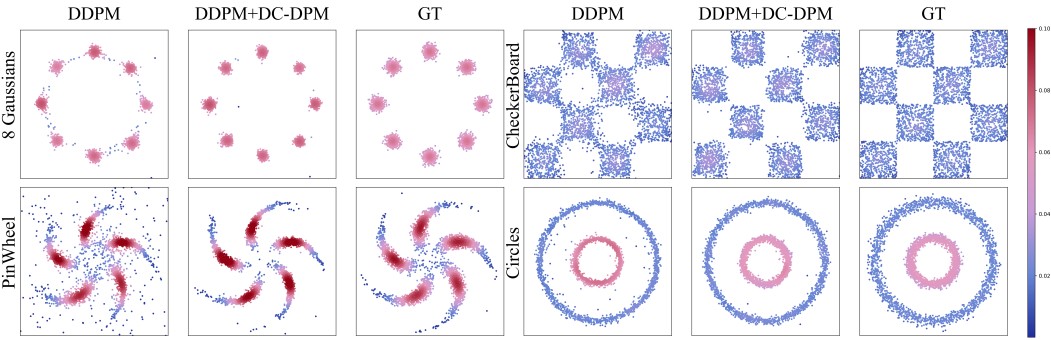

Figure 1: **DC-DPM improves generation quality on small timesteps**. Employing a divide-and-conquer approach to approximate transition kernels in diffusion reverse process, DC-DPM generates samples closer to ground truth distribution (GT) on 20 denoising steps. Colors represent data density.

result, we consolidate the preceding propositions and attain the convergence result. The properties and corollaries presented in this paper are all unique contributions from our end.

Inspired by the proof above, we propose DC-DPM, a novel approach to represent the transition kernel in the reverse process of diffusion probabilistic models (DPM) using a Divide-and-Conquer strategy. To attain the convergence of DC-DPM, we employ the convexity of the Kullback-Leibler divergence to transform the error term into the form of a single Gaussian case. This altered form is consistent with the scenario outlined in our newly proposed proof method. As a result, we easily prove the convergence of the newly proposed DC-DPM.

According to the convergence proof of DC-DPM, we cluster the data into different partitions, DC-DPM learns reversed transition kernels for each data cluster and models the overall transition kernel as a composition of these cluster-specific kernels. To determine how to combine the cluster-specific kernels, we design two strategies along with corresponding training and sampling methods. DC-DPM can collaborate with previous diffusion sampling optimization methods, particularly those focused on optimizing the Gaussian transition kernel design, such as Extended-Analytic-DPM and GMS, by utilizing the representations proposed in these works for each cluster-specific kernel in our DC-DPM. Moreover, the convergence is not dependent on the data partitions, implying that any data division pattern will lead to a convergent result.

Experimental results on 2D toy datasets and image datasets demonstrate that our method enhances the generation quality of diffusion models compared to the traditional single Gaussian transition kernel representation. Furthermore, our approach significantly improves the performance of previous transition kernel optimization methods, including Extended-Analytic-DPM and GMS, especially in scenarios with limited sampling steps.

Proofs for all Propositions are given in the Appendix.

## 2 BACKGROUND: DIFFUSION PROBABILISTIC MODELS AND ITS CONVERGENCE

### 2.1 DIFFUSION PROBABILISTIC MODELS AND TRANSITION KERNELS

Given a finite set of data samples $\{\boldsymbol{y}_i \in \mathbb{R}^d | i = 1, 2, \ldots, N\}$, where $d$ represents the data dimension and $N$ is the number of samples. The distribution of these samples is characterized by:

$$p_{data}(\boldsymbol{x}) = \frac{1}{N} \sum_{i=0}^{N} \delta(\boldsymbol{x} - \boldsymbol{y}_i), \tag{1}$$

where $\delta(\boldsymbol{x})$ represents Dirac delta function. As real training processes are typically conducted on such finite datasets, we assume that the ground truth data distribution adheres to Eq. (1).

Diffusion probabilistic models define two Markov chains including forward process and reverse process. The forward process is typically hand-designed with Gaussian transition kernel to perturb

data to noise and can be expressed as (Zhang et al., 2024):

$$p(\boldsymbol{x}_t, t | \boldsymbol{x}_s, s) = \mathcal{N}(\boldsymbol{x}_t; \alpha_{t|s}\boldsymbol{x}_s, \sigma_{t|s}^2 \boldsymbol{I}), \tag{2}$$

where $t, s$ are two timesteps and $0 \le s < t \le 1$. $\alpha_{t|s} = \frac{\alpha_t}{\alpha_s}$ and $\sigma_{t|s} = \sqrt{1 - \alpha_{t|s}^2}$, where $\alpha_t$ is a hyperparameter which decreases monotonically from 1 to 0 over time $t$ (Kingma et al., 2021).

Hence, the conditional distribution of $\boldsymbol{x}_t$ given $\boldsymbol{x}_0$ can be derived as:

$$p(\boldsymbol{x}_t, t | \boldsymbol{x}_0, 0) = \mathcal{N}(\boldsymbol{x}_t; \alpha_t \boldsymbol{x}_0, \sigma_t^2 \boldsymbol{I}). \tag{3}$$

Taking the initial condition Eq. (1) into account, the single time marginal distribution of $\boldsymbol{x}_t$ is:

$$p(\boldsymbol{x}_t, t) = \frac{1}{N} \sum_i (2\pi\sigma_t^2)^{-\frac{d}{2}} \exp(-\frac{||\boldsymbol{x}_t - \alpha_t \boldsymbol{y}_i||^2}{2\sigma_t^2}). \tag{4}$$

The reverse process reverses the forward one with a learned kernel. Based on Eq. (2) and (4), the ground truth reversed transition kernel can be derived with Bayes' rule:

$$\begin{aligned} p(\boldsymbol{x}_s, s | \boldsymbol{x}_t, t) &= p(\boldsymbol{x}_t, t | \boldsymbol{x}_s, s) \frac{p(\boldsymbol{x}_s, s)}{p(\boldsymbol{x}_t, t)} \\ &= (2\pi\sigma_{s|t})^{-\frac{d}{2}} \sum_i w_i(\boldsymbol{x}_t, t) \exp\{-\frac{1}{2\sigma_{s|t}^2}||\boldsymbol{x}_s - \frac{\alpha_{t|s}\sigma_s^2}{\sigma_t^2}\boldsymbol{x}_t - \frac{\alpha_s\sigma_{t|s}^2}{\sigma_t^2}\boldsymbol{y}_i||^2\}, \end{aligned} \tag{5}$$

where $\sigma_{s|t} = \sigma_{t|s}\frac{\sigma_s}{\sigma_t}$. $w_i(\boldsymbol{x}_t, t) = \frac{u_i(\boldsymbol{x}_t, t)}{\sum_j u_j(\boldsymbol{x}_t, t)}$ while $u_i(\boldsymbol{x}_t, t) = \exp(-\frac{||\boldsymbol{x}_t - \alpha_t \boldsymbol{y}_i||^2}{2\sigma_t^2})$.

Existing methods typically approximate the learnable reversed transition kernel as a single Gaussian distribution (Ho et al., 2020). The transition kernel can be expressed as (Zhang et al., 2024):

$$\tilde{p}(\boldsymbol{x}_s, s | \boldsymbol{x}_t, t) = (2\pi\sigma_{s|t})^{-\frac{d}{2}} \exp\{-\frac{1}{2\sigma_{s|t}^2}||\boldsymbol{x}_s - \frac{\alpha_{t|s}\sigma_s^2}{\sigma_t^2}\boldsymbol{x}_t - \frac{\alpha_s\sigma_{t|s}^2}{\sigma_t^2}\bar{\boldsymbol{y}}(x_t, t)||^2\}, \tag{6}$$

The mean of this Gaussian distribution is related to $\bar{\boldsymbol{y}}(x_t, t)$, which is estimated by a neural network $\boldsymbol{y}_\theta(x_t, t)$ in $\boldsymbol{x}$-prediction methods, while the variance is isotropic and only depends on the timestep $s$ and $t$. Another commonly used parameterization is $\boldsymbol{\epsilon}$-prediction, which employs a noise prediction network to estimate the noise $\boldsymbol{\epsilon}(\boldsymbol{x}_t, t)$ (Salimans & Ho, 2022). Despite its difference from $\boldsymbol{x}$-prediction, these two parameterizations are equivalent, as demonstrated by the relationship

$$\boldsymbol{x}_t = \alpha_t \bar{\boldsymbol{y}}(\boldsymbol{x}_t, t) + \sigma_t \boldsymbol{\epsilon}(\boldsymbol{x}_t, t). \tag{7}$$

In this paper, we utilize $\boldsymbol{x}$-prediction for simplicity in our proofs. For clarity, we will refer to $p(\boldsymbol{x}_t)$ and $p(\boldsymbol{x}_s \mid \boldsymbol{x}_t)$ instead of $p(\boldsymbol{x}_t, t)$ and $p(\boldsymbol{x}_s, s \mid \boldsymbol{x}_t, t)$ when there is no ambiguity.

## 2.2 CONVERGENCE WITH KOLMOGOROV EQUATIONS

Define $f_t = \frac{d\log\alpha_t}{dt}$ and $g_t = -2f_t$, and the stochastic differential equation (SDE)

$$d\boldsymbol{x}_t = f_t \boldsymbol{x}_t \, dt + g_t \, d\boldsymbol{B}_t, \tag{8}$$

where $\boldsymbol{B}_t$ is the standard Brownian motion. According to Anderson (1982), its reverse process is

$$d\boldsymbol{x}_t = (f_t \boldsymbol{x}_t - g_t^2 \nabla_{\boldsymbol{x}_t} p(\boldsymbol{x}_t)) \, dt + g_t \, d\tilde{\boldsymbol{B}}_t. \tag{9}$$

Previous efforts to prove the convergence of DPM heavily depend on the Kolmogorov equations of Eq. (9) For instance, Lee et al. (2022) defines the discretization approximation

$$d\boldsymbol{x}_t = (f_{t-}\boldsymbol{x}_{t-} - g_{t-}^2 \nabla_{\boldsymbol{x}_t} p(\boldsymbol{x}_{t-})) \, dt + g_t \, d\tilde{\boldsymbol{B}}_t, \tag{10}$$

and establish the corresponding Kolmogorov forward equation for the single time marginal distribution of Eq. (10), denoted as $q(\boldsymbol{x}_t)$. Ultimately, the Chi-square divergence $\chi^2(q(\boldsymbol{x}_t)||p(\boldsymbol{x}_t))$ is estimated using the Kolmogorov equations. Numerous subsequent studies have embraced this configuration (Lee et al., 2023; Chen et al., 2022; 2023b;a). However, this proof has its limitations as it's based on the Kolmogorov equations. This means it cannot be applied to other types of discretizations where constructing the Kolmogorov equations is challenging. Therefore, a proof that can be readily adapted to a wider range of discretizations would be beneficial.

# 3 METHOD

In this section, we first introduce a novel method to demonstrate that the distribution generated by the conventional diffusion model closely matches the actual data distribution without using Kolmogorov equations. We then propose to approximate the reverse process transition kernel in a divide-and-conquer manner and prove its convergence using this novel method. We further propose merging strategies for these kernels and present the corresponding training and sampling methods.

To start with, we outline some assumptions regarding the initial distribution and the neural network approximation errors, which will be referenced throughout this paper:

**Assumption 1** *The initial distribution is a sum of Dirac deltas and* $\max_{i,j} ||\boldsymbol{y}_i - \boldsymbol{y}_j|| \leq M$ *for some positive constant* $M$.

We adopt the assumption about the initial distribution from Karras et al. (2022), as it is precisely the conditions during training, given the finite quantity of training data available in real-world situations. Additionally, we can consider $y_i$s as independent and identically distributed samples from any underlying continuous distribution $\tilde{p}_{data}$, and $p_{data}$ in equation (1) will weakly converge to $\tilde{p}_{data}$ (Varadarajan, 1958). Furthermore, our method can be extended to any initial distribution with compact support. The only requirement is to replace the sum over $y_i$s with an integral and verify the conditions for interchanging this integral with other integrals and derivatives. The second component of this assumption is that the gathered data is bounded, which is invariably the case in practical applications.

**Assumption 2** *For all* $t \in [0, 1]$, $\boldsymbol{y}_\theta$ *and* $\bar{\boldsymbol{y}}$ *are close in* $L^2(p)$:

$$\int_{\mathbb{R}^d} p(\boldsymbol{x}_t) ||\boldsymbol{y}_\theta(\boldsymbol{x}_t, t) - \bar{\boldsymbol{y}}(\boldsymbol{x}_t, t)||^2 \, \mathrm{d}\boldsymbol{x}_t < \varepsilon_y^2 < 1. \tag{11}$$

This assumption has been adopted by previous studies (Lee et al., 2022; Chen et al., 2022; Lee et al., 2023) and is confirmed by Oko et al. (2023).

**Assumption 3** $\alpha_t$ *is a predefined function, which decreases monotonically from 1 to 0, with its derivatives bounded; specifically,* $0 > \frac{\mathrm{d}\alpha_t}{\mathrm{d}t} \geq -C_\alpha$ *for some positive constant* $C_\alpha$.

In practice, $\alpha_t$ is designed to be continuous and monotonically decreasing. This assumption is naturally satisfied unless an unusual scheduler induces an unbounded derivative at $t = 0$ or $t = 1$.

## 3.1 CONVERGENCE OF DPM FROM A NOVEL PERSPECTIVE

Considering that the sampling process occurs in discrete steps, we introduce the notation for time discretization as $\mathcal{D} = \{0 < t_{\min} = t_0 < t_1 < \cdots < t_T = t_{\max} < 1\}$. Subsequently, the approximated single-time marginal distribution with the accurate $\bar{\boldsymbol{y}}(x_t, t)$ is:

$$\tilde{p}(\boldsymbol{x}_{t_i}) = \int_{\mathbb{R}^d} \cdots \int_{\mathbb{R}^d} \tilde{p}(\boldsymbol{x}_{t_i}|\boldsymbol{x}_{t_{i+1}}) \cdots \tilde{p}(\boldsymbol{x}_{t_{T-1}}|\boldsymbol{x}_{t_T}) \tilde{p}(\boldsymbol{x}_{t_T}) \, \mathrm{d}\boldsymbol{x}_{t_{i+1}} \cdots \mathrm{d}\boldsymbol{x}_{t_T}. \tag{12}$$

By substituting $\bar{\boldsymbol{y}}(x_t, t)$ in Eq. (12) and (6) with the network prediction $\boldsymbol{y}_\theta(\boldsymbol{x}_t, t)$, we obtain $p_\theta(\boldsymbol{x}_{t_i})$ and $p_\theta(\boldsymbol{x}_{t_i}|\boldsymbol{x}_{t_{i+1}})$. We also define $\Delta t_i = t_{i+1} - t_i$ and denote the maximum $\Delta t_i$ as $|\mathcal{D}|$.

As pointed out in previous study (Zhang et al., 2024), singularities arise near $t = 0$ and $t = 1$, necessitating specific treatment. To address this, we divide the time interval into three distinct segments: the left interval $[0, t_{\min})$, the middle interval $[t_{\min}, t_{\max}]$, and the right interval $(t_{\max}, 1]$. Each section is handled independently. Previous work provides local error estimates for the middle and right intervals (Zhang et al., 2024). However, their assertions are not strong enough to achieve global convergence. We enhance these estimates to ensure global convergence. We refine the error bound for the middle interval from $(t - s)^{\frac{1}{4}}$ (Zhang et al., 2024) to $(t - s)^{\frac{3}{2} - \beta}$.

**Proposition 1** *For all $t_{min} \leq s < t \leq t_{max}$ and $0 < \beta < 1$, there exist $\delta > 0$ and $C_1, C_2 > 0$ depending on $\beta$, $t_{min}$ and $t_{max}$, such that if $t - s < \delta$, the inequality $KL(p(\boldsymbol{x}_s)||p_\theta(\boldsymbol{x}_s)) \leq KL(p(\boldsymbol{x}_t)||p_\theta(\boldsymbol{x}_t)) + C_1(t-s)^{\frac{3-\beta}{2}} + C_2(t-s)\varepsilon_y$ holds.*

To obtain this error bound, it's necessary to estimate the integral within the Kullback-Leibler divergence, which integrates over the entire $\mathbb{R}^d$ domain. Thanks to the light tail of Gaussian distributions, we design a $B$, which is related to $|\mathcal{D}|$, and bound the integral over $\{\boldsymbol{x} \in \mathbb{R}^d : ||\boldsymbol{x}|| \geq B\}$ (Lemma 2). As for $\{\boldsymbol{x} \in \mathbb{R}^d : ||\boldsymbol{x}|| < B\}$, since $||\boldsymbol{x}||$ is small in this case (Lemma 3), we use a Taylor expansion (Lemma 5) and find that the relationship $\bar{\boldsymbol{y}}(\boldsymbol{x}_t, t) = \sum_i w_i(\boldsymbol{x}_t, t)\boldsymbol{y}_i$ is crucial for canceling out the lower order terms of the error bound (Lemma 4). This justifies the training objective for diffusion models from another perspective.

For the right interval, we improve the error bound from $(1 - s)^{\frac{1}{2}}$ (Zhang et al., 2024), to $(1 - s)^2$.

**Proposition 2** *For all $0 < s < 1$, there are constants $C_1, C_2 > 0$, such that $KL(p(\boldsymbol{x}_s)||p_\theta(\boldsymbol{x}_s)) \leq C_1(1-s)^2 + C_2(1-s)^2\varepsilon_y$.*

As for the left interval, it's not feasible to compute the Kullback-Leibler divergence for Dirac deltas. We adapt the idea from Theorem 2.1 in prior work (Lee et al., 2023), which applies the Wasserstein distance to the left interval.

**Proposition 3** *Given $0 < t_{min} < 1$, the 2-Wasserstein distance*

$$W_2(p(\boldsymbol{x}_0), p(\boldsymbol{x}_{t_{min}})) < \sqrt{2dC_\alpha t_{min}}. \tag{13}$$

By combining the local error bounds above, we can establish global convergence as follows:

**Proposition 4** *For all $0 < \beta < 1$, there exist $\delta > 0$ and $C_1, C_2, C_3 > 0$, such that for all time discretizations $\mathcal{D}$ with $|\mathcal{D}| < \delta$, the Kullback-Leibler divergence $KL(p(\boldsymbol{x}_{t_{min}})||p_\theta(\boldsymbol{x}_{t_{min}})) \leq C_1|\mathcal{D}|^{\frac{1-\beta}{2}} + C_2\varepsilon_y$. Moreover, $W_2^2(p(\boldsymbol{x}_0), p(\boldsymbol{x}_{t_{min}})) < C_3|\mathcal{D}|$.*

## 3.2 DIVIDE-AND-CONQUER APPROXIMATION AND ITS CONVERGENCE

Based on the analysis above, besides the single Gaussian transition kernel, any distribution submitted to Proposition 4 could serve as the transition kernel.

As demonstrated in Eq. (5), the ground truth transition kernel of the diffusion reverse process is a mixture of standard Gaussian distributions. Previous work proves that traditional approaches approximate this Gaussian mixture kernel with a single Gaussian distribution and can significantly diverge from the true reverse transition kernel (Guo et al., 2024). This motivates our *divide-and-conquer* (DC) transition kernel approximation.

Specifically, we propose to partition data and use cluster-specific kernels to represent data samples in each segment. The true kernel is then approximated by integrating these cluster-specific kernels. Consider a scenario where the training data is divided into $L$ classes: $\{\boldsymbol{y}_i \in \mathbb{R}^d | i = 1, 2, \ldots, N\} = \bigcup_{l=1}^{L}\{\boldsymbol{y}_i^l \in \mathbb{R}^d | i = 1, 2, \ldots, N_l\}$. This partition can be arbitrary, and we will prove that any method of data division result in convergence. We define a new approximation

$$\hat{p}(\boldsymbol{x}_s|\boldsymbol{x}_t) = \sum_{l=1}^{L} a^l(\boldsymbol{x}_t, t)\hat{p}^l(\boldsymbol{x}_s|\boldsymbol{x}_t), \tag{14}$$

where

$$\hat{p}^l(\boldsymbol{x}_s|\boldsymbol{x}_t) = \sum_{i=1}^{N_l} (2\pi\sigma_{s|t})^{-\frac{d}{2}} u_i^l(\boldsymbol{x}_t, t) \exp\{-\frac{1}{2\sigma_{s|t}^2}||\boldsymbol{x}_s - \frac{\alpha_{t|s}\sigma_s^2}{\sigma_t^2}\boldsymbol{x}_t - \frac{\alpha_s\sigma_{t|s}^2}{\sigma_t^2}\bar{\boldsymbol{y}}^l(x_t, t)||^2\}, \tag{15}$$

$$w_i^l(\boldsymbol{x}_t, t) = \frac{\exp-\frac{||\boldsymbol{x}_t - \alpha_t \boldsymbol{y}_i^l||^2}{2\sigma_t^2}}{\sum_{l,j} \exp-\frac{||\boldsymbol{x}_t - \alpha_t \boldsymbol{y}_j^l||^2}{2\sigma_t^2}}, a^l(\boldsymbol{x}_t, t) = \sum_{i=1}^{N_l} w_i^l(\boldsymbol{x}_t, t), u_i^l(\boldsymbol{x}_t, t) = \frac{w_i^l(\boldsymbol{x}_t, t)}{a^l(\boldsymbol{x}_t, t)} \text{ and } \bar{\boldsymbol{y}}^l(\boldsymbol{x}_t, t) =$$

$\sum_{i=1}^{N_l} u_i^l(\boldsymbol{x}_t, t)\boldsymbol{y}_i^l$. The single time marginal distribution of this approximation $\hat{p}(\boldsymbol{x}_t)$ is defined in the same way as in Eq. (12). Each cluster-specific kernel $\hat{p}^l(\boldsymbol{x}_s|\boldsymbol{x}_t)$ can be approximated using any method suitable for a standard diffusion probabilistic model, including a single Gaussian approximation in DDPM, single Gaussian with optimized variances in Extended-Analytic-DPM (Bao et al., 2022a), as well as GMS (Guo et al., 2024), which computes high-order moments and estimates $\hat{p}^l(\boldsymbol{x}_s|\boldsymbol{x}_t)$ as a mixture of two Gaussians with hand-crafted weights.

In this scenario, we need L neural networks to approximate $\bar{\boldsymbol{y}}^l(\boldsymbol{x}_t, t)$. In practice, we use a conditional network $\boldsymbol{y}_\theta(\boldsymbol{x}_t, t, l)$ (also denoted as $\boldsymbol{y}_\theta^l(\boldsymbol{x}_t, t)$). Additionally, a neural network $\boldsymbol{a}_\phi(\boldsymbol{x}_t, t)$ is necessary to approximate $\boldsymbol{a}(\boldsymbol{x}_t, t) \overset{def}{=} (a^1(\boldsymbol{x}_t, t), \cdots, a^L(\boldsymbol{x}_t, t))^T \in \mathbb{R}^L$. To derive the error bound, we also make the assumption that $\boldsymbol{y}_\theta^l(\boldsymbol{x}_t, t)$ and $\boldsymbol{a}_\phi(\boldsymbol{x}_t, t)$ approximate $\bar{\boldsymbol{y}}^l(x_t, t)$ and $a^l(x_t, t)$ in $L^2(p)$.

**Assumption 4** *For all $t \in [t_{min}, 1]$ and $1 \leq l \leq L$, $\boldsymbol{y}_\theta^l$ and $a_\phi^l$ are close to $\bar{\boldsymbol{y}}^l$ and $\boldsymbol{a}^l$ in $L^2(p)$ respectively:*

$$\int_{\mathbb{R}^d} p(\boldsymbol{x}_t)||\boldsymbol{y}_\theta^l(\boldsymbol{x}_t, t) - \bar{\boldsymbol{y}}^l(\boldsymbol{x}_t, t)||^2 \, d\boldsymbol{x}_t < \varepsilon_{yl}^2 < 1, \tag{16}$$

*and*

$$\int_{\mathbb{R}^d} p(\boldsymbol{x}_t)(a_\phi^l(\boldsymbol{x}_t, t) - a^l(\boldsymbol{x}_t, t))^2 \, d\boldsymbol{x}_t < \varepsilon_{al}^2 < 1. \tag{17}$$

*Moreover, $a_\phi^l(\boldsymbol{x}_t, t)$ and $a^l(x_t, t)$ are uniformly lower bounded by a constant $C_a$.*

Then, $\hat{p}_\theta(\boldsymbol{x}_s|\boldsymbol{x}_t)$ is defined by $\boldsymbol{y}_\theta^l$s and $a_\phi^l$s. $\hat{p}_\theta(\boldsymbol{x}_t)$ is defined in a manner consistent with equation (12).

To estimate the error boundary of the Divide-and-Conquer Diffusion Probabilistic Models (DC-DPM), we employ a strategy that transforms it into a single Gaussian case. Taking into account that

$$p(\boldsymbol{x}_s|\boldsymbol{x}_t) = \sum_{l=1}^{L} a^l(\boldsymbol{x}_t, t)p^l(\boldsymbol{x}_s|\boldsymbol{x}_t), \tag{18}$$

where

$$p^l(\boldsymbol{x}_s|\boldsymbol{x}_t) = \sum_{i=1}^{N_l} (2\pi\sigma_{s|t})^{-\frac{d}{2}} u_i^l(\boldsymbol{x}_t, t) \exp\{-\frac{1}{2\sigma_{s|t}^2}||\boldsymbol{x}_s - \frac{\alpha_{t|s}\sigma_s^2}{\sigma_t^2}\boldsymbol{x}_t - \frac{\alpha_s\sigma_{t|s}^2}{\sigma_t^2}\boldsymbol{y}_i^l||^2\}, \tag{19}$$

and given the convexity of the Kullback-Leibler divergence, we can deduce that

$$KL(p(\boldsymbol{x}_s|\boldsymbol{x}_t)||\hat{p}(\boldsymbol{x}_s|\boldsymbol{x}_t)) \leq \sum_l a^l(\boldsymbol{x}_t, t)KL(p^l(\boldsymbol{x}_s|\boldsymbol{x}_t)||\hat{p}^l(\boldsymbol{x}_s|\boldsymbol{x}_t)). \tag{20}$$

Eq. (20) allows us to bound the error of the divide-and-conquer approximations by the sum of its individual components. Utilizing Propositions 1, 2, and 4, we can deduce the following corollaries:

**Corollary 1** *For all $t_{min} \leq s < t \leq t_{max}$ and $0 < \beta < 1$, there exist $\delta > 0$ and $C_1, C_2, C_3 > 0$ depending on $\beta$, $t_{min}$ and $t_{max}$, such that if $t - s < \delta$, the inequality $KL(p(\boldsymbol{x}_s)||\hat{p}_\theta(\boldsymbol{x}_s)) \leq KL(p(\boldsymbol{x}_t)||\hat{p}_\theta(\boldsymbol{x}_t)) + C_1(t - s)^{\frac{3-\beta}{2}} + C_2(t - s)\varepsilon_{yl} + C_3\varepsilon_{al}$ holds.*

**Corollary 2** *For all $0 < s < 1$, there are constants $C_1, C_2, C_3 > 0$, such that $KL(p(\boldsymbol{x}_s)||p_\theta(\boldsymbol{x}_s)) \leq C_1(1 - s)^2 + C_2\varepsilon_{yl} + C_3\varepsilon_{al}$.*

**Corollary 3** *For all* $0 < \beta < 1$, *there exist* $\delta > 0$ *and* $C_1, C_2, C_3, C_4 > 0$, *such that for all time discretizations* $\mathcal{D}$ *with* $|\mathcal{D}| < \delta$, *the Kullback-Leibler divergence* $KL(p(\boldsymbol{x}_{t_{min}})||\hat{p}_\theta(\boldsymbol{x}_{t_{min}})) \leq C_1|\mathcal{D}|^{\frac{1-\beta}{2}} + C_2\varepsilon_{yl} + C_3T\varepsilon_{al}$. *Moreover,* $W_2(p(\boldsymbol{x}_0), p(\boldsymbol{x}_{t_{min}})) < C_4|\mathcal{D}|$.

It is worthy to note that Corollaries above can not be easily proved with the methods based on Kolmogorov equations as in the previous works, because it is not trivial to construct a Ito diffusion with equation (14) being the solution to its corresponding Kolmogorov equations.

The term $\varepsilon_{al}$ in Corollary 3 includes a coefficient $T$, representing the inference time step. This factor inhibits the error from converging to zero as $|\mathcal{D}|$ approaches zero. However, due to the simplistic structure of $\boldsymbol{a}(\boldsymbol{x}_t, t)$, the network $\boldsymbol{a}_\phi(\boldsymbol{x}_t, t)$ is relatively easy to train. This results in $\varepsilon_{al}$ being significantly smaller than $\varepsilon_{yl}$. Consequently, this maintains the error of DC-DPM at a reasonably low value.

### 3.3 MERGING CLUSTER-SPECIFIC KERNELS

The divide-and-conquer representation of the reversed transition in Eq. (14) consists of combination coefficients $a^l(\boldsymbol{x}_t, t)$, referred to as the *class part*, and cluster-specific kernels $\hat{p}^l(\boldsymbol{x}_s|\boldsymbol{x}_t)$, referred to as the *diffusion part*. For the diffusion part, to learn $L$ cluster-specific kernels, we propose training a single conditional network $\boldsymbol{y}_\theta(\boldsymbol{x}_t, t, l)$ to represent them, rather than training $L$ independent networks, in order to save computational overhead. For the class part, we propose two approaches to estimate it: *label diffusion approximation (LD)* and *fixed class approximation (FC)*.

Label diffusion approximation (LD) learns the class part in a manner similar to the diffusion part. Define $L_i$ as the one-hot vector representing the class to which data point $y_i$ belongs. Then we construct $a(x_t, t)$ as:

$$\boldsymbol{a}(\boldsymbol{x}_t, t) = \sum_i w_i(\boldsymbol{x}_t, t)L_i, \tag{21}$$

where $w_i(\boldsymbol{x}_t, t)$ represents the coefficients in the ground truth transition kernel in Eq. (5). Substituting Eq. (21) into Eq. (18) aligns with the ground truth transition kernel in Eq. (5). Given that the structure of $\boldsymbol{a}(\boldsymbol{x}_t, t)$ closely mirrors that of $\bar{\boldsymbol{y}}(\boldsymbol{x}_t, t)$, a neural network $\boldsymbol{a}_\phi(\boldsymbol{x}_t, t)$ can be trained in a manner analogous to the $\boldsymbol{x}$-prediction networks in diffusion models. The training process to learn the class part can be formulated as:

**Proposition 5** *Let* $L(\boldsymbol{x}_0)$ *denote the one-hot class vector of* $\boldsymbol{x}_0$, *the optimal* $\boldsymbol{a}_\phi(\boldsymbol{x}_t, t)$ *for the two objective functions*

$$\mathcal{L}_2 = \mathbb{E}_{\boldsymbol{x}_0 \sim p_{data}, \boldsymbol{x}_t \sim p(\boldsymbol{x}_t|\boldsymbol{x}_0), t \sim \mathbf{U}(0,1)}||L(\boldsymbol{x}_0) - \boldsymbol{a}_\phi(\boldsymbol{x}_t, t)||^2, \tag{22}$$

*and*

$$\mathcal{L}_{CE} = \mathbb{E}_{\boldsymbol{x}_0 \sim p_{data}, \boldsymbol{x}_t \sim p(\boldsymbol{x}_t|\boldsymbol{x}_0), t \sim \mathbf{U}(0,1)} CE(L(\boldsymbol{x}_0), \boldsymbol{a}_\phi(\boldsymbol{x}_t, t)), \tag{23}$$

*are the same and equal to* $\boldsymbol{a}(\boldsymbol{x}_t, t)$, *where CE represents the cross-entropy loss.*

Based on the analysis above, we present the algorithms for training the diffusion model $\boldsymbol{y}_\theta(\boldsymbol{x}_t, t, l)$ in Algorithm 1 and the label model $\boldsymbol{a}_\phi(\boldsymbol{x}_t, t)$ in Algorithm 2.

---

**Algorithm 1** Training of diffusion model $y_\theta$

1: **Repeat**
2: $\boldsymbol{x}_0 \sim p_{data}$
3: $t \sim \text{Uniform}(t_1, t_2, ..., t_T)$
4: $\boldsymbol{x}_t \sim p(\boldsymbol{x}_t|\boldsymbol{x}_0)$
5: Take gradient descent step on
   $\nabla_\theta \left\| \boldsymbol{y}_\theta(\boldsymbol{x}_t, t, l(\boldsymbol{x}_0)) - \boldsymbol{x}_0 \right\|^2$
6: **Until** converged

---

**Algorithm 2** Training of label model $a_\theta$

1: **Repeat**
2: $\boldsymbol{x}_0 \sim p_{data}$
3: $t \sim \text{Uniform}(t_1, t_2, ..., t_T)$
4: $\boldsymbol{x}_t \sim p(\boldsymbol{x}_t|\boldsymbol{x}_0)$
5: Take gradient descent step on
   $\nabla_\phi \left\| \boldsymbol{a}_\phi(\boldsymbol{x}_t, t) - L(\boldsymbol{x}_0) \right\|^2$
6: **Until** converged

---

The second approach, *fixed class approximation (FC)*, first samples a label $l$ from Eq. (21) at $t = 1$. In this scenario, $\boldsymbol{a}(x_1, 1) = (b^1, b^2, \cdots, b^L)^T$, where $b^l = \frac{N_l}{N}$ represents the proportion of samples in cluster $l$ relative to the total number of samples in the dataset. Then the label adheres to this value along time $t$:

$$\boldsymbol{a}(\boldsymbol{x}_t, t) = \boldsymbol{e}^{(l)}. \tag{24}$$

Since the sampled label remains consistent over time $t$ in the FC approximation, the class part $\boldsymbol{a}_\phi(\boldsymbol{x}_t, t)$ is necessitated solely at $t = 1$.

### 3.4 SAMPLING METHOD FOR DC-DPM

Conventionally, DDPM reverse approximation in Eq. (6) can be realized by the trajectory:

$$\boldsymbol{x}_{t_{i-1}} = \frac{\alpha_{t_i|t_{i-1}}\sigma_{t_{i-1}}^2}{\sigma_{t_i}^2}\boldsymbol{x}_{t_i} + \frac{\alpha_{t_{i-1}}\sigma_{t_i|t_{i-1}}^2}{\sigma_{t_i}^2}\boldsymbol{y}_\theta(\boldsymbol{x}_{t_i}, t_i) + \sigma_{t_{i-1}|t_i}\boldsymbol{z}_{t_i}, \tag{25}$$

where $\boldsymbol{z}_{t_i} \sim \mathcal{N}(\boldsymbol{0}, \boldsymbol{I})$ and $\boldsymbol{x}_{t_T} \sim p(\boldsymbol{x}_{t_T}, t_T)$. Our reverse process, using a mixture of cluster-specific kernels, requires an additional random variable $\boldsymbol{y}(\boldsymbol{x}_{t_i}, t_i)$ to represent the trajectory:

$$\boldsymbol{x}_{t_{i-1}} = \frac{\alpha_{t_i|t_{i-1}}\sigma_{t_{i-1}}^2}{\sigma_{t_i}^2}\boldsymbol{x}_{t_i} + \frac{\alpha_{t_{i-1}}\sigma_{t_i|t_{i-1}}^2}{\sigma_{t_i}^2}\boldsymbol{y}(\boldsymbol{x}_{t_i}, t_i) + \sigma_{t_{i-1}|t_i}\boldsymbol{z}_{t_i}. \tag{26}$$

The density of $\boldsymbol{y}(\boldsymbol{x}_{t_i}, t_i)$ is

$$p(\boldsymbol{y}(\boldsymbol{x}_{t_i}, t_i) = \boldsymbol{y}) = \sum_l a^l(\boldsymbol{x}_{t_i}, t_i)\delta(\boldsymbol{y} - \boldsymbol{y}_\theta(\boldsymbol{x}_{t_i}, t_i, l)), \tag{27}$$

and $\boldsymbol{y}(\boldsymbol{x}_{t_i}, t_i)$ is independent of $\boldsymbol{z}_{t_i}$.

For label diffusion approximation (LD), our method samples two random variables in each step: weight sampling in line 3 and diffusion sampling in line 5 as shown in Algorithm 3. Fixed class approximation (FC) samples the weight from the discrete distribution $(b^1, b^2, \cdots, b^L)^T$. Since the weight term remains consistent over time $t$, weight sampling is executed only once, as shown in line 2 of Algorithm 4. After weight sampling, the model generates samples within one fixed class $l_0$. The generated distribution is:

$$\hat{p}^{l_0}(\boldsymbol{x}_{t_i}) = \int_{\mathcal{R}^d} \cdots \int_{\mathcal{R}^d} \hat{p}^{l_0}(\boldsymbol{x}_{t_i}|\boldsymbol{x}_{t_{i+1}}) \cdots \hat{p}^{l_0}(\boldsymbol{x}_{t_{T-1}}|\boldsymbol{x}_{t_T})p^{l_0}(\boldsymbol{x}_{t_T}, t_T)\,\mathrm{d}\boldsymbol{x}_{t_{i+1}} \cdots \mathrm{d}\boldsymbol{x}_{t_T}. \tag{28}$$

According to Proposition 4, $\hat{p}^{l_0}(\boldsymbol{x}_0)$ approximates $p_{data}^{l_0}(\boldsymbol{x}) = \frac{1}{N_{l_0}}\sum_i \delta(\boldsymbol{x} - \boldsymbol{y}_i^{l_0})$. Thus the distribution $\sum_l b^l \hat{p}^l(\boldsymbol{x}_0)$ approximate $p(\boldsymbol{x}_0) = \sum_l b^l p_{data}^l$, where $b^l = \frac{N_l}{N}$.

---

**Algorithm 3** Sampling process of label diffusion approximation (LD)

---

1: $\boldsymbol{x}_{t_T} \sim \mathcal{N}(\boldsymbol{0}, \boldsymbol{I})$
2: **for** $i = T, ..., 1$ **do**
3:    $l \sim \boldsymbol{a}_\theta(\boldsymbol{x}_{t_i}, t_i)$
4:    $\boldsymbol{z} \sim \mathcal{N}(\boldsymbol{0}, \boldsymbol{I})$
5:    $\boldsymbol{x}_{t_{i-1}} = \frac{\alpha_{t_i|t_{i-1}}\sigma_{t_{i-1}}^2}{\sigma_{t_i}^2}\boldsymbol{x}_{t_i}$
     $+ \frac{\alpha_{t_{i-1}}\sigma_{t_i|t_{i-1}}^2}{\sigma_{t_i}^2}\boldsymbol{y}_\theta(x_{t_i}, t_i, l) + \sigma_{t_{i-1}|t_i}\boldsymbol{z}$
6: **end for**
7: **return** $\boldsymbol{x}_{t_0}$

---

**Algorithm 4** Sampling process of fixed class approximation (FC)

---

1: $\boldsymbol{x}_{t_T} \sim \mathcal{N}(\boldsymbol{0}, \boldsymbol{I})$
2: $l \sim (b^1, b^2, \cdots, b^C)$
3: **for** $i = T, ..., 1$ **do**
4:    $\boldsymbol{z} \sim \mathcal{N}(\boldsymbol{0}, \boldsymbol{I})$
5:    $\boldsymbol{x}_{t_{i-1}} = \frac{\alpha_{t_i|t_{i-1}}\sigma_{t_{i-1}}^2}{\sigma_{t_i}^2}\boldsymbol{x}_{t_i}$
     $+ \frac{\alpha_{t_{i-1}}\sigma_{t_i|t_{i-1}}^2}{\sigma_{t_i}^2}\boldsymbol{y}_\theta(\boldsymbol{x}_{t_i}, t_i, l) + \sigma_{t_{i-1}|t_i}\boldsymbol{z}$
6: **end for**
7: **return** $\boldsymbol{x}_{t_0}$

---

**Algorithm 5** Sampling process for LD approximation with ODE-based methods

1: $\boldsymbol{x}_{t_T} \sim \mathcal{N}(\boldsymbol{0}, \boldsymbol{I})$
2: **for** $i = T, ..., 1$ **do**
3:   $\boldsymbol{z} \sim \mathcal{N}(\boldsymbol{0}, \boldsymbol{I})$
4:   $\boldsymbol{x}_{t_{i-1}} = \boldsymbol{ODE}(\boldsymbol{x}_{t_i}, t_i, l)$
5: **end for**
6: **return** $\boldsymbol{x}_{t_0}$

**Algorithm 6** Sampling process for FC approximation with ODE-based methods

1: $\boldsymbol{x}_{t_T} \sim \mathcal{N}(\boldsymbol{0}, \boldsymbol{I})$
2: $l \sim (b^1, b^2, \cdots, b^L)$
3: **for** $i = T, ..., 1$ **do**
4:   $\boldsymbol{x}_{t_{i-1}} = \boldsymbol{ODE}(\boldsymbol{x}_{t_i}, t_i, l)$
5: **end for**
6: **return** $\boldsymbol{x}_{t_0}$

As the probability flow ODE keeps the single-time marginals (Song et al., 2020b), we can replace the diffusion sampling method with probability flow ODE-based methods, such as DDIM (Song et al., 2020a), DPM Solver (Lu et al., 2022a), PNDM (Liu et al., 2022) etc. We summarize this in Algorithm 5, where $ODE(\boldsymbol{x}_t, t, l)$ represents the ODE-based sampling methods. Similarly, the fixed class approximation is also applicable to ODE-based methods, as presented in Algorithm 6

## 4 EXPERIMENTS

### 4.1 IMAGE-SPACE RESULTS

Table 1: **FID ↓ on CIFAR-10 Dataset**. Employing our Divide-and-Conquer (DC) kernel approximation strategy on previous DPM methods enhances their generation quality especially on small timesteps. LD represents merging kernels with label diffusion approximation while FC represents fixed class approximation. SN-DDPM is short for Extended-Analytic-DPM (Bao et al., 2022a).

| | CIFAR-10 (Linear Schedule) | | | | | | CIFAR-10 (Cosine Schedule) | | | | | |
|---|---|---|---|---|---|---|---|---|---|---|---|---|
| # TIMESTEPS | 10 | 25 | 50 | 100 | 200 | 1000 | 10 | 25 | 50 | 100 | 200 | 1000 |
| DDPM | 43.14 | 21.63 | 15.21 | 10.94 | 8.23 | 5.11 | 34.76 | 16.18 | 11.11 | 8.38 | 6.66 | 4.92 |
| **+DC-LD** (Ours) | 39.40 | 21.95 | 15.54 | 10.78 | 7.91 | 4.98 | 27.78 | 15.52 | 10.12 | 7.29 | 5.61 | 4.11 |
| **+DC-FC** (Ours) | **34.48** | **21.05** | **15.12** | **10.67** | **7.82** | **4.50** | **25.80** | **14.58** | **9.66** | **6.72** | **5.03** | **3.46** |
| SN-DDPM | 21.87 | 6.91 | 4.58 | 3.74 | 3.34 | 3.71 | 16.33 | 6.05 | 4.19 | 3.83 | 3.72 | 4.08 |
| **+DC-LD** (Ours) | 16.77 | 6.39 | 4.29 | 3.40 | 2.97 | 3.30 | 12.85 | 6.54 | 4.56 | 3.63 | 3.35 | 3.51 |
| **+DC-FC** (Ours) | **11.90** | **4.98** | **3.62** | **2.98** | **2.55** | **2.93** | **9.92** | **4.95** | **3.35** | **2.67** | **2.53** | **2.74** |
| GMS | 17.43 | 5.96 | 4.16 | 3.26 | 3.01 | 2.76 | 13.80 | 5.48 | 4.00 | 3.46 | 3.34 | 4.23 |
| **+DC-LD** (Ours) | 14.54 | 5.89 | 4.22 | 3.41 | 3.58 | 5.19 | 10.80 | 6.22 | 4.53 | 3.64 | 3.34 | 4.35 |
| **+DC-FC** (Ours) | **10.40** | **4.84** | **3.61** | **3.00** | **3.00** | **2.86** | **8.76** | **4.91** | **3.43** | **2.76** | **2.60** | **3.35** |

We quantitatively compare the sample quality using the widely recognized Fréchet Inception Distance (FID) score (Heusel et al., 2017). Utilizing the semantic labels from the CIFAR-10 dataset, we categorize the data into 10 classes. We then apply our proposed divide-and-conquer approximation to various transition kernel designs, including DDPM (Ho et al., 2020), Extended-Analytic-DPM (Bao et al., 2022a), and GMS (Guo et al., 2024). These kernels are merged using both the label diffusion (LD) and fixed class (FC) approximation strategies. As illustrated in Table 1, our DC-DPM approach significantly enhances the performance of existing methods, particularly at smaller denoising timesteps. Specifically, DC-DPM achieves improvements of 25.78% for DDPM, 45.58% for Extended-Analytic-DPM, and 40.33% for GMS in scenarios with 10 denoising steps.

### 4.2 LATENT-SPACE RESULTS

We also apply DC-DPM to latent diffusion models (Rombach et al., 2022). We perform comparative experiments for unconditional generation on the CelebA-HQ-256 image dataset. To classify the data, we first compute the VAE latent space of each image Kingma & Welling (2013), extract the primary dimension using principal component analysis (PCA) Abdi & Williams (2010), and then cluster the images into 10 classes using the K-Means algorithm. Both the quantitative results in Table 3 demonstrate that DC-DPM improves the generation quality of diffusion models in latent space.

Table 2: **FID ↓ on CIFAR-10 (Linear Schedule) with DDIM.** DC-DPM can be applied to ODE-based samplers like DDIM.

| # STEPS | 10 | 25 | 50 |
|---|---|---|---|
| DDIM | 21.31 | 10.70 | 7.74 |
| **+DC-LD** (Ours) | 20.43 | 11.39 | 8.38 |
| **+DC-FC** (Ours) | **16.54** | **9.15** | **6.60** |

Table 3: **FID ↓ on CelebA-HQ-256.** DC-DPM is applicable to latent diffusion models to improve the generation quality.

| # STEPS | 10 | 25 | 50 |
|---|---|---|---|
| DDPM | 35.21 | 18.60 | 14.16 |
| **+DC-LD** (Ours) | 30.58 | 15.76 | 12.25 |
| **+DC-FC** (Ours) | **30.47** | **15.37** | **12.16** |

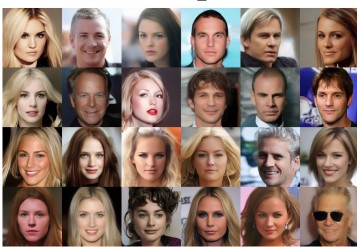

Figure 2: **Qualitative Results with DDIM.** Applying DC-DPM to ODE-based methods like DDIM, enables higher quality generation on small timesteps for latent diffusion model on CelebA-HQ-256.

### 4.3 DETERMINISTIC SAMPLING METHODS

While we have only validated our DC-DPM with stochastic sampling methods, it is not guaranteed to work with deterministic methods like DDIM. However, our experiments indicate that DC-DPM can indeed function with deterministic sampling. As shown in Table 2, applying DC-DPM to the DDIM sampler improves generation quality by 22.38% with 10 steps on the CIFAR-10 dataset. Figure 2 provides qualitative results, further demonstrating the compatibility of DC-DPM with DDIM.

## 5 RELATED WORK

Significant research has focused on improving diffusion model performance on fewer timesteps, broadly categorized into three approaches. Training-based methods includes trainable sampling schedules (Watson et al., 2021), truncated diffusion (Lyu et al., 2022; Zheng et al., 2022), neural operators (Zheng et al., 2023a), and distillation (Salimans & Ho, 2022; Sauer et al., 2023; Meng et al., 2023; Song et al., 2023; Luo et al., 2023). The second category enhances the efficiency of SDE and ODE solvers in the reverse process, including faster SDE and ODE solvers (Lu et al., 2022a;b; Zheng et al., 2023b; Xu et al., 2023; Li et al., 2024), adaptive step size solvers (Jolicoeur-Martineau et al., 2021), predictor-corrector methods (Song et al., 2020b; Zhao et al., 2023), and stochastic-calculus-based optimization (Sabour et al., 2024).

The third category focuses on improving the design of the transition kernel in the diffusion reverse process. Analytic-DPM (Bao et al., 2022b) and Extended-Analytic-DPM (Bao et al., 2022a) estimate the optimal variance. Our work also falls within this category, with the most closely related prior work being GMS (Guo et al., 2024). GMS represents the transition kernel as a mixture of two Gaussians based on the estimation of higher-order moments. In contrast, the highlight of our method is to divide data into clusters and construct the kernel function in a divide-and-conquer manner. We construct a more general framework and previous Analytic-DPM, Extended-Analytic-DPM, and GMS can serve as the cluster-specific kernel in our method.

## 6 CONCLUSION

In this paper, we propose DC-DPM, a novel divide-and-conquer approach for approximating the transition kernel in the reverse process of diffusion probabilistic models. We provide convergence proof for diffusion models from a new perspective, generalizing the transition kernel representation from a conventional single Gaussian to a divide-and-conquer framework. This framework utilizes cluster-specific kernels to represent segmented data, which are then merged to form an overall representation. We propose two merging strategies along with their corresponding training and sampling methods. Experimental results demonstrate the effectiveness of our approach, significantly enhancing generation quality, particularly over a limited number of timesteps.

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

# A  PROOFS

## A.1  PROOF OF PROPOSITION 1

**Lemma 1** *For all positive integrable functions* $f(\boldsymbol{x})$, $g(\boldsymbol{x}) : \mathbb{R}^d \to \mathbb{R}_+$*, we have*

$$\int_{\mathbb{R}^d} f(\boldsymbol{x}) \, \mathrm{d}\boldsymbol{x} \log \frac{\int_{\mathbb{R}^d} f(\boldsymbol{x}) \, \mathrm{d}\boldsymbol{x}}{\int_{\mathbb{R}^d} g(\boldsymbol{x}) \, \mathrm{d}\boldsymbol{x}} \leq \int_{\mathbb{R}^d} f(\boldsymbol{x}) \log \frac{f(\boldsymbol{x})}{g(\boldsymbol{x})} \, \mathrm{d}\boldsymbol{x} \tag{29}$$

*proof.* Let $F = \int_{\mathbb{R}^d} f(\boldsymbol{x}) \, \mathrm{d}\boldsymbol{x}$, $G = \int_{\mathbb{R}^d} g(\boldsymbol{x}) \, \mathrm{d}\boldsymbol{x}$ and $h(t) = t \log t$. $h(t)$ is convex because

$$\frac{\mathrm{d}^2}{\mathrm{d}t^2} h(t) = \frac{1}{t} > 0. \tag{30}$$

And then

$$\int_{\mathbb{R}^d} f(\boldsymbol{x}) \log \frac{f(\boldsymbol{x})}{g(\boldsymbol{x})} \, \mathrm{d}\boldsymbol{x} = \int_{\mathbb{R}^d} g(\boldsymbol{x}) h(\frac{f(\boldsymbol{x})}{g(\boldsymbol{x})}) \, \mathrm{d}\boldsymbol{x}$$
$$= G \int_{\mathbb{R}^d} \frac{g(\boldsymbol{x})}{G} h(\frac{f(\boldsymbol{x})}{g(\boldsymbol{x})}) \, \mathrm{d}\boldsymbol{x}. \tag{31}$$

According to the probabilistic form of Jensen's inequality

$$G \int_{\mathbb{R}^d} \frac{g(\boldsymbol{x})}{G} h(\frac{f(\boldsymbol{x})}{g(\boldsymbol{x})}) \, \mathrm{d}\boldsymbol{x} \geq G h(\int_{\mathbb{R}^d} \frac{g(\boldsymbol{x})}{G} \frac{f(\boldsymbol{x})}{g(\boldsymbol{x})} \, \mathrm{d}\boldsymbol{x}) = G h(\frac{1}{G} \int_{\mathbb{R}^d} f(\boldsymbol{x}) \, \mathrm{d}\boldsymbol{x})$$
$$= G h(\frac{F}{G}) = F \log \frac{F}{G} \tag{32}$$
$$= \int_{\mathbb{R}^d} f(\boldsymbol{x}) \, \mathrm{d}\boldsymbol{x} \log \frac{\int_{\mathbb{R}^d} f(\boldsymbol{x}) \, \mathrm{d}\boldsymbol{x}}{\int_{\mathbb{R}^d} g(\boldsymbol{x}) \, \mathrm{d}\boldsymbol{x}}.$$

Note that the integrability of $f$ ensures the validity of Jensen's inequality. $\qquad\square$

**Lemma 2** *Let* $\sigma > 0$, $0 < \beta < 1$ *and* $B = \sigma \sqrt{(d+2) \log \frac{1}{\sigma^4} + \sigma^2 M}$, *for all* $\boldsymbol{v} \in \mathbb{R}^d$ *with* $|\boldsymbol{v}| \leq M$, *Let* $\delta = \min(e^{-\frac{1}{4}}, e^{\frac{1}{4} W_{-1}(-\frac{1}{d+2})})$ *and* $C = M(1 + \sqrt{2\pi})$*, then* $\sigma < \delta$ *indicates*

$$(2\pi\sigma^2)^{-\frac{d}{2}} \int_{|\boldsymbol{x}|>B} \exp(-\frac{||\boldsymbol{x} + \sigma^2 \boldsymbol{v}||^2}{2\sigma^2})(-(\boldsymbol{x} + \sigma^2 \boldsymbol{v})^T \boldsymbol{v}) \, \mathrm{d}\boldsymbol{x} \leq C\sigma^4 \tag{33}$$

*proof.*

$$(2\pi\sigma^2)^{-\frac{d}{2}} \int_{||\boldsymbol{x}||>B} \exp(-\frac{||\boldsymbol{x} + \sigma^2 \boldsymbol{v}||^2}{2\sigma^2})(-(\boldsymbol{x} + \sigma^2 \boldsymbol{v})^T \boldsymbol{v}) \, \mathrm{d}\boldsymbol{x}$$

$$\overset{(1)}{=} (2\pi\sigma^2)^{-\frac{d}{2}} \int_{||\boldsymbol{z} - \sigma^2 \boldsymbol{v}||>B} \exp(-\frac{||\boldsymbol{z}||^2}{2\sigma^2})(-\boldsymbol{z}^T \boldsymbol{v}) \, \mathrm{d}\boldsymbol{z}$$

$$\overset{(2)}{\leq} (2\pi\sigma^2)^{-\frac{d}{2}} \int_{||\boldsymbol{z}||>B-\sigma^2 M} \exp(-\frac{||\boldsymbol{z}||^2}{2\sigma^2})(M||\boldsymbol{z}||) \, \mathrm{d}\boldsymbol{z} \tag{34}$$

$$= (2\pi\sigma^2)^{-\frac{d}{2}} M \int_0^\pi \cdots \int_0^\pi \int_0^{2\pi} A(d) \sin^{d-2}(\varphi_1) \sin^{d-3}(\varphi_2) \cdots \sin(\varphi_{d-2}) \, \mathrm{d}\varphi_{d-1} \cdots \mathrm{d}\varphi_2 \, \mathrm{d}\varphi_1$$

$$\overset{(3)}{=} (2\pi)^{-\frac{d}{2}} M \frac{2\pi^{\frac{d}{2}}}{\Gamma(\frac{d}{2})} A(d),$$

where

$$A(d) = \sigma^{-d} \int_{r > B - \sigma^2 M} \exp(-\frac{r^2}{2\sigma^2}) M r^d \, \mathrm{d}r, \tag{35}$$

and the derivation of equation (1) is attributed to the change in the integral variable of $Z = x + \sigma^2 Y$. The inequality in equation (2) arises from a broader integral domain and a Cauchy inequality. Equation (3) is derived from the calculation of the $d - 1$ dimensional sphere $S^{d-1}$.

$$A(d) = \sigma \int_{r > B - \sigma^2 M} \exp(-\frac{r^2}{2\sigma^2}) M (\frac{r}{\sigma})^d \, \mathrm{d}\frac{r}{\sigma} \tag{36}$$

$$= \sigma M \int_{r' > \frac{B - \sigma^2 M}{\sigma}} \exp(-\frac{r'^2}{2}) r'^d \, \mathrm{d}r'$$

$$= \sigma M E(d).$$

Let $\delta = e^{-\frac{1}{4}}$ and then $\log \frac{1}{\sigma^4} > 1$.

$$E(d) = \int_{r > \frac{B - \sigma^2 M}{\sigma}} \exp(-\frac{r^2}{2}) r^d \, \mathrm{d}r$$

$$= -r^{d-1} \exp(-\frac{r^2}{2}) \Big|_{\frac{B - \sigma^2 M}{\sigma}}^{\infty} + \int_{r > \frac{B - \sigma^2 M}{\sigma}} \exp(-\frac{r^2}{2})(d-1) r^{d-2} \, \mathrm{d}r$$

$$= (\frac{B - \sigma^2 M}{\sigma})^{d-1} \exp(-\frac{(B - \sigma^2 M)^2}{2\sigma^2}) + (d-1) E(d-2)$$

$$= ((d+2)\log \frac{1}{\sigma^4})^{\frac{d-1}{2}} \exp(-\frac{(d+2)\log \frac{1}{\sigma^4}}{2}) + (d-1) E(d-2)$$

$$= ((d+2)\log \frac{1}{\sigma^4})^{\frac{d-1}{2}} (\sigma^4)^{\frac{d+2}{2}} + (d-1) E(d-2)$$

$$= ((d+2)\log \frac{1}{\sigma^4})^{\frac{d-1}{2}} (\sigma^4)^{\frac{d+2}{2}} + (d-1)((d+2)\log \frac{1}{\sigma^4})^{\frac{d-3}{2}} (\sigma^4)^{\frac{d+2}{2}} + \cdots \tag{37}$$

$$+ \begin{cases} (d-1)(d-3)\cdots 4((d+2)\log \frac{1}{\sigma^4})(\sigma^4)^{\frac{d+2}{2}} + 2E(1) & d \text{ is odd,} \\ (d-1)(d-3)\cdots 3((d+2)\log \frac{1}{\sigma^4})^{\frac{1}{2}}(\sigma^4)^{\frac{d+2}{2}} + E(0) & d \text{ is even,} \end{cases}$$

$$\le \frac{d+2}{2}((d+2)\log \frac{1}{\sigma^4})^{\frac{d-1}{2}} (\sigma^4)^{\frac{d+2}{2}} + \begin{cases} 2E(1) & d \text{ is odd,} \\ E(0) & d \text{ is even,} \end{cases}$$

$$\le ((d+2)\log \frac{1}{\sigma^4}\sigma^4)^{\frac{d+1}{2}} \sigma^4 + \begin{cases} 2E(1) & d \text{ is odd,} \\ E(0) & d \text{ is even.} \end{cases}$$

Let $\delta = e^{\frac{1}{4} W_{-1}(-\frac{1}{d+2})}$, we have $(d+2)\log \frac{1}{\sigma^4}\sigma^4 \le 1$, where $W_{-1}$ is the branch of Lambert W function labelled by -1.

$$E(1) = \int_{r > \frac{B - \sigma^2 M}{\sigma}} \exp(-\frac{r^2}{2}) r \, \mathrm{d}r$$

$$= -\int_{r > \frac{B - \sigma^2 M}{\sigma}} \mathrm{d}\exp(-\frac{r^2}{2})$$

$$= -\exp(-\frac{r^2}{2}) \Big|_{\frac{B - \sigma^2 M}{\sigma}}^{\infty} \tag{38}$$

$$= \exp(-\frac{(B - \sigma^2 M)^2}{2\sigma^2})$$
$$= (\sigma^4)^{\frac{d+2}{2}},$$

Since all $\lambda > 0$, $\exp(\lambda r - \lambda\sqrt{(d+2)\log\frac{1}{\sigma^4}}) \geq 1$, we have

$$
\begin{aligned}
E(0) &= \int_{r > \frac{B - \sigma^2 M}{\sigma}} \exp(-\frac{r^2}{2})\, \mathrm{d}r \\
&\leq \int_{\mathbb{R}} \exp(\lambda r - \lambda\sqrt{(d+2)\log\frac{1}{\sigma^4}})\exp(-\frac{r^2}{2})\, \mathrm{d}r \\
&= \sqrt{2\pi}\exp(\frac{\lambda^2}{2} - \lambda\sqrt{(d+2)\log\frac{1}{\sigma^4}}) \\
&\leq \sqrt{2\pi}\exp(-\frac{(d+2)\log\frac{1}{\sigma^4}}{2}) \\
&= \sqrt{2\pi}(\sigma^4)^{\frac{d+2}{2}}.
\end{aligned}
\tag{39}
$$

As a result, by setting $\delta = \min(e^{-\frac{1}{4}}, e^{\frac{1}{4}W_{-1}(-\frac{1}{d+2})})$ and $C = M(1 + \sqrt{2\pi})$, we can achieve the required inequality.

**Lemma 3** *Given the notations from Lemma 2, if $\delta = \min(e^{-\frac{1}{4(d+2)}}, e^{\frac{1}{4}W_{-1}(-\frac{1}{d+2})})$ we have*

$$\frac{B + \sigma^2 M}{\sigma}\exp(-\frac{|B + \sigma^2 M|^2}{2\sigma^2}) < \sigma^4. \tag{40}$$

*proof.* When $\delta = \min(e^{-\frac{1}{4(d+2)}}, e^{\frac{1}{4}W_{-1}(-\frac{1}{d+2})})$, we have $\frac{B - \sigma^2 M}{\sigma} \geq 1$ and $(d+2)\log\frac{1}{\sigma^4}\sigma^4 \leq 1$. Since the function $t\exp(-\frac{t^2}{2})$ is decreasing when $t \geq 1$, we have

$$
\begin{aligned}
\frac{B + \sigma^2 M}{\sigma}\exp(-\frac{|B + \sigma^2 M|^2}{2\sigma^2}) &\leq \frac{B - \sigma^2 M}{\sigma}\exp(-\frac{|B - \sigma^2 M|^2}{2\sigma^2}) \\
&= \sqrt{(d+2)\log\frac{1}{\sigma^4}}(\sigma^4)^{\frac{d+2}{2}} \\
&= \sqrt{(d+2)\log\frac{1}{\sigma^4}\sigma^4}(\sigma^4)^{\frac{d-1}{2}}\sigma^4 \\
&\leq \sigma^4.
\end{aligned}
\tag{41}
$$

**Lemma 4** *Given the notations from Lemma 2, consider a set of vectors $\{v_i \in \mathbb{R}^d \mid 1 \leq i \leq N\}$ such that $\max_i \|v_i\| \leq M$, along with corresponding weights $w_i$ for each vector $v_i$ satisfying $\sum_i w_i = 1$. For all $1 \leq i \leq N$, let $C = \frac{1}{2}\mathrm{Tr}(\sum_j w_j\|v_i\|^2) + M^3 + \frac{3}{2}M^4$ and $\delta = e^{\frac{1}{2\beta}W_{-1}(\frac{-\beta}{2(d+2)})}$ such that $\sigma < \delta$ indicates*

$$\underbrace{(2\pi\sigma^2)^{-\frac{d}{2}}\int_{\|x\| \leq B}\exp(-\frac{\|x + \sigma^2 y_i\|^2}{2\sigma^2})\sum_j w_j\frac{1}{2}v_j^T x x^T v_j\, \mathrm{d}x - \sum_j w_j\frac{1}{2}\sigma^2\|v_j\|^2}_{I} < C\sigma^{3-\beta}.$$

$$\tag{42}$$

*proof.*

Let $\delta = e^{\frac{1}{2\beta}W_{-1}(-\frac{\beta}{2(d+2)})}$, we have $\sigma^{2\beta}\log\frac{1}{\sigma^4} \leq \frac{1}{d+2}$, which means $\sigma((d+2)\log\frac{1}{\sigma^4})^{-\frac{1}{2}} < \sigma^{1-\beta}$

Since the matrix $\boldsymbol{V} \overset{def}{=} \sum_j w_j \boldsymbol{v}_j \boldsymbol{v}_j^T$ is symmetric, it can be diagonalized: $\boldsymbol{V} = \boldsymbol{U}\boldsymbol{\Lambda}\boldsymbol{U}^T$ where $\boldsymbol{\Lambda}$ is an diagonal matrix, $\boldsymbol{U}$ is an orthogonal matrix and

$$\text{Tr}(\boldsymbol{\Lambda}) = \text{Tr}(\sum_j w_j \boldsymbol{v}_j \boldsymbol{v}_j^T) = \sum_j w_j \text{Tr}(\boldsymbol{v}_j^T \boldsymbol{v}_j) = \sum_j w_j ||\boldsymbol{v}_i||^2. \tag{43}$$

With the change of variable $\boldsymbol{z} = \boldsymbol{U}^T(\boldsymbol{x} + \sigma^2 \boldsymbol{v}_i)$, we have

$$\text{I} = (2\pi\sigma^2)^{-\frac{d}{2}} \int_{||\boldsymbol{x}|| \leq B} \exp(-\frac{||\boldsymbol{x} + \sigma^2 \boldsymbol{y}_i||^2}{2\sigma^2}) \sum_j w_j \frac{1}{2} ||(\boldsymbol{x} + \sigma^2 \boldsymbol{v}_i - \sigma^2 \boldsymbol{v}_i)^T \boldsymbol{v}_j||^2 \, \mathrm{d}\boldsymbol{x}$$

$$= (2\pi\sigma^2)^{-\frac{d}{2}} \int_{||\boldsymbol{U}\boldsymbol{z} - \sigma^2 \boldsymbol{v}_i|| \leq B} \exp(-\frac{||\boldsymbol{z}||^2}{2\sigma^2}) \sum_j w_j \frac{1}{2} (||\boldsymbol{z}^T \boldsymbol{U}^T \boldsymbol{v}_j||^2 - 2\sigma^2 \boldsymbol{z}^T \boldsymbol{U}^T \boldsymbol{v}_j \boldsymbol{v}_i^T \boldsymbol{v}_j$$

$$+ \sigma^4 |\boldsymbol{v}_i^T \boldsymbol{v}_j|^2) \, \mathrm{d}\boldsymbol{z}$$

$$\leq (2\pi\sigma^2)^{-\frac{d}{2}} \int_{||\boldsymbol{z}|| \leq B + \sigma^2 M} \exp(-\frac{||\boldsymbol{z}||^2}{2\sigma^2}) \frac{1}{2} \sum_j w_j ||\boldsymbol{z}^T \boldsymbol{U}^T \boldsymbol{v}_j||^2 \, \mathrm{d}\boldsymbol{z} + M^3 \sigma^2 (B + \sigma^2 M) + \frac{1}{2} M^4 \sigma^4$$

$$= (2\pi\sigma^2)^{-\frac{d}{2}} \int_{||\boldsymbol{z}|| \leq B + \sigma^2 M} \exp(-\frac{||\boldsymbol{z}||^2}{2\sigma^2}) \frac{1}{2} \boldsymbol{z}^T \boldsymbol{\Lambda} \boldsymbol{z} \, \mathrm{d}\boldsymbol{z} + M^3 \sigma^{3-\beta} + \frac{3}{2} M^4 \sigma^4$$

$$\leq (2\pi\sigma^2)^{-\frac{d}{2}} \frac{1}{2} \sum_l \int_{|z_l| \leq B + \sigma^2 M} \exp(-\frac{||\boldsymbol{z}||^2}{2\sigma^2}) \boldsymbol{\Lambda}_{ll} z_l^2 \, \mathrm{d}z_1 \prod_{j=1, j \neq l}^{d} \int_{-\infty}^{\infty} \exp(-\frac{z_j^2}{2\sigma^2}) \, \mathrm{d}z_j \tag{44}$$

$$+ M^3 \sigma^{3-\beta} + \frac{3}{2} M^4 \sigma^4$$

$$= \text{Tr}(\boldsymbol{\Lambda})(2\pi\sigma^2)^{-\frac{1}{2}} \frac{1}{2} \int_{|z| < B + \sigma^2 M} \exp(-\frac{z^2}{2\sigma^2}) z^2 \, \mathrm{d}z + M^3 \sigma^{3-\beta} + \frac{3}{2} M^4 \sigma^4$$

$$= \text{Tr}(\boldsymbol{\Lambda}) \frac{\sigma^2}{\sqrt{2\pi}} \frac{1}{2} [2 \frac{B + \sigma^2 M}{\sigma} \exp(-\frac{|B + \sigma^2 M|^2}{2\sigma^2}) + \int_{|z| \leq B + \sigma^2 M} \exp(-\frac{z^2}{2\sigma^2}) \, \mathrm{d}\frac{z}{\sigma}]$$

$$+ M^3 \sigma^{3-\beta} + \frac{3}{2} M^4 \sigma^4$$

$$\leq \text{Tr}(\boldsymbol{\Lambda}) \sigma^2 \frac{1}{2} (\sigma^4 + 1) + M^3 \sigma^{3-\beta} + \frac{3}{2} M^4 \sigma^4$$

$$= \sum_j \frac{1}{2} \sigma^2 w_j |\boldsymbol{y}_j|^2 + (\text{Tr}(\boldsymbol{\Lambda}) \frac{1}{2} \sigma^{(3+\beta)} + M^3 + \frac{3}{2} M^4 \sigma^{1+\beta}) \sigma^{3-\beta}.$$

Let $C = \frac{1}{2} \text{Tr}(\boldsymbol{\Lambda}) + M^3 + \frac{3}{2} M^4$, we get the required equation (42). $\quad\square$

**Lemma 5** *When* $||\boldsymbol{x}|| \leq B$ *and* $\sum_i w_i \boldsymbol{v}_i = 0$, *let* $\delta = \min(e^{\frac{1}{2\beta'} W_{-1}(-\frac{\beta'}{2(d+2)})}, (1+M)^{\beta'-1})$ *and* $C = \frac{1}{2}(1+M) M^3 + \frac{1}{8} M^4 + \frac{e}{6}(1+M)^3 M^3 + \frac{1}{2}(1+M)^2 M^4 + \frac{3}{2}(1+M) M^4 + \frac{1}{6} M^6$. *If* $\sigma \leq \delta$, *we have*

$$\sum_j w_j \exp(-\boldsymbol{x}^T \boldsymbol{v}_j - \frac{1}{2} \sigma^2 ||\boldsymbol{v}_j||^2) - 1$$

$$\leq \sum_j w_j (\frac{1}{2} \sigma^2 (\boldsymbol{v}_j^T \boldsymbol{x} \boldsymbol{x}^T \boldsymbol{v}_j - ||\boldsymbol{v}_j||^2)) + C\sigma^{3-\beta} \tag{45}$$

*proof.*

Let $\delta = \min(e^{\frac{1}{2\beta'} W_{-1}(-\frac{\beta'}{2(d+2)})}, (1+M)^{\beta'-1})$ and then $B = \sigma\sqrt{(d+2)\log\frac{1}{\sigma^4}} + \sigma^2 M < (1 + \sigma^{1+\beta'} M)\sigma^{1-\beta'} < (1+M)\sigma^{1-\beta'}$. Thus $\boldsymbol{x} \leq B \leq (1+M)\sigma^{1-\beta'} \leq 1$ and $|\boldsymbol{v}_j| \leq M$, where $\beta' = \frac{\beta}{3}$.

Expanding the function $e^t$ at $t = 0$ with Lagrange's remainder, where $0 \leq \xi \leq B$

$$\sum_j w_j \exp(-\boldsymbol{x}^T \boldsymbol{v}_j - \frac{1}{2}\sigma^2 ||\boldsymbol{v}_j||^2) - 1$$

$$= \sum_j w_j \{-\boldsymbol{x}^T \boldsymbol{v}_j - \frac{1}{2}\sigma^2 ||\boldsymbol{v}_j||^2 + \frac{1}{2}(-\boldsymbol{x}^T \boldsymbol{v}_j - \frac{1}{2}\sigma^2 ||\boldsymbol{v}_j||^2)^2 + \frac{1}{6}e^\xi (-\boldsymbol{x}^T \boldsymbol{v}_j - \frac{1}{2}\sigma^2 ||\boldsymbol{v}_j||^2)^3\}$$

$$\leq \sum_j w_j \{[-\frac{1}{2}\sigma^2 ||\boldsymbol{v}_j||^2 + \frac{1}{2}\boldsymbol{v}_j^T \boldsymbol{x}\boldsymbol{x}^T \boldsymbol{v}_j] + \frac{1}{2}\sigma^2 \boldsymbol{x}^T \boldsymbol{v}_j ||\boldsymbol{v}_j||^2 + \frac{1}{8}\sigma^4 ||\boldsymbol{v}_j||^4$$

$$+ \frac{1}{6}e^B (||\boldsymbol{x}^T \boldsymbol{v}_j||^3 + 3\sigma^2 ||\boldsymbol{x}^T \boldsymbol{v}_j||^2 ||\boldsymbol{v}_j||^2 + 3\sigma^4 ||\boldsymbol{x}^T \boldsymbol{v}_j|| ||\boldsymbol{v}_j||^4 + \sigma^6 ||\boldsymbol{v}_j||^6)\} \tag{46}$$

$$= \sum_j w_j (\frac{1}{2}\sigma^2 (\boldsymbol{v}_j^T \boldsymbol{x}\boldsymbol{x}^T \boldsymbol{v}_j - ||\boldsymbol{v}_j||^2)) + \sum_j w_j (\frac{1}{2}(1 + M)M^3 \sigma^{3-\beta'} + \frac{1}{8}M^4 \sigma^4$$

$$+ \frac{1}{6}e(1 + M)^3 M^3 \sigma^{3-3\beta'} + \frac{1}{2}(1 + M)^2 M^4 \sigma^{4-2\beta'} + \frac{1}{2}3(1 + M)M^5 \sigma^{5-\beta'} + \frac{1}{6}M^6 \sigma^6)$$

$$\leq \sum_j w_j (\frac{1}{2}\sigma^2 (\boldsymbol{v}_j^T \boldsymbol{x}\boldsymbol{x}^T \boldsymbol{v}_j - ||\boldsymbol{v}_j||^2)) + C\sigma^{3-\beta},$$

where $C = \frac{1}{2}(1 + M)M^3 + \frac{1}{8}M^4 + \frac{e}{6}(1 + M)^3 M^3 + \frac{1}{2}(1 + M)^2 M^4 + \frac{3}{2}(1 + M)M^4 + \frac{1}{6}M^6$. $\square$

**Lemma 6** *For all $t_{min} \leq s < t \leq t_{max}$ and $0 < \beta < 1$, there exists a $\delta > 0$ and $C > 0$, depending on $\beta$, $t_{min}$ and $t_{max}$, such that if $t - s < \delta$, the inequality $KL(p(\boldsymbol{x}_s)||\tilde{p}(\boldsymbol{x}_s)) \leq KL(p(\boldsymbol{x}_t)||\tilde{p}(\boldsymbol{x}_t)) + C(t - s)^{\frac{3-\beta}{2}}$ holds.*

*proof.*

Noting that

$$KL(p(\boldsymbol{x}_s)||\tilde{p}(\boldsymbol{x}_s)) = \int_{\mathbb{R}^d} \int_{\mathbb{R}^d} p(\boldsymbol{x}_s|\boldsymbol{x}_t)p(\boldsymbol{x}_t)\,\mathrm{d}\boldsymbol{x}_t \log \frac{\int_{\mathbb{R}^d} p(\boldsymbol{x}_s|\boldsymbol{x}_t)p(\boldsymbol{x}_t)\,\mathrm{d}\boldsymbol{x}_t}{\int_{\mathbb{R}^d} \tilde{p}(\boldsymbol{x}_s|\boldsymbol{x}_t)\tilde{p}(\boldsymbol{x}_t)\,\mathrm{d}\boldsymbol{x}_t}\,\mathrm{d}\boldsymbol{x}_s$$

$$\leq \int_{\mathbb{R}^d} \int_{\mathbb{R}^d} p(\boldsymbol{x}_s|\boldsymbol{x}_t)p(\boldsymbol{x}_t) \log \frac{p(\boldsymbol{x}_s|\boldsymbol{x}_t)p(\boldsymbol{x}_t)}{\tilde{p}(\boldsymbol{x}_s|\boldsymbol{x}_t)\tilde{p}(\boldsymbol{x}_t)}\,\mathrm{d}\boldsymbol{x}_t\,\mathrm{d}\boldsymbol{x}_s$$

$$= \underbrace{\int_{\mathbb{R}^d} \int_{\mathbb{R}^d} p(\boldsymbol{x}_s|\boldsymbol{x}_t)p(\boldsymbol{x}_t) \log \frac{p(\boldsymbol{x}_s|\boldsymbol{x}_t)}{\tilde{p}(\boldsymbol{x}_s|\boldsymbol{x}_t)}\,\mathrm{d}\boldsymbol{x}_t\,\mathrm{d}\boldsymbol{x}_s}_{\text{I}} \tag{47}$$

$$+ \underbrace{\int_{\mathbb{R}^d} \int_{\mathbb{R}^d} p(\boldsymbol{x}_s|\boldsymbol{x}_t)p(\boldsymbol{x}_t) \log \frac{p(\boldsymbol{x}_t)}{\tilde{p}(\boldsymbol{x}_t)}\,\mathrm{d}\boldsymbol{x}_t\,\mathrm{d}\boldsymbol{x}_s}_{\text{II}}.$$

$$\text{II} = \int_{\mathbb{R}^d} \int_{\mathbb{R}^d} p(\boldsymbol{x}_s|\boldsymbol{x}_t)p(\boldsymbol{x}_t) \log p(\boldsymbol{x}_t)\,\mathrm{d}\boldsymbol{x}_t\,\mathrm{d}\boldsymbol{x}_s - \int_{\mathbb{R}^d} \int_{\mathbb{R}^d} p(\boldsymbol{x}_s|\boldsymbol{x}_t)p(\boldsymbol{x}_t) \log \tilde{p}(\boldsymbol{x}_t)\,\mathrm{d}\boldsymbol{x}_t\,\mathrm{d}\boldsymbol{x}_s \tag{48}$$

Since $p(\boldsymbol{x}_s|\boldsymbol{x}_t)p(\boldsymbol{x}_t) \log p(\boldsymbol{x}_t) \geq 0$ and $p(\boldsymbol{x}_s|\boldsymbol{x}_t)p(\boldsymbol{x}_t) \log \tilde{p}(\boldsymbol{x}_t) \leq 0$ for all $\boldsymbol{x}_t$ and $\boldsymbol{x}_s$, according to Fubini's theorem, we have

$$\int_{\mathbb{R}^d} \int_{\mathbb{R}^d} p(\boldsymbol{x}_s|\boldsymbol{x}_t)p(\boldsymbol{x}_t) \log p(\boldsymbol{x}_t)\,\mathrm{d}\boldsymbol{x}_t\,\mathrm{d}\boldsymbol{x}_s = \int_{\mathbb{R}^d} \int_{\mathbb{R}^d} p(\boldsymbol{x}_s|\boldsymbol{x}_t)p(\boldsymbol{x}_t) \log p(\boldsymbol{x}_t)\,\mathrm{d}\boldsymbol{x}_s\,\mathrm{d}\boldsymbol{x}_t$$

$$= \int_{\mathbb{R}^d} p(\boldsymbol{x}_t) \log p(\boldsymbol{x}_t)\,\mathrm{d}\boldsymbol{x}_t \tag{49}$$

and

$$\int_{\mathbb{R}^d} \int_{\mathbb{R}^d} p(\boldsymbol{x}_s|\boldsymbol{x}_t)p(\boldsymbol{x}_t) \log \tilde{p}(\boldsymbol{x}_t)\,\mathrm{d}\boldsymbol{x}_t\,\mathrm{d}\boldsymbol{x}_s = \int_{\mathbb{R}^d} \int_{\mathbb{R}^d} p(\boldsymbol{x}_s|\boldsymbol{x}_t)p(\boldsymbol{x}_t) \log p(\boldsymbol{x}_t)\,\mathrm{d}\boldsymbol{x}_s\,\mathrm{d}\boldsymbol{x}_t$$

$$= \int_{\mathbb{R}^d} p(\boldsymbol{x}_t) \log \tilde{p}(\boldsymbol{x}_t) \, \mathrm{d}\boldsymbol{x}_t. \tag{50}$$

Since the entropy of Gaussian mixtures and the cross entropy between Gaussians are all finite, we have $\int_{\mathbb{R}^d} \int_{\mathbb{R}^d} p(\boldsymbol{x}_s|\boldsymbol{x}_t) p(\boldsymbol{x}_t) \log p(\boldsymbol{x}_t) \, \mathrm{d}\boldsymbol{x}_t \, \mathrm{d}\boldsymbol{x}_s$ and $\int_{\mathbb{R}^d} \int_{\mathbb{R}^d} p(\boldsymbol{x}_s|\boldsymbol{x}_t) p(\boldsymbol{x}_t) \log \tilde{p}(\boldsymbol{x}_t) \, \mathrm{d}\boldsymbol{x}_t \, \mathrm{d}\boldsymbol{x}_s$ are both integrable. As a result,

$$\mathrm{II} = \int_{\mathbb{R}^d} \int_{\mathbb{R}^d} p(\boldsymbol{x}_s|\boldsymbol{x}_t) p(\boldsymbol{x}_t) \log p(\boldsymbol{x}_t) \, \mathrm{d}\boldsymbol{x}_t \, \mathrm{d}\boldsymbol{x}_s - \int_{\mathbb{R}^d} \int_{\mathbb{R}^d} p(\boldsymbol{x}_s|\boldsymbol{x}_t) p(\boldsymbol{x}_t) \log \tilde{p}(\boldsymbol{x}_t) \, \mathrm{d}\boldsymbol{x}_t \, \mathrm{d}\boldsymbol{x}_s$$

$$= \int_{\mathbb{R}^d} \int_{\mathbb{R}^d} p(\boldsymbol{x}_s|\boldsymbol{x}_t) p(\boldsymbol{x}_t) \log p(\boldsymbol{x}_t) \, \mathrm{d}\boldsymbol{x}_s \, \mathrm{d}\boldsymbol{x}_t - \int_{\mathbb{R}^d} \int_{\mathbb{R}^d} p(\boldsymbol{x}_s|\boldsymbol{x}_t) p(\boldsymbol{x}_t) \log \tilde{p}(\boldsymbol{x}_t) \, \mathrm{d}\boldsymbol{x}_s \, \mathrm{d}\boldsymbol{x}_t \tag{51}$$

$$= \int_{\mathbb{R}^d} \int_{\mathbb{R}^d} p(\boldsymbol{x}_s|\boldsymbol{x}_t) p(\boldsymbol{x}_t) \log \frac{p(\boldsymbol{x}_t)}{\tilde{p}(\boldsymbol{x}_t)} \, \mathrm{d}\boldsymbol{x}_s \, \mathrm{d}\boldsymbol{x}_t$$

$$= \mathrm{KL}(p(\boldsymbol{x}_t) \| \tilde{p}(\boldsymbol{x}_t)).$$

Now, let us delve into a detailed analysis of Part I.

$$\mathrm{I} = \int_{\mathbb{R}^d} \int_{\mathbb{R}^d} p(\boldsymbol{x}_s|\boldsymbol{x}_t) \log \frac{p(\boldsymbol{x}_s|\boldsymbol{x}_t)}{\tilde{p}(\boldsymbol{x}_s|\boldsymbol{x}_t)} \, \mathrm{d}\boldsymbol{x}_s p(\boldsymbol{x}_t) \, \mathrm{d}\boldsymbol{x}_t$$

$$= \int_{\mathbb{R}^d} [(2\pi\sigma_{s|t}^2)^{-\frac{d}{2}} \int_{\mathbb{R}^d} \sum_i w_i(\boldsymbol{x}_t, t) \exp(-\frac{\|\boldsymbol{x}_s - \frac{\alpha_{t|s}\sigma_s^2}{\sigma_t^2}\boldsymbol{x}_t - \frac{\alpha_s\sigma_{t|s}^2}{\sigma_t^2}\boldsymbol{y}_i\|^2}{2\sigma_{s|t}^2}) \log(\sum_j w_j(\boldsymbol{x}_t, t) \exp($$

$$- \underbrace{((\boldsymbol{x}_s - \frac{\alpha_{t|s}\sigma_s^2}{\sigma_t^2}\boldsymbol{x}_t - \frac{\alpha_s\sigma_{t|s}^2}{\sigma_t^2}\bar{\boldsymbol{y}}(\boldsymbol{x}_t, t))^T}_{\boldsymbol{x}} \underbrace{\frac{\alpha_s}{\sigma_s^2}(\bar{\boldsymbol{y}}(\boldsymbol{x}_t, t) - \boldsymbol{y}_j)}_{\boldsymbol{y}_j}$$

$$+ \frac{1}{2}\sigma_{s|t}^2 \|\frac{\alpha_s}{\sigma_s^2}(\bar{\boldsymbol{y}}(\boldsymbol{x}_t, t) - \boldsymbol{y}_j)\|^2)))] \, \mathrm{d}\boldsymbol{x}_s p(\boldsymbol{x}_t) \, \mathrm{d}\boldsymbol{x}_t$$

$$= \int_{\mathbb{R}^d} [(2\pi\sigma_{s|t}^2)^{-\frac{d}{2}} \underbrace{\int_{A(\boldsymbol{x}_t, B, \bar{\boldsymbol{y}}(\boldsymbol{x}_t, t))}}_{\mathrm{III}} + \underbrace{\int_{A^c(\boldsymbol{x}_t, B, \bar{\boldsymbol{y}}(\boldsymbol{x}_t, t))}}_{\mathrm{IV}} \sum_i w_i(\boldsymbol{x}_t, t) \exp($$

$$- \frac{\|\boldsymbol{x} + \sigma_{s|t}^2 \boldsymbol{y}_i\|^2}{2\sigma_{s|t}^2}) \log \sum_j w_j(\boldsymbol{x}_t, t) \exp(-\boldsymbol{x}^T \boldsymbol{y}_i - \frac{1}{2}\sigma_{s|t}^2 \|\boldsymbol{y}_i\|^2)] \, \mathrm{d}\boldsymbol{x}_s \, p(\boldsymbol{x}_t) \, \mathrm{d}\boldsymbol{x}_t. \tag{52}$$

According to Lemma 2, let $\delta_{\mathrm{III}} = \min(e^{-\frac{1}{4}}, e^{\frac{1}{4}W_{-1}(-\frac{1}{d+2})})$ and $C_{\mathrm{III}} = M_\sigma(1 + \sqrt{2\pi})$, where $M_\sigma = \frac{M}{\sigma_{t_{min}}^2}$, when $\sigma_{s|t} \leq \delta_{\mathrm{III}}$,

$$\mathrm{III} = \int_{\mathbb{R}^d} (2\pi\sigma_{s|t}^2)^{-\frac{d}{2}} \int_{A(\boldsymbol{x}_t, B, \bar{\boldsymbol{y}}(\boldsymbol{x}_t, t))} \sum_i w_i(\boldsymbol{x}_t, t) \exp(-\frac{\|\boldsymbol{x} + \sigma_{s|t}^2 \Delta\boldsymbol{y}_i\|^2}{2\sigma_{s|t}^2}) \log($$

$$\frac{\sum_j w_j(\boldsymbol{x}_t, t) \exp(-\frac{\|\boldsymbol{x} + \sigma_{s|t}^2 \Delta\boldsymbol{y}_j\|^2}{2\sigma_{s|t}^2})}{\sum_j w_j(\boldsymbol{x}_t, t) \exp(-\frac{\|\boldsymbol{x}\|^2}{2\sigma_{s|t}^2})}) \, \mathrm{d}\boldsymbol{x}_s \, p(\boldsymbol{x}_t) \, \mathrm{d}\boldsymbol{x}_t$$

$$\leq \int_{\mathbb{R}^d} (2\pi\sigma_{s|t}^2)^{-\frac{d}{2}} \int_{A(\boldsymbol{x}_t, B, \bar{\boldsymbol{y}}(\boldsymbol{x}_t, t))} \sum_i w_i(\boldsymbol{x}_t, t) \exp($$

$$- \frac{\|\boldsymbol{x} + \sigma_{s|t}^2 \Delta\boldsymbol{y}_i\|^2}{2\sigma_{s|t}^2}) \log(\frac{\exp(-\frac{\|\boldsymbol{x} + \sigma_{s|t}^2 \Delta\boldsymbol{y}_j\|^2}{2\sigma_{s|t}^2})}{\exp(-\frac{\|\boldsymbol{x}\|^2}{2\sigma_{s|t}^2})}) \, \mathrm{d}\boldsymbol{x}_s \, p(\boldsymbol{x}_t) \, \mathrm{d}\boldsymbol{x}_t \tag{53}$$

$$= \int_{\mathbb{R}^d} (2\pi\sigma_{s|t}^2)^{-\frac{d}{2}} \int_{A(\boldsymbol{x}_t, B, \bar{\boldsymbol{y}}(\boldsymbol{x}_t, t))} \sum_i w_i(\boldsymbol{x}_t, t) \exp($$

$$- \frac{\|\boldsymbol{x} + \sigma_{s|t}^2 \Delta\boldsymbol{y}_i\|^2}{2\sigma_{s|t}^2})(-\boldsymbol{x}^T \Delta\boldsymbol{y}_i - \frac{1}{2}\sigma_{s|t}^2 \|\boldsymbol{y}_i\|^2) \, \mathrm{d}\boldsymbol{x}_s \, p(\boldsymbol{x}_t) \, \mathrm{d}\boldsymbol{x}_t$$

$$\leq \int_{\mathbb{R}^d} C_{\mathrm{III}} \sigma_{s|t}^4 p(\boldsymbol{x}_t) \, \mathrm{d}\boldsymbol{x}_t = C_{\mathrm{III}} \sigma_{s|t}^4.$$

According to Lemma 4 and Lemma 5, let $C_{\mathrm{IV}} = \max(\frac{1}{2}M_\sigma + M_\sigma^3 + \frac{3}{2}M_\sigma^4, \frac{1}{2}(1+M_\sigma)M_\sigma^3 + \frac{1}{8}M_\sigma^4 + \frac{e}{6}(1+M_\sigma)^3 M_\sigma^3 + \frac{1}{2}(1+M_\sigma)^2 M_\sigma^4 + \frac{3}{2}(1+M_\sigma)M_\sigma^4 + \frac{1}{6}M_\sigma^6$ and $\delta_{\mathrm{IV}} = \min(e^{\frac{1}{3\beta}W_{-1}(-\frac{\beta}{6(d+2)})}, (1+M_\sigma)^{\frac{1}{3}\beta-1})$, where $M_\sigma = \frac{M}{\sigma_{t_{min}}^2}$, when $\sigma_{s|t} \leq \delta_{\mathrm{IV}}$

$$
\begin{aligned}
\mathrm{IV} &= \sum_i \int_{\mathbb{R}^d} w_i(\boldsymbol{x}_t, t)(2\pi\sigma_{s|t}^2)^{-\frac{d}{2}} \int_{|\boldsymbol{x}|<B} \exp(-\frac{||\boldsymbol{x}+\sigma_{s|t}^2\boldsymbol{y}_i||^2}{2\sigma_{s|t}^2}) \log[ \\
&\qquad \sum_j w_j(\boldsymbol{x}_t, t)\exp(-\boldsymbol{x}^T\boldsymbol{y}_j - \frac{1}{2}\sigma_{s|t}^2||\boldsymbol{y}_j||^2)] \, \mathrm{d}\boldsymbol{x} p(\boldsymbol{x}_t) \, \mathrm{d}\boldsymbol{x}_t \\
&= \sum_i \int_{\mathbb{R}^d} w_i(\boldsymbol{x}_t, t)(2\pi\sigma_{s|t}^2)^{-\frac{d}{2}} \int_{||\boldsymbol{x}||<B} \exp(-\frac{||\boldsymbol{x}+\sigma_{s|t}^2\boldsymbol{y}_i||^2}{2\sigma_{s|t}^2})[ \\
&\qquad \sum_j w_j(\boldsymbol{x}_t, t)\exp(-\boldsymbol{x}^T\boldsymbol{y}_j - \frac{1}{2}\sigma_{s|t}^2||\boldsymbol{y}_j||^2) - 1] \, \mathrm{d}\boldsymbol{x}_s p(\boldsymbol{x}_t) \, \mathrm{d}\boldsymbol{x}_t \\
&\leq \sum_i \int_{\mathbb{R}^d} w_i(\boldsymbol{x}_t, t)C_{\mathrm{IV}}\sigma_{s|t}^{3-\beta} p(\boldsymbol{x}_t) \, \mathrm{d}\boldsymbol{x}_t \\
&= C_{\mathrm{IV}}\sigma_{s|t}^{3-\beta}.
\end{aligned}
\tag{54}
$$

Because $\sigma_{s|t}^2 = (1 - \frac{\alpha_t^2}{\alpha_s^2})\frac{1-\alpha_s^2}{1-\alpha_t^2} \leq \frac{\alpha_s+\alpha_t}{\alpha_s^2}(\alpha_s - \alpha_t) \leq \frac{2C_\alpha}{\alpha_{t_{min}}^2}(t-s)$. Let $\delta = \min(\delta_{\mathrm{III}}, \delta_{\mathrm{III}})$ and $C = (C_{\mathrm{III}} + C_{\mathrm{IV}})(\frac{2C_\alpha}{\alpha_{t_{min}}^2})^{\frac{3}{2}}$, we get the result. $\qquad\square$

> **Proposition 1** *For all $t_{min} \leq s < t \leq t_{max}$ and $0 < \beta < 1$, there exist $\delta > 0$ and $C_1, C_2 > 0$ depending on $\beta$, $t_{min}$ and $t_{max}$, such that if $t - s < \delta$, the inequality $KL(p(\boldsymbol{x}_s)||p_\theta(\boldsymbol{x}_s)) \leq KL(p(\boldsymbol{x}_t)||p_\theta(\boldsymbol{x}_t)) + C_1(t-s)^{\frac{3-\beta}{2}} + C_2(t-s)\varepsilon_y$ holds.*

*proof.*

Since $|\bar{\boldsymbol{y}}(\boldsymbol{x}_t, t)| = |\sum_i w_i(\boldsymbol{x}_t, t)\boldsymbol{y}_i| \leq \sum_i w_i(\boldsymbol{x}_t, t)|\boldsymbol{y}_i| \leq \sum_i w_i(\boldsymbol{x}_t, t)M = M$, we have

$$\int_{\mathbb{R}^d} p(\boldsymbol{x}_t)|\bar{\boldsymbol{y}}(\boldsymbol{x}_t, t)|^2 < M^2. \tag{55}$$

$$
\begin{aligned}
\mathrm{KL}(p(\boldsymbol{x}_s)||p_\theta(\boldsymbol{x}_s)) &\leq \int_{\mathbb{R}^d}\int_{\mathbb{R}^d} p(\boldsymbol{x}_s|\boldsymbol{x}_t)p(\boldsymbol{x}_t)\log\frac{p(\boldsymbol{x}_s|\boldsymbol{x}_t)p(\boldsymbol{x}_t)}{p_\theta(\boldsymbol{x}_s|\boldsymbol{x}_t)p_\theta(\boldsymbol{x}_t)} \, \mathrm{d}\boldsymbol{x}_t \, \mathrm{d}\boldsymbol{x}_s \\
&= \int_{\mathbb{R}^d}\int_{\mathbb{R}^d} p(\boldsymbol{x}_s|\boldsymbol{x}_t)p(\boldsymbol{x}_t, t)\log\frac{p(\boldsymbol{x}_s|\boldsymbol{x}_t)}{p_\theta(\boldsymbol{x}_s|\boldsymbol{x}_t)} \, \mathrm{d}\boldsymbol{x}_t \, \mathrm{d}\boldsymbol{x}_s \\
&\quad + \int_{\mathbb{R}^d}\int_{\mathbb{R}^d} p(\boldsymbol{x}_s|\boldsymbol{x}_t)p(\boldsymbol{x}_t, t)\log\frac{p(\boldsymbol{x}_t, t)}{p_\theta(\boldsymbol{x}_t, t)} \, \mathrm{d}\boldsymbol{x}_t \, \mathrm{d}\boldsymbol{x}_s. \\
&= \int_{\mathbb{R}^d}\int_{\mathbb{R}^d} p(\boldsymbol{x}_s|\boldsymbol{x}_t)p(\boldsymbol{x}_t, t)[\log\frac{p(\boldsymbol{x}_s|\boldsymbol{x}_t)}{\tilde{p}(\boldsymbol{x}_s|\boldsymbol{x}_t)} + \log\frac{\tilde{p}(\boldsymbol{x}_s|\boldsymbol{x}_t)}{p_\theta(\boldsymbol{x}_s|\boldsymbol{x}_t)}] \, \mathrm{d}\boldsymbol{x}_t \, \mathrm{d}\boldsymbol{x}_s \quad (56) \\
&\quad + \mathrm{KL}(p(\boldsymbol{x}_t)||p_\theta(\boldsymbol{x}_t)) \\
&\overset{(1)}{\leq} \underbrace{\int_{\mathbb{R}^d}\int_{\mathbb{R}^d} p(\boldsymbol{x}_s|\boldsymbol{x}_t)p(\boldsymbol{x}_t, t)\log\frac{\tilde{p}(\boldsymbol{x}_s|\boldsymbol{x}_t)}{p_\theta(\boldsymbol{x}_s|\boldsymbol{x}_t)} \, \mathrm{d}\boldsymbol{x}_t \, \mathrm{d}\boldsymbol{x}_s}_{\mathrm{I}} \\
&\quad + C_1(t-s)^{\frac{3-\beta}{2}} + \mathrm{KL}(p(\boldsymbol{x}_t)||p_\theta(\boldsymbol{x}_t))
\end{aligned}
$$

The inequality (1) use the conclusion in Proposition 6.

Since $\tilde{p}(\boldsymbol{x}_s|\boldsymbol{x}_t)$ and $p_\theta(\boldsymbol{x}_s|\boldsymbol{x}_t)$ are Gaussians with the same covariance matrix,

$$
\mathrm{I} = \int_{\mathbb{R}^d} \int_{\mathbb{R}^d} p(\boldsymbol{x}_t, t) p(\boldsymbol{x}_s|\boldsymbol{x}_t) \frac{1}{2\sigma_{s|t}^2} [(2\boldsymbol{x}_s - 2\frac{\alpha_{t|s}\sigma_s^2}{\sigma_t^2}\boldsymbol{x}_t
$$
$$
- \frac{\alpha_s \sigma_{t|s}^2}{\sigma_t^2}(\bar{\boldsymbol{y}}(\boldsymbol{x}_t, t) + y_\theta(\boldsymbol{x}_t, t)))^T \frac{\alpha_s \sigma_{t|s}^2}{\sigma_t^2}(y_\theta(\boldsymbol{x}_t, t) - \bar{\boldsymbol{y}}(\boldsymbol{x}_t, t))] \, \mathrm{d}\boldsymbol{x}_s \, \mathrm{d}\boldsymbol{x}_t \tag{57}
$$
$$
= \underbrace{- \int_{\mathbb{R}^d} \int_{\mathbb{R}^d} p(\boldsymbol{x}_t, t) p(\boldsymbol{x}_s|\boldsymbol{x}_t) \frac{1}{2\sigma_{s|t}^2}(\frac{\alpha_s \sigma_{t|s}^2}{\sigma_t^2})^2 (y_\theta(\boldsymbol{x}_t, t) + \bar{\boldsymbol{y}}(\boldsymbol{x}_t, t))^T (y_\theta(\boldsymbol{x}_t, t) - \bar{\boldsymbol{y}}(\boldsymbol{x}_t, t)) \, \mathrm{d}\boldsymbol{x}_s \, \mathrm{d}\boldsymbol{x}_t}_{\mathrm{I}_1}
$$
$$
+ \underbrace{\int_{\mathbb{R}^d} \int_{\mathbb{R}^d} p(\boldsymbol{x}_t, t) p(\boldsymbol{x}_s|\boldsymbol{x}_t) \frac{1}{\sigma_{s|t}^2}(\boldsymbol{x}_s - \frac{\alpha_{t|s}\sigma_s^2}{\sigma_t^2}\boldsymbol{x}_t)^T \frac{\alpha_s \sigma_{t|s}^2}{\sigma_t^2}(y_\theta(\boldsymbol{x}_t, t) - \bar{\boldsymbol{y}}(\boldsymbol{x}_t, t)) \, \mathrm{d}\boldsymbol{x}_s \, \mathrm{d}\boldsymbol{x}_t}_{\mathrm{I}_2}
$$

$$
\mathrm{I}_1 = -\int_{\mathbb{R}^d} p(\boldsymbol{x}_t) \frac{\alpha_s^2 \sigma_{s|t}^2}{2\sigma_s^4}(y_\theta(\boldsymbol{x}_t, t) + \bar{\boldsymbol{y}}(\boldsymbol{x}_t, t))^T (y_\theta(\boldsymbol{x}_t, t) - \bar{\boldsymbol{y}}(\boldsymbol{x}_t, t)) \, \mathrm{d}\boldsymbol{x}_t
$$
$$
\leq \frac{\alpha_s^2 \sigma_{s|t}^2}{2\sigma_s^4}(\int_{\mathbb{R}^d} p(\boldsymbol{x}_t)||(y_\theta(\boldsymbol{x}_t, t) + \bar{\boldsymbol{y}}(\boldsymbol{x}_t, t))||^2 \, \mathrm{d}\boldsymbol{x}_t \int_{\mathbb{R}^d} p(\boldsymbol{x}_t)||(y_\theta(\boldsymbol{x}_t, t) - \bar{\boldsymbol{y}}(\boldsymbol{x}_t, t))||^2)^{\frac{1}{2}} \, \mathrm{d}\boldsymbol{x}_t
$$
$$
\leq \frac{\alpha_s^2 \sigma_{s|t}^2}{2\sigma_s^4}\varepsilon_y(\int_{\mathbb{R}^d} p(\boldsymbol{x}_t)||(y_\theta(\boldsymbol{x}_t, t) - \bar{\boldsymbol{y}}(\boldsymbol{x}_t, t) + 2\bar{\boldsymbol{y}}(\boldsymbol{x}_t, t)||^2 \, \mathrm{d}\boldsymbol{x}_t)^{\frac{1}{2}} \tag{58}
$$
$$
\leq \frac{\alpha_s^2 \sigma_{s|t}^2}{2\sigma_s^4}\varepsilon_y(\int_{\mathbb{R}^d} p(\boldsymbol{x}_t)3(||y_\theta(\boldsymbol{x}_t, t) - \bar{\boldsymbol{y}}(\boldsymbol{x}_t, t)||^2 + 4||\bar{\boldsymbol{y}}(\boldsymbol{x}_t, t)||^2) \, \mathrm{d}\boldsymbol{x}_t)^{\frac{1}{2}}
$$
$$
\leq 3\frac{\alpha_s^2 \sigma_{s|t}^2}{2\sigma_s^4}\varepsilon_y(\varepsilon_y^2 + 4M^2)^{\frac{1}{2}}
$$
$$
\leq \frac{3}{2\sigma_{t_{min}}^4}(1 + 4M^2)^{\frac{1}{2}}\sigma_{s|t}^2\varepsilon_y.
$$

$$
\mathrm{I}_2 = \frac{\alpha_s}{\sigma_s^2}\int_{\mathbb{R}^d} p(\boldsymbol{x}_t)\int_{\mathbb{R}^d} p(\boldsymbol{x}_s|\boldsymbol{x}_t)(\boldsymbol{x}_s - \frac{\alpha_{t|s}\sigma_s^2}{\sigma_t^2}\boldsymbol{x}_t)^T \, \mathrm{d}\boldsymbol{x}_s(y_\theta(\boldsymbol{x}_t, t) - \bar{\boldsymbol{y}}(\boldsymbol{x}_t, t)) \, \mathrm{d}\boldsymbol{x}_t
$$
$$
= \frac{\alpha_s}{\sigma_s^2}\int_{\mathbb{R}^d} p(\boldsymbol{x}_t)\sum_i w_i(\boldsymbol{x}_t, t)\frac{\alpha_s \sigma_{t|s}^2}{\sigma_t^2}\boldsymbol{y}_i^T(y_\theta(\boldsymbol{x}_t, t) - \bar{\boldsymbol{y}}(\boldsymbol{x}_t, t)) \, \mathrm{d}\boldsymbol{x}_t \tag{59}
$$
$$
\leq \frac{\alpha_s^2 \sigma_{s|t}^2}{\sigma_s^4}M(\int_{\mathbb{R}^d} p(\boldsymbol{x}_t)||y_\theta(\boldsymbol{x}_t, t) - \bar{\boldsymbol{y}}(\boldsymbol{x}_t, t)||^2 \, \mathrm{d}\boldsymbol{x}_t)^{\frac{1}{2}}
$$
$$
\leq \frac{\alpha_s^2}{\sigma_s^4}M\sigma_{s|t}^2\varepsilon_y \leq \frac{M}{\sigma_{t_{min}}^4}\sigma_{s|t}^2\varepsilon_y
$$

As as result, let $C_2 = (\frac{3}{2\sigma_{t_{min}}^4}(1 + 4M^2)^{\frac{1}{2}} + \frac{M}{\sigma_{t_{min}}^4})(\frac{2C_\alpha}{\alpha_{t_{min}}^2})^{\frac{3}{2}}$ and $\delta$ use the value in Lemma 6, we get the required result. $\qquad \square$

## A.2 PROOF OF PROPOSITION 2

**Proposition 2** *For all $0 < s < 1$, there are constants $C_1, C_2 > 0$, such that* $KL(p(\boldsymbol{x}_s)||p_\theta(\boldsymbol{x}_s)) \leq C_1(1 - s)^2 + C_2(1 - s)^2\varepsilon_y.$

*proof.*

First, we consider the difference between $p(\boldsymbol{x}_s)$ and $\tilde{p}(\boldsymbol{x}_s)$. Since $p(\boldsymbol{x}_1) = \tilde{p}(\boldsymbol{x}_1) \sim \mathcal{N}(\boldsymbol{0}, \boldsymbol{I})$,

$$
\begin{aligned}
&\mathrm{KL}(p(\boldsymbol{x}_s) || \tilde{p}(\boldsymbol{x}_s)) \\
&= \int_{\mathbb{R}^d} \int_{\mathbb{R}^d} p(\boldsymbol{x}_s|\boldsymbol{x}_1) p(\boldsymbol{x}_1) \, \mathrm{d}\boldsymbol{x}_1 \log \frac{\int_{\mathbb{R}^d} p(\boldsymbol{x}_s|\boldsymbol{x}_1) p(\boldsymbol{x}_1) \, \mathrm{d}\boldsymbol{x}_1}{\int_{\mathbb{R}^d} \tilde{p}(\boldsymbol{x}_s|\boldsymbol{x}_1) \tilde{p}(\boldsymbol{x}_1) \, \mathrm{d}\boldsymbol{x}_1} \, \mathrm{d}\boldsymbol{x}_s \\
&\leq \int_{\mathbb{R}^d} \int_{\mathbb{R}^d} p(\boldsymbol{x}_s|\boldsymbol{x}_1) p(\boldsymbol{x}_1) \log \frac{p(\boldsymbol{x}_s|\boldsymbol{x}_1) p(\boldsymbol{x}_1)}{\tilde{p}(\boldsymbol{x}_s|\boldsymbol{x}_1) \tilde{p}(\boldsymbol{x}_1)} \, \mathrm{d}\boldsymbol{x}_1 \, \mathrm{d}\boldsymbol{x}_s \\
&= \int_{\mathbb{R}^d} \int_{\mathbb{R}^d} p(\boldsymbol{x}_s|\boldsymbol{x}_1) p(\boldsymbol{x}_1) \log \frac{p(\boldsymbol{x}_s|\boldsymbol{x}_1)}{\tilde{p}(\boldsymbol{x}_s|\boldsymbol{x}_1)} \, \mathrm{d}\boldsymbol{x}_1 \, \mathrm{d}\boldsymbol{x}_s \\
&= \int_{\mathbb{R}^d} (2\pi\sigma_s^2)^{-\frac{d}{2}} \int_{\mathbb{R}^d} \sum_i w_i(\boldsymbol{x}_1, 1) \exp(-\frac{||\boldsymbol{x}_s - \alpha_s \boldsymbol{y}_i||^2}{2\sigma_s^2}) \log[ \\
&\qquad \frac{\sum_i w_i(\boldsymbol{x}_1, 1) \exp(-\frac{||\boldsymbol{x}_s - \alpha_s \boldsymbol{y}_i||^2}{2\sigma_s^2})}{\exp(-\frac{||\boldsymbol{x}_s - \alpha_s \bar{\boldsymbol{y}}(\boldsymbol{x}_1, 1)||^2}{2\sigma_s^2})}] \, \mathrm{d}\boldsymbol{x}_s p(\boldsymbol{x}_1) \, \mathrm{d}\boldsymbol{x}_1 \\
&\leq \int_{\mathbb{R}^d} (2\pi\sigma_s^2)^{-\frac{d}{2}} \int_{\mathbb{R}^d} \sum_i w_i(\boldsymbol{x}_1, 1) \exp(-\frac{||\boldsymbol{x}_s - \alpha_s \boldsymbol{y}_i||^2}{2\sigma_s^2}) \log[ \\
&\qquad \frac{\exp(-\frac{||\boldsymbol{x}_s - \alpha_s \boldsymbol{y}_i||^2}{2\sigma_s^2})}{\exp(-\frac{||\boldsymbol{x}_s - \alpha_s \bar{\boldsymbol{y}}(\boldsymbol{x}_1, 1)||^2}{2\sigma_s^2})}] \, \mathrm{d}\boldsymbol{x}_s p(\boldsymbol{x}_1) \, \mathrm{d}\boldsymbol{x}_1 \\
&\leq \int_{\mathbb{R}^d} (2\pi\sigma_s^2)^{-\frac{d}{2}} \int_{\mathbb{R}^d} \sum_i w_i(\boldsymbol{x}_1, 1) \exp(-\frac{||\boldsymbol{x}_s - \alpha_s \boldsymbol{y}_i||^2}{2\sigma_s^2}) \\
&\qquad \cdot \frac{\alpha_s}{\sigma_s^2}(\boldsymbol{x}_s - \alpha_s \boldsymbol{y}_i)^T (\boldsymbol{y}_i - \bar{\boldsymbol{y}}(\boldsymbol{x}_1, 1)) + \frac{\alpha_s^2}{2\sigma_s^2}||\boldsymbol{y}_i - \bar{\boldsymbol{y}}(\boldsymbol{x}_1, 1)||^2 \, \mathrm{d}\boldsymbol{x}_s p(\boldsymbol{x}_1) \, \mathrm{d}\boldsymbol{x}_1 \\
&\leq \int_{\mathbb{R}^d} \frac{\alpha_s^2}{2\sigma_s^2} M^2 p(\boldsymbol{x}_1) \, \mathrm{d}\boldsymbol{x}_1 \leq \frac{M^2}{2\sigma_{t_{min}}^2} \alpha_s^2 \leq \frac{C_\alpha^2 M^2}{2\sigma_{t_{min}}^2}(1-s)^2.
\end{aligned}
\tag{60}
$$

Upon considering equations (56), (58), and (59), and designating $C_2 = C_\alpha^2 (\frac{3}{2\sigma_{t_{min}}^4}(1 + 4M^2)^{\frac{1}{2}} + \frac{M}{\sigma_{t_{min}}^4})$, we are able to derive the desired conclusion. $\qquad\square$

The method used to prove the previous Proposition cannot be applied to prove Proposition 1 because there is a $\sigma_{s|t}^2$ in the denominator. This results in an error bound of $\sigma_{s|t}^2$, which does not allow for global convergence.

### A.3 PROOF OF PROPOSITION 3

**Proposition 3** *Given $0 < t_{min} < 1$, the 2-Wasserstein distance*

$$
W_2(p(\boldsymbol{x}_0), p(\boldsymbol{x}_{t_{min}})) < \sqrt{2dC_\alpha t_{min}}.
\tag{61}
$$

*proof.*

$$
\begin{aligned}
W_2(p(\boldsymbol{x}_0), p(\boldsymbol{x}_{t_{min}})) &\leq \frac{1}{N} \sum_{i=1}^N W_2(\delta(x - \boldsymbol{y}_i), \mathcal{N}(\boldsymbol{y}_i, \sigma_{t_{min}}^2 \boldsymbol{I})) \\
&= \frac{1}{N} \sum_{i=1}^N \sqrt{d}\sigma_{t_{min}}
\end{aligned}
\tag{62}
$$

$$\leq \frac{1}{N} \sum_{i=1}^{N} \sqrt{2dC_\alpha t_{min}} = \sqrt{2dC_\alpha t_{min}}.$$

$\square$

### A.4 Proof of Proposition 4

**Proposition 4** *For all $0 < \beta < 1$, there exist $\delta > 0$ and $C_1, C_2, C_3 > 0$, such that for all time discretizations $\mathcal{D}$ with $|\mathcal{D}| < \delta$, the Kullback-Leibler divergence $KL(p(\boldsymbol{x}_{t_{min}})||p_\theta(\boldsymbol{x}_{t_{min}})) \leq C_1|\mathcal{D}|^{\frac{1-\beta}{2}} + C_2\varepsilon_y$. Moreover, $W_2^2(p(\boldsymbol{x}_0), p(\boldsymbol{x}_{t_{min}})) < C_3|\mathcal{D}|$.*

*proof.*

According to Proposition 1 and 2, for all $i \in \{1, 2, \ldots, T\}$

$$\mathrm{KL}(p(x_{t_i})||p_\theta(x_{t_i})) \leq \mathrm{KL}(p(x_{t_{i+1}})||p_\theta(x_{t_{i+1}})) + C_1(t_{i+1} - t_i)^{\frac{3-\beta}{2}} + C_2(t_{i+1} - t_i)\varepsilon_y$$

$$\leq \mathrm{KL}(p(\boldsymbol{x}_1)||p_\theta(\boldsymbol{x}_1)) + C_1|\mathcal{D}|^{\frac{1-\beta}{2}} + C_2\varepsilon_y. \tag{63}$$

The final estimation using the 2-Wasserstein is simply the Proposition 3. $\square$

### A.5 Proof of Corollary 1

**Corollary 1** *For all $t_{min} \leq s < t \leq t_{max}$ and $0 < \beta < 1$, there exists a $\delta > 0$ and $C_1, C_2, C_3 > 0$, depending on $\beta$, $t_{min}$ and $t_{max}$, such that if $t - s < \delta$, the inequality $KL(p(\boldsymbol{x}_s)||\hat{p}_\theta(\boldsymbol{x}_s)) \leq KL(p(\boldsymbol{x}_t)||\hat{p}_\theta(\boldsymbol{x}_t)) + C_1(t-s)^{\frac{3-\beta}{2}} + C_2(t-s)\varepsilon_{yl} + C_3\varepsilon_{al}$ holds.*

*proof.* In accordance with the convexity of the Kullback-Leibler divergence, we have

$$KL(p(\boldsymbol{x}_s|\boldsymbol{x}_t)||\hat{p}(\boldsymbol{x}_s|\boldsymbol{x}_t)) \leq \sum_l a^l(\boldsymbol{x}_t, t) KL(p^l(\boldsymbol{x}_s|\boldsymbol{x}_t), \hat{p}^l(\boldsymbol{x}_s|\boldsymbol{x}_t))$$

$$\overset{(1)}{=} \sum_l a^l(\boldsymbol{x}_t, t) C_1(t-s)^{\frac{3-\beta}{2}} \tag{64}$$

$$= C_1(t-s)^{\frac{3-\beta}{2}}.$$

The equality in step (1) is a direct consequence of Proposition 1.

$$\mathrm{KL}(p(\boldsymbol{x}_s)||\hat{p}_\theta(\boldsymbol{x}_s)) \leq \int_{\mathbb{R}^d} \int_{\mathbb{R}^d} p(\boldsymbol{x}_s|\boldsymbol{x}_t)p(\boldsymbol{x}_t) \log \frac{p(\boldsymbol{x}_s|\boldsymbol{x}_t)p(\boldsymbol{x}_t)}{\hat{p}_\theta(\boldsymbol{x}_s|\boldsymbol{x}_t)\hat{p}_\theta(\boldsymbol{x}_t)} \, \mathrm{d}\boldsymbol{x}_t \, \mathrm{d}\boldsymbol{x}_s$$

$$= \int_{\mathbb{R}^d} \int_{\mathbb{R}^d} p(\boldsymbol{x}_s|\boldsymbol{x}_t)p(\boldsymbol{x}_t, t) \log \frac{p(\boldsymbol{x}_s|\boldsymbol{x}_t)}{\hat{p}_\theta(\boldsymbol{x}_s|\boldsymbol{x}_t)} \, \mathrm{d}\boldsymbol{x}_t \, \mathrm{d}\boldsymbol{x}_s$$

$$+ \int_{\mathbb{R}^d} \int_{\mathbb{R}^d} p(\boldsymbol{x}_s|\boldsymbol{x}_t)p(\boldsymbol{x}_t, t) \log \frac{p(\boldsymbol{x}_t, t)}{\hat{p}_\theta(\boldsymbol{x}_t, t)} \, \mathrm{d}\boldsymbol{x}_t \, \mathrm{d}\boldsymbol{x}_s.$$

$$= \int_{\mathbb{R}^d} \int_{\mathbb{R}^d} p(\boldsymbol{x}_s|\boldsymbol{x}_t)p(\boldsymbol{x}_t, t)[\log \frac{p(\boldsymbol{x}_s|\boldsymbol{x}_t)}{\hat{p}(\boldsymbol{x}_s|\boldsymbol{x}_t)} + \log \frac{\hat{p}(\boldsymbol{x}_s|\boldsymbol{x}_t)}{\hat{p}_\theta(\boldsymbol{x}_s|\boldsymbol{x}_t)}] \, \mathrm{d}\boldsymbol{x}_t \, \mathrm{d}\boldsymbol{x}_s \tag{65}$$

$$+ \mathrm{KL}(p(\boldsymbol{x}_t)||\hat{p}_\theta(\boldsymbol{x}_t))$$

$$\overset{(1)}{\leq} \underbrace{\int_{\mathbb{R}^d} \int_{\mathbb{R}^d} p(\boldsymbol{x}_s|\boldsymbol{x}_t)p(\boldsymbol{x}_t, t) \log \frac{\hat{p}(\boldsymbol{x}_s|\boldsymbol{x}_t)}{\hat{p}_\theta(\boldsymbol{x}_s|\boldsymbol{x}_t)} \, \mathrm{d}\boldsymbol{x}_t \, \mathrm{d}\boldsymbol{x}_s}_{\mathrm{I}}$$

$$+ C_1(t-s)^{\frac{3-\beta}{2}} + \mathrm{KL}(p(\boldsymbol{x}_t)||\hat{p}_\theta(\boldsymbol{x}_t))$$

The inequality (1) results from equation (64).

$$I = \int_{\mathbb{R}^d} \int_{\mathbb{R}^d} p(\boldsymbol{x}_s|\boldsymbol{x}_t)p(\boldsymbol{x}_t,t) \log \frac{\sum_{l=1}^L a^l(\boldsymbol{x}_t,t)\hat{p}^l(\boldsymbol{x}_s|\boldsymbol{x}_t)}{\sum_{l=1}^L a_\phi^l(\boldsymbol{x}_t,t)\hat{p}_\theta^l(\boldsymbol{x}_s|\boldsymbol{x}_t)} \, d\boldsymbol{x}_t \, d\boldsymbol{x}_s$$

$$\leq \int_{\mathbb{R}^d} \int_{\mathbb{R}^d} p(\boldsymbol{x}_s|\boldsymbol{x}_t)p(\boldsymbol{x}_t,t) \frac{1}{\sum_{k=1}^L a^k(\boldsymbol{x}_t,t)\hat{p}^k(\boldsymbol{x}_s|\boldsymbol{x}_t)}$$

$$\cdot \sum_{l=1}^L a^l(\boldsymbol{x}_t,t)\hat{p}^l(\boldsymbol{x}_s|\boldsymbol{x}_t) \log \frac{a^l(\boldsymbol{x}_t,t)\hat{p}^l(\boldsymbol{x}_s|\boldsymbol{x}_t)}{a_\phi^l(\boldsymbol{x}_t,t)\hat{p}_\theta^l(\boldsymbol{x}_s|\boldsymbol{x}_t)} \, d\boldsymbol{x}_t \, d\boldsymbol{x}_s$$

$$\leq \int_{\mathbb{R}^d} \int_{\mathbb{R}^d} p(\boldsymbol{x}_s|\boldsymbol{x}_t)p(\boldsymbol{x}_t,t) \sum_l \log \frac{\hat{p}^l(\boldsymbol{x}_s|\boldsymbol{x}_t)}{\hat{p}_\theta^l(\boldsymbol{x}_s|\boldsymbol{x}_t)} \, d\boldsymbol{x}_t \, d\boldsymbol{x}_s$$

$$+ \int_{\mathbb{R}^d} \int_{\mathbb{R}^d} p(\boldsymbol{x}_s|\boldsymbol{x}_t)p(\boldsymbol{x}_t,t) \sum_l \log \frac{a^l(\boldsymbol{x}_t,t)}{a_\phi^l(\boldsymbol{x}_t,t)} \, d\boldsymbol{x}_t \, d\boldsymbol{x}_s \tag{66}$$

$$\overset{(1)}{\leq} C_2 \sigma_{s|t}^2 \varepsilon_{yl} + \int_{\mathbb{R}^d} p(\boldsymbol{x}_t,t) \sum_l \log \frac{a^l(\boldsymbol{x}_t,t)}{a_\phi^l(\boldsymbol{x}_t,t)} \, d\boldsymbol{x}_t$$

$$\leq C_2 \sigma_{s|t}^2 \varepsilon_{yl} + \int_{\mathbb{R}^d} p(\boldsymbol{x}_t,t) \sum_l \frac{1}{C_a}|a^l(\boldsymbol{x}_t,t) - a_\phi^l(\boldsymbol{x}_t,t)| \, d\boldsymbol{x}_t$$

$$\leq C_2 \sigma_{s|t}^2 \varepsilon_{yl} + (\int_{\mathbb{R}^d} p(\boldsymbol{x}_t,t) \sum_l \frac{1}{C_a}|a^l(\boldsymbol{x}_t,t) - a_\phi^l(\boldsymbol{x}_t,t)|^2 \, d\boldsymbol{x}_t)^{\frac{1}{2}}$$

$$= C_2 \sigma_{s|t}^2 \varepsilon_{yl} + C_3 \varepsilon_{al}.$$

Given the established relationship between $\sigma_{s|t}$ and $t - s$, we are able to derive the necessary conclusion.

## A.6 PROOF OF COROLLARY 2

**Corollary 2** *For all* $0 < s < 1$, *there are constants* $C_1, C_2, C_3 > 0$, *such that* $KL(p(\boldsymbol{x}_s)||p_\theta(\boldsymbol{x}_s)) \leq C_1(1-s)^2 + C_2\varepsilon_{yl} + C_3\varepsilon_{al}$.

*proof.*

$$KL(p(\boldsymbol{x}_s)||\hat{p}^\theta(\boldsymbol{x}_s))$$

$$= \int_{\mathbb{R}^d} \int_{\mathbb{R}^d} p(\boldsymbol{x}_s|\boldsymbol{x}_1)p(\boldsymbol{x}_1) \, d\boldsymbol{x}_1 \log \frac{\int_{\mathbb{R}^d} p(\boldsymbol{x}_s|\boldsymbol{x}_1)p(\boldsymbol{x}_1) \, d\boldsymbol{x}_1}{\int_{\mathbb{R}^d} \hat{p}^\theta(\boldsymbol{x}_s|\boldsymbol{x}_1)\hat{p}^\theta(\boldsymbol{x}_1) \, d\boldsymbol{x}_1} \, d\boldsymbol{x}_s$$

$$\leq \int_{\mathbb{R}^d} \int_{\mathbb{R}^d} p(\boldsymbol{x}_s|\boldsymbol{x}_1)p(\boldsymbol{x}_1) \log \frac{p(\boldsymbol{x}_s|\boldsymbol{x}_1)p(\boldsymbol{x}_1)}{\hat{p}^\theta(\boldsymbol{x}_s|\boldsymbol{x}_1)\hat{p}^\theta(\boldsymbol{x}_1)} \, d\boldsymbol{x}_1 \, d\boldsymbol{x}_s$$

$$= \int_{\mathbb{R}^d} \int_{\mathbb{R}^d} p(\boldsymbol{x}_s|\boldsymbol{x}_1)p(\boldsymbol{x}_1) \log \frac{p(\boldsymbol{x}_s|\boldsymbol{x}_1)}{\hat{p}^\theta(\boldsymbol{x}_s|\boldsymbol{x}_1)} \, d\boldsymbol{x}_1 \, d\boldsymbol{x}_s$$

$$= \int_{\mathbb{R}^d} \int_{\mathbb{R}^d} p(\boldsymbol{x}_s|\boldsymbol{x}_1)p(\boldsymbol{x}_1) \log \frac{p(\boldsymbol{x}_s|\boldsymbol{x}_1)}{\hat{p}(\boldsymbol{x}_s|\boldsymbol{x}_1)} \, d\boldsymbol{x}_1 \, d\boldsymbol{x}_s$$

$$+ \int_{\mathbb{R}^d} \int_{\mathbb{R}^d} p(\boldsymbol{x}_s|\boldsymbol{x}_1)p(\boldsymbol{x}_1) \log \frac{\hat{p}(\boldsymbol{x}_s|\boldsymbol{x}_1)}{\hat{p}^\theta(\boldsymbol{x}_s|\boldsymbol{x}_1)} \, d\boldsymbol{x}_1 \, d\boldsymbol{x}_s \tag{67}$$

$$\leq \int_{\mathbb{R}^d} \int_{\mathbb{R}^d} \sum_l a^l(\boldsymbol{x}_t,t)p^l(\boldsymbol{x}_s|\boldsymbol{x}_1)p(\boldsymbol{x}_1) \log \frac{p^l(\boldsymbol{x}_s|\boldsymbol{x}_1)}{\hat{p}^l(\boldsymbol{x}_s|\boldsymbol{x}_1)} \, d\boldsymbol{x}_1 \, d\boldsymbol{x}_s$$

$$+ \int_{\mathbb{R}^d} \int_{\mathbb{R}^d} p(\boldsymbol{x}_s|\boldsymbol{x}_1)p(\boldsymbol{x}_1) \frac{1}{\sum_k a^k(\boldsymbol{x}_t,1)\hat{p}^l(\boldsymbol{x}_s|\boldsymbol{x}_1)}$$

$$\dot{\sum}_l a^l(\boldsymbol{x}_s|\boldsymbol{x}_1)\hat{p}^l(\boldsymbol{x}_s|\boldsymbol{x}_1) \log \frac{a^l(\boldsymbol{x}_1,1)\hat{p}^l(\boldsymbol{x}_s|\boldsymbol{x}_1)}{a_\phi^l(\boldsymbol{x}_1,1)\hat{p}^\theta(\boldsymbol{x}_s|\boldsymbol{x}_1)} \, d\boldsymbol{x}_1 \, d\boldsymbol{x}_s$$

$$\overset{(1)}{\leq} C_1(1-s)^2 + C_2\varepsilon_{yl} + C_3\varepsilon_{al}$$

The inequality (1) utilizes Proposition 2 in conjunction with the method employed in the proof of Corollary 1. □

### A.7 PROOF OF COROLLARY 3

**Corollary 3** *For all $0 < \beta < 1$, there is a $\delta > 0$ and $C_1, C_2, C_3, C_4 > 0$, such that for all time discretizations $\mathcal{D}$ with $|\mathcal{D}| < \delta$, the Kullback-Leibler divergence $KL(p(\boldsymbol{x}_{t_{min}})||\hat{p}_\theta(\boldsymbol{x}_{t_{min}})) \leq C_1|\mathcal{D}|^{\frac{1-\beta}{2}} + C_2\varepsilon_{yl} + C_3 T\varepsilon_{al}$. Moreover, $W_2(p(\boldsymbol{x}_0), p(\boldsymbol{x}_{t_{min}})) < C_4|\mathcal{D}|$.*

*proof.*

The methodology employed to prove this corollary mirrors that used in the proof of Proposition 4. The sole divergence lies in the inclusion of an additional term with $\varepsilon_{al}$. □

### A.8 PROOF OF PROPOSITION 5

**Proposition 5** *Let $L(\boldsymbol{x}_0)$ denote the one-hot class vector of $\boldsymbol{x}_0$, the optimal $\boldsymbol{a}_\phi(\boldsymbol{x}_t, t)$ for the two objective functions*

$$\mathcal{L}_2 = \mathbb{E}_{\boldsymbol{x}_0 \sim p_{data}, \boldsymbol{x}_t \sim p(\boldsymbol{x}_t|\boldsymbol{x}_0), t \sim \mathbf{U}(0,1)} ||L(\boldsymbol{x}_0) - \boldsymbol{a}_\phi(\boldsymbol{x}_t, t)||^2, \tag{68}$$

*and*

$$\mathcal{L}_{CE} = \mathbb{E}_{\boldsymbol{x}_0 \sim p_{data}, \boldsymbol{x}_t \sim p(\boldsymbol{x}_t|\boldsymbol{x}_0), t \sim \mathbf{U}(0,1)} CE(L(\boldsymbol{x}_0), \boldsymbol{a}_\phi(\boldsymbol{x}_t, t)), \tag{69}$$

*are the same and equal to $\boldsymbol{a}(\boldsymbol{x}_t, t)$, where CE represents the cross-entropy loss.*

*proof.*

Given that the subscript is utilized for data indices, we opt to use superscripts for vector components within the context of this proof. Let $L_i$ denotes the one-hot class vector of the data $\boldsymbol{y}_i$.

(1) Loss $\mathcal{L}_2$. It is a constrained optimization problem:

$$\begin{cases} \underset{\boldsymbol{a}_\phi}{\arg\min} \quad \mathcal{L}_2, \\ s.t. \ \mathbb{1}^T\boldsymbol{a}_\phi = 1, \ a_\theta^l \geq 0, \end{cases} \tag{70}$$

where $\mathbb{1}$ is a column vector, all of whose elements are 1s. Using the KKT condition Nocedal & Wright (1999)

$$0 = \nabla_{\boldsymbol{a}_\phi(\boldsymbol{x}_t,t)}\mathcal{L}_2 + \nu(\mathbb{1}^T\boldsymbol{a}_\phi(\boldsymbol{x}_t,t) - 1) - \mu^T\boldsymbol{a}_\phi(\boldsymbol{x}_t,t)$$

$$= \nabla_{\boldsymbol{a}_\phi(\boldsymbol{x}_t,t)} \sum_i \underbrace{\frac{1}{N}(2\pi\sigma_t^2)^{-\frac{d}{2}}}_{A_t} v_i(\boldsymbol{x}_t,t)|L_i - \boldsymbol{a}_\phi(\boldsymbol{x}_t,t)|^2 + \nu(\mathbb{1}^T\boldsymbol{a}_\phi(\boldsymbol{x}_t,t) - 1) - \mu^T\boldsymbol{a}_\phi(\boldsymbol{x}_t,t)$$

$$= \sum_i A_t v_i(\boldsymbol{x}_t,t)(\boldsymbol{a}_\phi(\boldsymbol{x}_t,t) - L_i) + \nu\mathbb{1} - \mu$$

$$= A_t \sum_i v_i(\boldsymbol{x}_t,t)\boldsymbol{a}_\phi(\boldsymbol{x}_t,t) - A_t \sum_i v_i(\boldsymbol{x}_t,t)L_i + \nu\mathbb{1} - \mu, \tag{71}$$

which leads to

$$\boldsymbol{a}_\phi^*(\boldsymbol{x}_t,t) = \frac{\sum_i v_i(\boldsymbol{x}_t,t)L_i - \nu\mathbb{1}/A_t + \mu/A_t}{\sum_j v_j(\boldsymbol{x}_t,t)}, \tag{72}$$

where $\mu^l \geq 0, \forall c$. Because $\mathbb{1}^T\boldsymbol{a}_\phi^*(\boldsymbol{x}_t,t) = 1$, we have

$$\mathbb{1}^T\boldsymbol{a}_\phi^*(\boldsymbol{x}_t,t) = \frac{\sum_i v_i(\boldsymbol{x}_t,t)\mathbb{1}L_i - \nu L/A_t + \mathbb{1}^T\mu/A_t}{\sum_j v_j(\boldsymbol{x}_t,t)} = 1 - \nu L/A_t + \mathbb{1}^T\mu/A_t, \tag{73}$$

which indicates $\nu L = \mathbb{1}^T \mu \geq 0$. Since $(\boldsymbol{a}_\phi^*(\boldsymbol{x}_t, t))^l \mu^l = 0, \forall l$, we have

$$(\frac{\sum_i v_i(\boldsymbol{x}_t, t) L_i - \mathbb{1}\nu/A_t + \mu/A_t}{\sum_j v_j(\boldsymbol{x}_t, t)})^l \mu^l = 0. \tag{74}$$

If

$$\sum_i v_i(\boldsymbol{x}_t, t) L_i^l - \nu/A_t + \mu^l/A_t = 0, \tag{75}$$

then

$$\nu > \mu^l \geq 0, \tag{76}$$

which lead to the contradiction

$$\nu L > \mathbb{1}^T \mu. \tag{77}$$

As as result, $\mu = 0$ and $\nu = 0$, and

$$\boldsymbol{a}_\phi^*(\boldsymbol{x}_t, t) = \frac{\sum_i v_i(\boldsymbol{x}_t, t) L_i - \mathbb{1}\nu/A_t + \mu/A_t}{\sum_j v_j(\boldsymbol{x}_t, t)} = \sum_i w_i(\boldsymbol{x}_t, t) L_i = \boldsymbol{a}(\boldsymbol{x}_t, t). \tag{78}$$

The Lagrange multiplier $\nu$ and $\mu$ are zero, which means we can omit the constrain $\mathbb{1}^T \boldsymbol{a}_\phi(\boldsymbol{x}_t, t) = 1$ and $a_\phi^l \geq 0$. As the object function and the feasible set are all convex, the KKT condition is alson sufficient.

(2) The loss $\mathcal{L}_{CE}$. Using the KKT condition Nocedal & Wright (1999)

$$0 = \nabla_{\boldsymbol{a}_\phi(\boldsymbol{x}_t, t)} \mathcal{L}_{CE} + \nu(\mathbb{1}^T \boldsymbol{a}_\phi(\boldsymbol{x}_t, t) - 1) - \mu^T \boldsymbol{a}_\phi(\boldsymbol{x}_t, t)$$

$$= \nabla_{\boldsymbol{a}_\phi(\boldsymbol{x}_t, t)} \sum_i \underbrace{\frac{1}{N}(2\pi\sigma_t^2)^{-\frac{d}{2}}}_{A_t} v_i(\boldsymbol{x}_t, t) \sum_l -L_i^l \log(a_\theta^l(\boldsymbol{x}_t, t)) + \nu(\mathbb{1}^T \boldsymbol{a}_\phi(\boldsymbol{x}_t, t) - 1) - \mu$$

$$= -\sum_i A_t v_i(\boldsymbol{x}_t, t) \begin{bmatrix} L_i^1/a_\theta^1(\boldsymbol{x}_t, t) \\ \vdots \\ L_i^L/a_\theta^L(\boldsymbol{x}_t, t) \end{bmatrix} + \nu\mathbb{1} - \mu, \tag{79}$$

which leads to

$$(\boldsymbol{a}_\phi^*(\boldsymbol{x}_t, t))^l = \sum_i \frac{A_t v_i(\boldsymbol{x}_t, t)}{\nu - \mu^l} L_i^l, \tag{80}$$

where $\mu^l \geq 0$. Since $(\boldsymbol{a}_\phi^*(\boldsymbol{x}_t, t))^l \mu^l = 0$, we must have $\mu^l = 0, \forall l$.

Because $\mathbb{1}^T \boldsymbol{a}_\phi^*(\boldsymbol{x}_t, t) = 1$, we have $\nu = A_t \sum_i v_i(\boldsymbol{x}_t, t)$. Thus

$$\boldsymbol{a}_\phi^*(\boldsymbol{x}_t, t) = \sum_i \frac{A_t v_i(\boldsymbol{x}_t, t)}{A_t \sum_j v_j(\boldsymbol{x}_t, t)} L_i = \sum_i w_i(\boldsymbol{x}_t, t) L_i = \boldsymbol{a}(\boldsymbol{x}_t, t). \tag{81}$$

In this case, the Lagrange multiplier $\nu$ is not zero, thus the constrain $\mathbb{1}^T \boldsymbol{a}_\phi(\boldsymbol{x}_t, t) = 1$ is essential. As the object function and the feasible set are all convex, the KKT condition is also sufficient. $\square$

### A.9 EXTENSION TO THE GENERAL UNDERLYING DISTRIBUTION

Our theory is developed based on the assumption that the initial distribution $p_0$ is in a Dirac sum form as shown in Assumption 1. This is exactly what happens in diffusion training, that is, we train the diffusion models based on the dataset, which can only be a Dirac sum form. However, some works Oko et al. (2023) assume the existence of a more general underlying initial distribution $p_0^{\text{general}}$, and the dataset distribution $p_0$ is an i.i.d. $N$-sample of the $p_0^{\text{general}}$. In this section, we will establish results that our $p_\theta$ trained on the dataset can also converge to $p_0^{\text{general}}$ as $N \to \infty$. We first give a finite momentum assumption on the underlying initial distribution $p_0^{\text{general}}$.

**Assumption 5** *The underlying initial distribution $p_0^{general}$ has a compact support, that is:*

$$p_0^{general} \in \mathcal{P}_c(\mathbb{R}^d) \tag{82}$$

*Note that this is a rather mild assumption; it does not specify the continuity or differentiability of $p_0^{general}$. This assumption of compactness is commonly met in image, text, and video distributions.*

Building upon Assumption 5, we are now in a position to establish the convergence of $p_\theta$ and $p_0^{general}$. However, before this, it is crucial to revisit the relationship of the Wasserstein distance between an arbitrary distribution and its corresponding $N$-sample distribution.

**Theorem 6** *(Fournier & Guillin, 2015, Theorem 1) Let $\mu \in \mathcal{P}\left(\mathbb{R}^d\right)$ and let $p > 0$. Assume that $M_q(\mu) < \infty$ for some $q > p$. There exists a constant $C$ depending only on $p, d, q$ such that, for all $N \geq 1$,*

$$\mathbb{E}\left(\mathcal{W}_p\left(\mu_N, \mu\right)\right) \leq C M_q^{p/q}(\mu) \begin{cases} N^{-1/2} + N^{-(q-p)/q} & \text{if } p > d/2 \text{ and } q \neq 2p, \\ N^{-1/2}\log(1+N) + N^{-(q-p)/q} & \text{if } p = d/2 \text{ and } q \neq 2p, \\ N^{-p/d} + N^{-(q-p)/q} & \text{if } p \in (0, d/2) \text{ and } q \neq d/(d-p) \end{cases} \tag{83}$$

*where $M_q(\mu)$ is the q-order momentum of $\mu$, $\mu_N$ is a $N$-sample empirical measure of $\mu$.*

Now, we are ready to establish the convergence property of learned distribution towards the general underlying data distribution.

**Proposition 7** *For all $0 < \beta < 1$, there exist $\delta > 0$, $N_1 > 0$ and $C_1, C_2, C_4 > 0$, such that for all time discretizations $\mathcal{D}$ with $|\mathcal{D}| < \delta$. Let $p_0^{N_1}$ be a $N_1$-sample of the general underlying data distribution $p_0^{general}$, and $p_\theta$ be trained by the $N_1$-sample $p_0^{N_1}$. Then, the Kullback-Leibler divergence $KL(p(\boldsymbol{x}_{t_{min}})||p_\theta(\boldsymbol{x}_{t_{min}})) \leq C_1|\mathcal{D}|^{\frac{1-\beta}{2}} + C_2\varepsilon_y$. Moreover, $W_2^2(p_0^{general}(\boldsymbol{x}_0), p(\boldsymbol{x}_{t_{min}})) < C_4|\mathcal{D}|$.*

*proof.*

Based on the compactness property stipulated in Assumption 5, the $q$ value of $p_0^{general}(\boldsymbol{x}_0)$ in Theorem 6 can be any arbitrary integer. According to Theorem 6, the gap will approach 0 as $N \to \infty$ in either case. Therefore, for any $\delta_2 > 0$, there exists a $N_2$ such that for all $N > N_2$, an $N$-sample empirical distribution $p_0^N$ of $p_0^{general}$ will satisfy the following condition:

$$W_2(p_0^{general}(\boldsymbol{x}_0), p^N(\boldsymbol{x}_{t_{min}})) < \delta_2. \tag{84}$$

For the purpose of this proof, we set $\delta_2 = \sqrt{C_3\mathcal{D}}$, and $N_1 = 2 * N_2(\delta_2)$. In this case, the following holds:

$$W_2(p_0^{general}(\boldsymbol{x}_0), p_0^{N_1}(\boldsymbol{x}_0)) < \sqrt{C_3\mathcal{D}}. \tag{85}$$

By applying Proposition 4 to $p_0^{N_1}$, we obtain:

$$\begin{aligned} &W_2^2(p_0^{general}(\boldsymbol{x}_0), p(\boldsymbol{x}_{t_{min}})) \\ \leq &W_2^2(p_0^{general}(\boldsymbol{x}_0), p_0^{N_1}(\boldsymbol{x}_0)) + W_2^2(p_0^{N_1}(\boldsymbol{x}_0), p(\boldsymbol{x}_{t_{min}})) \\ < &(\sqrt{C_3\mathcal{D}})^2 + C_3\mathcal{D} \\ = &2C_3\mathcal{D}. \end{aligned} \tag{86}$$

The proof can be concluded by setting $C_4 = 2C_3$. The values of $C_1$ and $C_2$ are derived from the application of Proposition 4. □

## B EXPERIMENTAL RESULTS ON IMAGENET

We further evaluate our proposed method on the challenging ImageNet64x64 dataset (Deng et al., 2009), as shown in Table 4. Our DC-DPM applied to the raw DDPM not only outperforms DDPM but also achieves lower FID scores than the higher-order SN-DDPM and GMS, which use enhanced reverse kernels for diffusion models. This demonstrates the effectiveness of our approach.

Table 4: **FID ↓ on ImageNet64x64 Dataset**. Employing our Divide-and-Conquer (DC) kernel approximation strategy on DDPM enables better generation quality than previous methods.

| # TIMESTEPS | 25 | 50 | 100 | 200 | 400 |
|---|---|---|---|---|---|
| DDPM (Ho et al., 2020) | 29.21 | 21.71 | 19.12 | 17.81 | 17.48 |
| SN-DDPM (Bao et al., 2022b) | 27.58 | 20.74 | 18.04 | 16.72 | 16.37 |
| GMS (Guo et al., 2024) | 26.50 | 20.13 | 17.29 | 16.60 | 15.98 |
| **DDPM+DC-DPM** (Ours) | **24.60** | **18.91** | **16.46** | **14.93** | **14.00** |

## C COMBINATION WITH PREVIOUS DIFFUSION ACCELERATION METHODS

As our DC-DPM improves the representation of diffusion model reverse transition kernels, it is orthogonal to previous acceleration methods including faster SDE/ODE solvers like DPM Solver (Lu et al., 2022a) and DPM Solver++ (Lu et al., 2022b). Therefore, our DC-DPM can be combined with these methods and we provide evaluation on CIFAR10 dataset to showcase the further improvement brought by our methods in Table 5

Table 5: **FID ↓ of Combining Our Method with Faster ODE solvers on CIFAR10 Dataset**. Our Divide-and-Conquer (DC) kernel approximation strategy can be applied to previous diffusion acceleration methods to enable better generation quality.

| # TIMESTEPS | 10 | 25 | 30 | 50 |
|---|---|---|---|---|
| DPM Solver (Lu et al., 2022a) | 7.95 | 6.54 | 6.17 | 3.37 |
| **+DC-DPM** (Ours) | **5.78** | **3.49** | **3.28** | **2.90** |
| DPM Solver++ (Lu et al., 2022b) | 11.11 | 7.41 | 6.76 | 3.42 |
| **+DC-DPM** (Ours) | **10.94** | **4.29** | **3.80** | **2.99** |

## D COMBINATION WITH EDM

Our DC-DPM can also be seamlessly combined with advanced diffusion methods such as EDM (Karras et al., 2022). As shown in Table 6, applying DC-DPM can improve the generation quality of EDM, showcasing the powerful effectiveness of our proposed method.

## E ABLATION STUDY

We conduct experiments to examine the impact of different classification approaches and varying numbers of classes on the CIFAR-10 dataset. We flatten the input images and apply K-Means clustering to the raw image values, classifying the training data into different numbers of clusters. The

Table 6: **FID ↓ of Combining Our Method with EDM(Karras et al., 2022) on CIFAR10 Dataset**. Our DC-DPM can be applied to advanced diffusion methods like EDM for further improvement.

| # TIMESTEPS | 10 | 25 | 30 | 50 |
|---|---|---|---|---|
| EDM (Karras et al., 2022) | 49.30 | 26.65 | 24.32 | 19.57 |
| **+DC-DPM** (Ours) | **36.86** | **21.14** | **19.21** | **14.72** |

FIDs for various denoising timesteps are presented in Fig. (3a). Our results indicate that the quality of generated images improves as the number of classes increases, although the rate of improvement diminishes with a higher number of classes. Notably, the generation quality for clusters created via K-Means remains inferior to that achieved using semantic labels, even when divided into 16 clusters.

In Fig. (3b), we compare scenarios with different numbers of semantic labels. The label S3 denotes three semantic classes. Specifically, we consolidated the original 10 classes of the CIFAR-10 dataset into three broader categories: vehicles, animals, and others. For S40, we divide each of the original 10 classes into four finer sub-classes using K-Means, resulting in a total of 40 classes. The generation quality initially improves and then deteriorates as the number of classes increases. While having more classes makes each cluster-specific kernel easier to learn, it simultaneously raises the complexity of managing all these classes within a single conditional diffusion network $y_\theta(x_t, t, l)$.

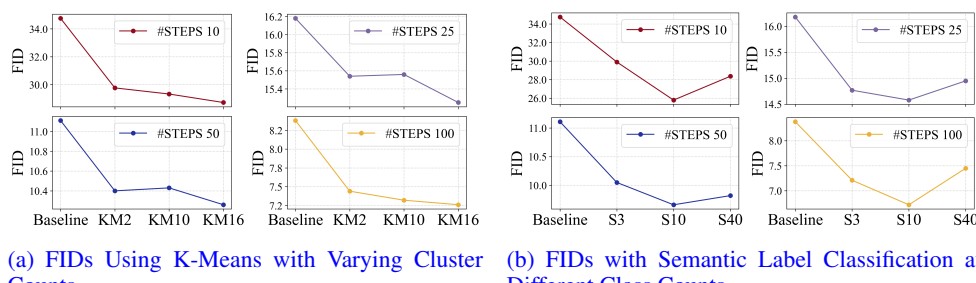

(a) FIDs Using K-Means with Varying Cluster Counts.

(b) FIDs with Semantic Label Classification at Different Class Counts.

Figure 3: **Ablations on Classification Approaches and Number of Classes on CIFAR-10.**

## F EXPERIMENTAL DETAILS

### F.1 TRAINING DETAILS

We use aligned training setting to that of the noise prediction network in Extended-Analytic-DPM (Bao et al., 2022a) and GMS (Guo et al., 2024) for image-space experiments on CIFAR-10. We use an exponential moving average (EMA) with a rate of 0.9999 and set the batch size as 128, learning rate as 2e-4. We train 600K iterations and save a checkpoint every 10K iterations. For the latent-space experiments on CelebA-HQ-256, we align the setting with LDM (Rombach et al., 2022) and set the batch size as 48, learning rate as 9.6e-5. We train 500K iterations and choose the best checkpoint to evaluate. We use the same training setting for the label model in label diffusion merging approximation. Training on CIFAR-10 and CelebA-HQ-256 both take about 48 hours on 8 Tesla V100 GPUs.

To apply our method to Extended-Analytic-DPM and GMS, two higher-order noise prediction networks need to be trained. We align the settings with Extended-Analytic-DPM and GMS and train two additional light-weight prediction heads with the backbone model frozen. Please refer to these two original papers for more details.

### F.2 EVALUATION DETAILS

Following Extended-Analytic-DPM and GMS, we calculate the FID score on 50K generated samples, using the official implementation of FID for pytorch (https: //github.com/mseitzer/pytorch-fid). The reference distribution statistics of FID are computed on the full training set. The parameters in

| #STEPS | Patterns (Number of Classes) | | | | | | | | | | | | | | |
|---|---|---|---|---|---|---|---|---|---|---|---|---|---|---|---|
| | Circles (2) | | | Moons (2) | | | Pinwheel (5) | | | CheckerBoard (8) | | | Gaussians (8) | | |
| | 1 GS | **FC** | **LD** | 1 GS | **FC** | **LD** | 1 GS | **FC** | **LD** | 1 GS | **FC** | **LD** | 1 GS | **FC** | **LD** |
| 500 | 2.788 | 8.017 | **0.8717** | 3.935 | 7.051 | **2.590** | 3.672 | 6.658 | **3.056** | -1.301 | 6.561 | **-2.912** | 6.339 | 7.945 | **2.567** |
| 100 | 7.685 | 5.066 | **2.088** | 5.724 | 4.111 | **0.7585** | 3.891 | 9.002 | **3.789** | 0.1639 | 7.913 | **-4.804** | 6.364 | 11.80 | **5.768** |
| 50 | 12.42 | 4.764 | **2.594** | 13.21 | 5.457 | **0.08196** | 14.05 | 14.79 | **3.265** | 9.516 | 8.759 | **0.8225** | 18.93 | 13.66 | **7.536** |
| 30 | 20.82 | **10.86** | 15.55 | 21.38 | 8.021 | **3.508** | 24.37 | 17.64 | **8.619** | 33.57 | 12.18 | **7.559** | 45.70 | 8.005 | **1.572** |
| 20 | 50.96 | **18.90** | 40.57 | 40.36 | 10.44 | **10.12** | 52.51 | 20.48 | **9.817** | 115.5 | 19.88 | **10.22** | 90.04 | 6.098 | **3.801** |
| 10 | 121.0 | **33.44** | 71.94 | 149.1 | **43.29** | 44.30 | 169.3 | 21.07 | **12.32** | 494.9 | **54.94** | 90.71 | 211.1 | 22.03 | **16.62** |

Table 7: **Comparison on Synthetic Datasets.** 1 GS indicates the baseline which approximates each step as a single Gaussian. FC and LD represent our methods. Generation quality is assessed by Maximum Mean Discrepancy (MMD) $\downarrow$. Values in the table have been rescaled by a factor of $10^{-5}$.

sampling are kept aligned with those in Extended-Analytic-DPM, please refer to Appendix F.5 in the original paper of Extended-Analytic-DPM (Bao et al., 2022a) for more details.

### F.3   RESULTS ON 2D SYNTHETIC DATASET

We validate our approach on five synthetic 2D datasets with varying distributions. Each dataset consists of continuous 2D points $(x, y) \in \mathbb{R}^2$, assigned class labels based on natural clustering. For each experiment, we generated 4K samples and assessed generation quality using Maximum Mean Discrepancy (MMD) with a Laplace kernel (bandwidth 0.1) (Gretton et al., 2012). Each computation was repeated 8 times, and we report the average MMD value, with lower values indicating better generation quality. As shown in Table 7, our LD and FC methods outperform the single Gaussian baseline, achieving lower MMD across different timesteps.

## G   QUALITATIVE RESULTS

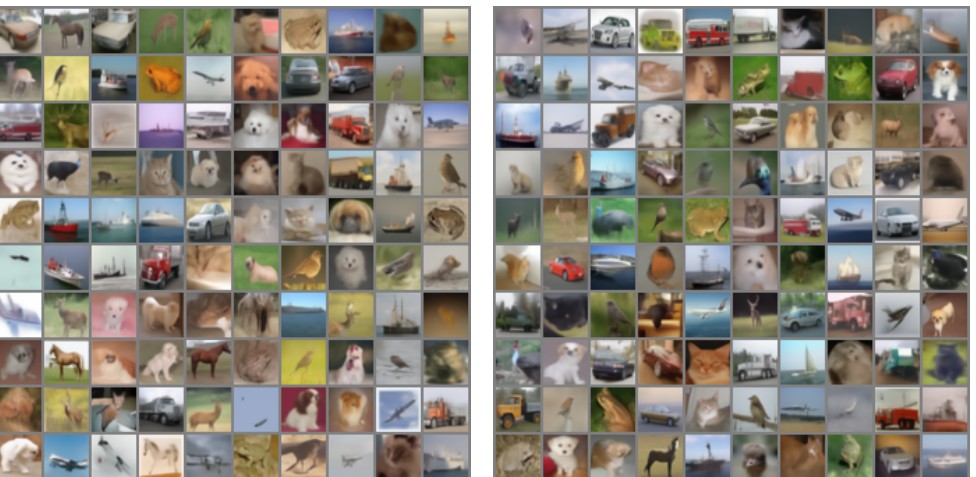

Figure 4: **DDPM + DC-DPM on 10 Denoising Steps.**

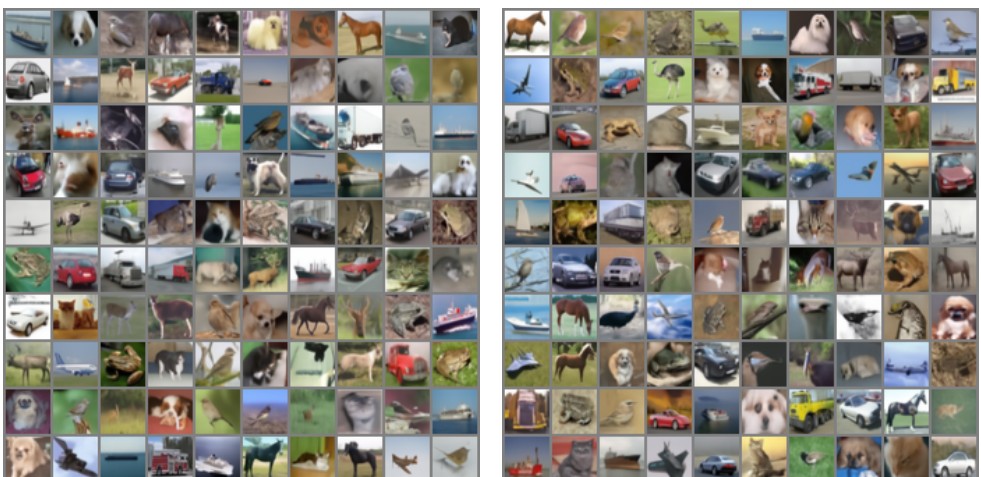

Figure 5: **DDPM + DC-DPM on 25 Denoising Steps.**

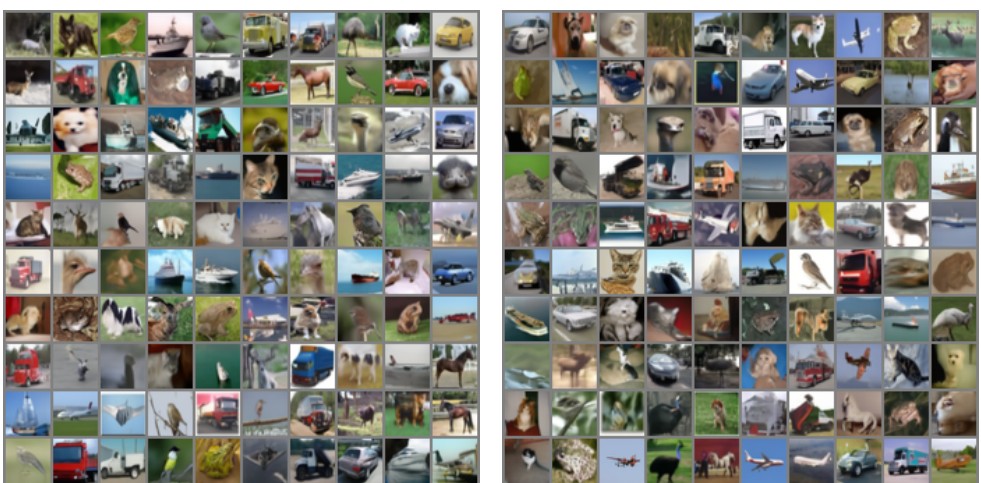

Figure 6: **DDPM + DC-DPM on 50 Denoising Steps.**

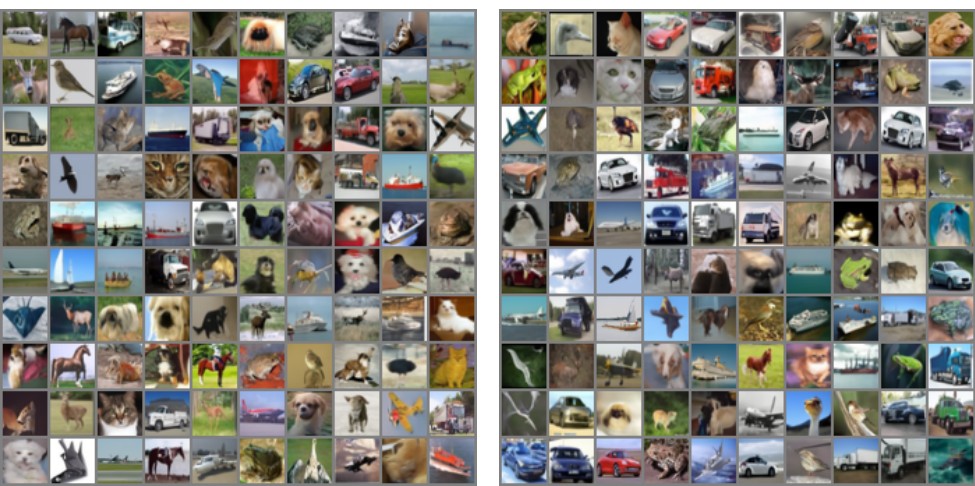

Figure 7: **DDPM + DC-DPM on 100 Denoising Steps.**

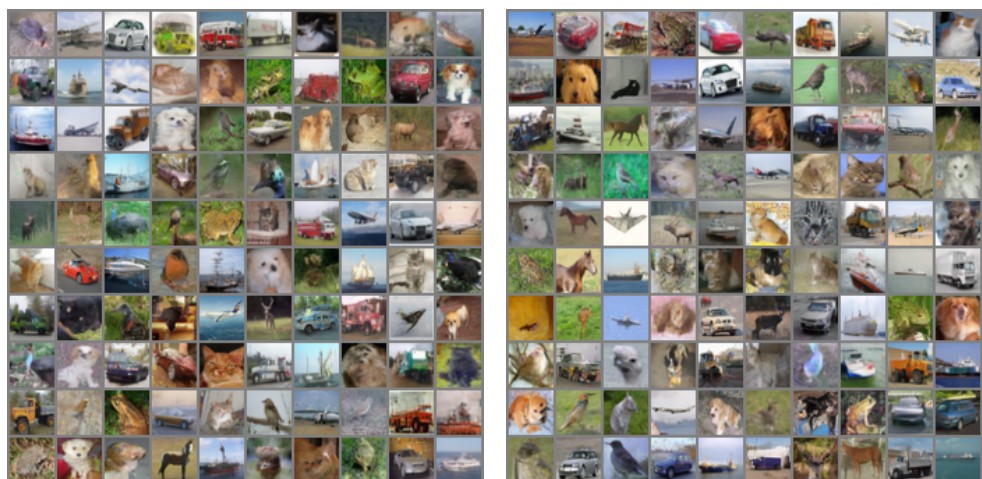

Figure 8: **SN-DDPM (Bao et al., 2022a) + DC-DPM on 10 Denoising Steps.**

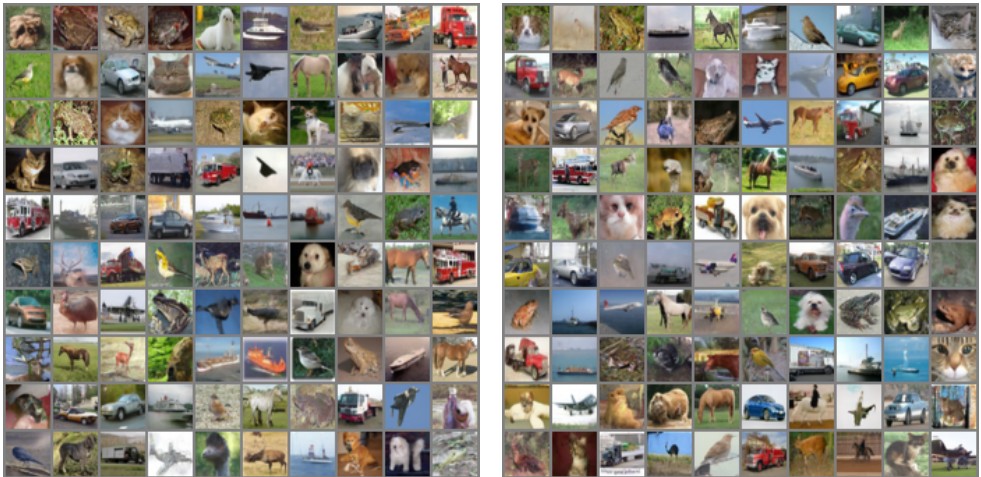

Figure 9: **SN-DDPM + DC-DPM on 25 Denoising Steps.**

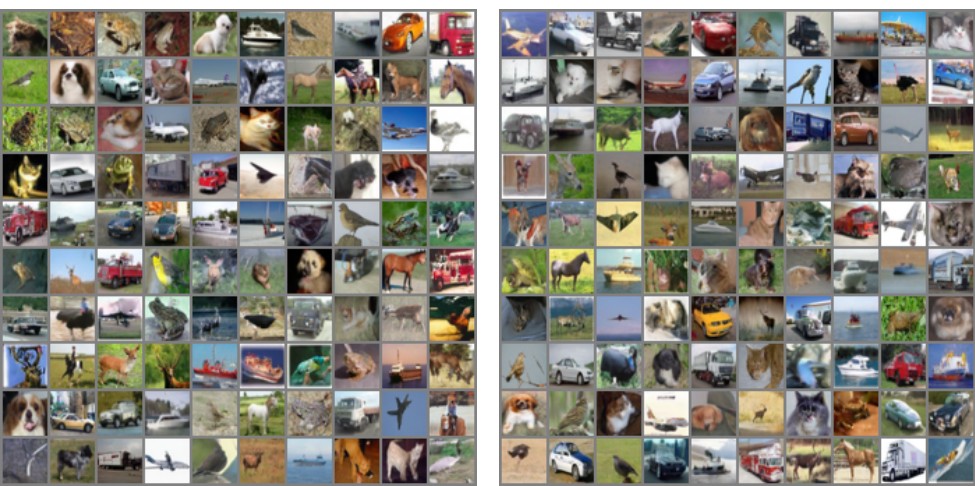

Figure 10: **SN-DDPM + DC-DPM on 50 Denoising Steps.**

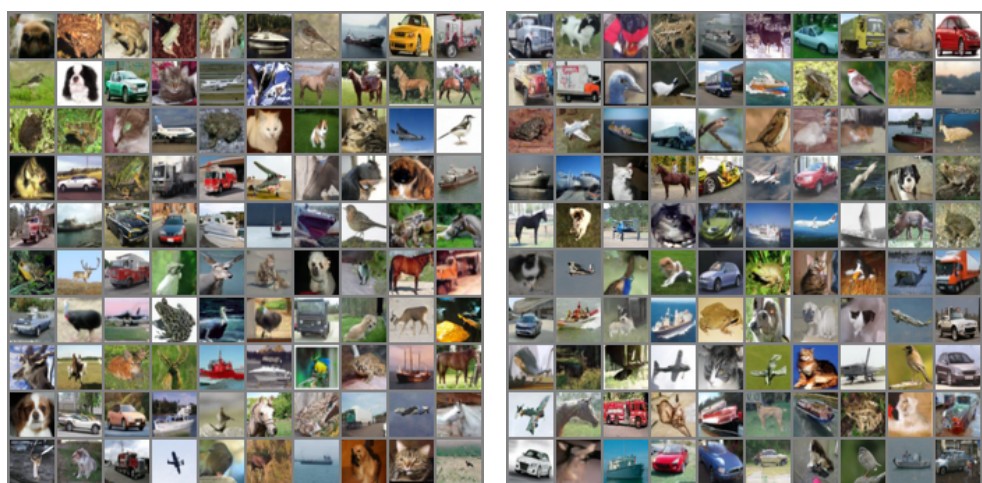

Figure 11: **SN-DDPM + DC-DPM on 100 Denoising Steps.**

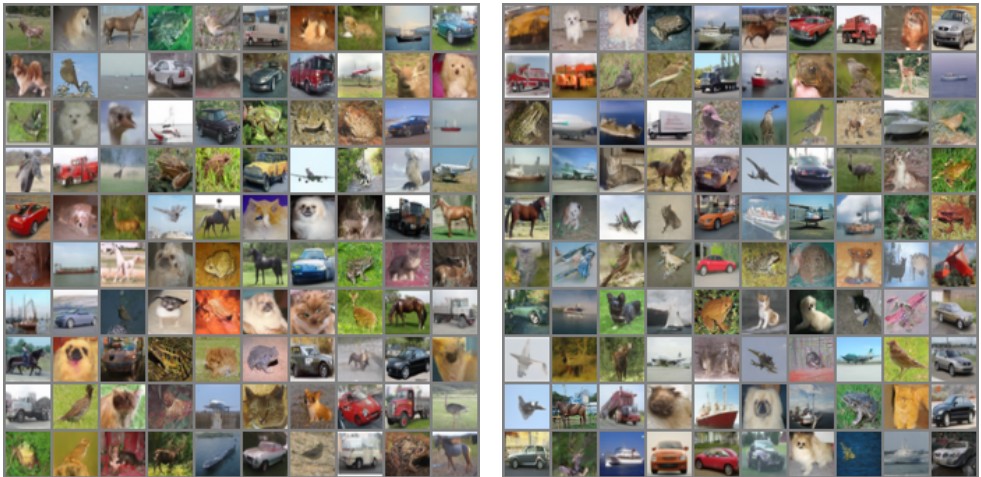

Figure 12: **GMS (Guo et al., 2024) + DC-DPM on 10 Denoising Steps.**

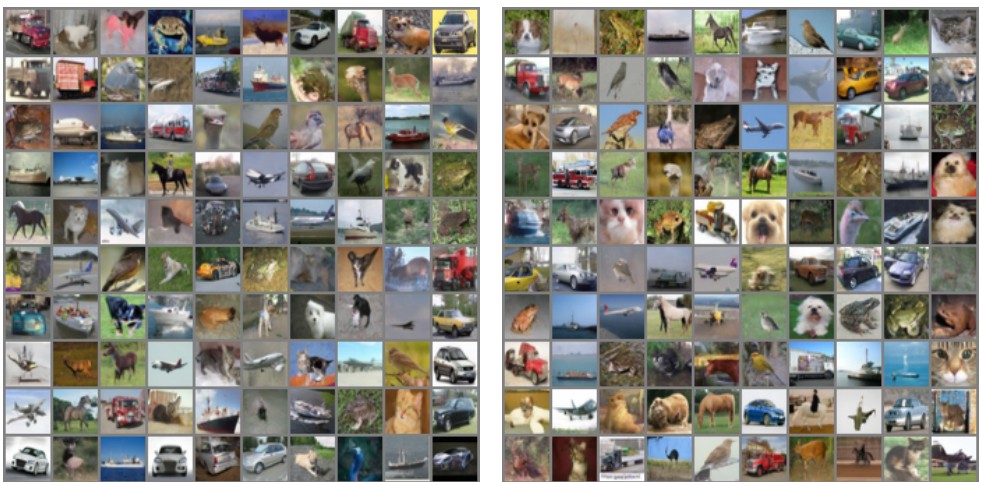

Figure 13: **GMS + DC-DPM on 25 Denoising Steps.**

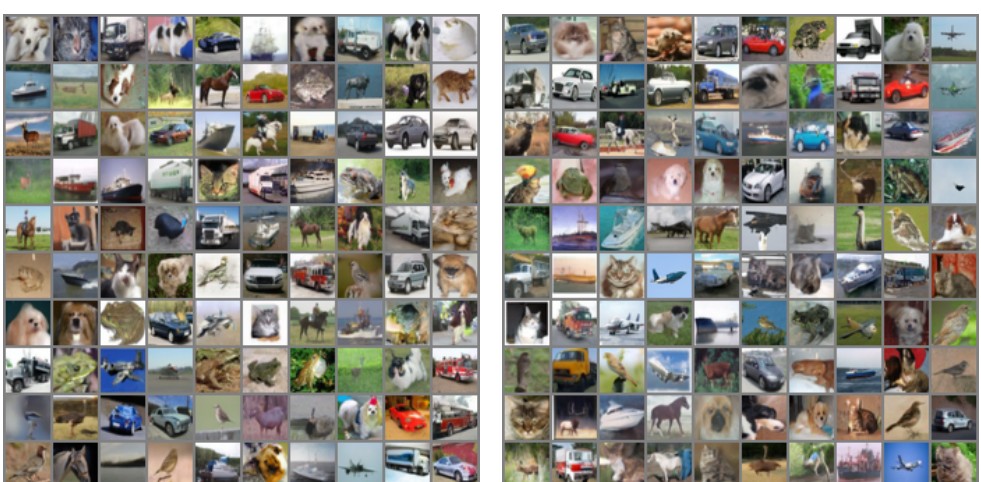

Figure 14: **GMS + DC-DPM on 50 Denoising Steps.**

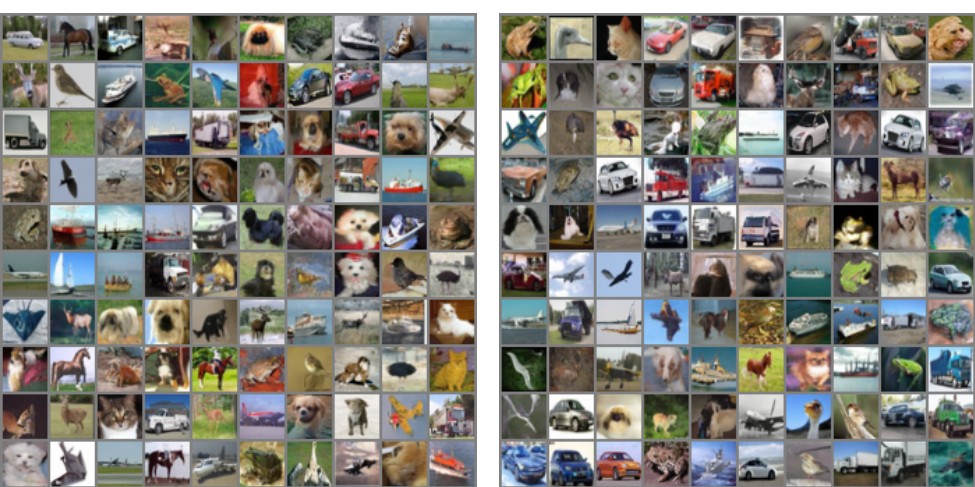

Figure 15: **GMS + DC-DPM on 100 Denoising Steps.**

