# OpenReview forum: "DC-DPM: A Divide-and-Conquer Approach for Diffusion Reverse Process"
_ICLR.cc/2025/Conference — ICLR 2025 Conference Withdrawn Submission_

### Official Review · Reviewer_skMA · 2024-10-31

**Soundness:** 2
**Presentation:** 2
**Contribution:** 2
**Rating:** 3
**Confidence:** 4

**Summary:**

This paper explores accelerating diffusion model sampling by improving the transition kernel of the reverse process. The authors aim to narrow the gap between the ground-truth reverse kernel and the conventional unimodal Gaussian kernel by employing a divide-and-conquer approach—using distinct kernels tailored to clustered data. Leveraging the proposed proof technique, the authors demonstrate that the improved kernel can guarantee convergence to the ground-truth reverse process. Experimental results indicate that this approach may effectively speed up diffusion models while maintaining performance.

**Strengths:**

1. This paper addresses a crucial problem in accelerating diffusion model sampling from a relatively under-explored perspective, i.e., improving the reverse process kernel.
2. The paper provides a substantial amount of theoretical analysis, along with a novel proof technique that can support convergence analysis.
3. The presentation is clear and easy to follow.

**Weaknesses:**

1. The motivation of the proposed approach comes from the inherent discrepancy between the ground truth reverse process, expressible as a mixture of Gaussians (in Eq. 5), and the traditional reverse kernel based on uni-modal Gaussian (i.e., Eq. 6). While this is a natural approach, a limitation remains as the authors still rely on the unimodal Gaussian kernel to approximate the MoG kernel for clustered data, as noted in L244-246. The convergence result to the ground-truth reverse process with this uni-modal approximation seems also missing.
2. Although the proposed approach offers advantages in accelerating diffusion models, it necessitates specialized training, making integration into conventional diffusion frameworks—proven to be highly effective—challenging.
3. Empirical validation on challenging, large-scale benchmarks (such as ImageNet) is lacking. The paper primarily focuses on relatively small, simpler datasets, such as CIFAR-10 and CelebA-HQ-256, raising questions about the scalability of the proposed approach when applied to more complex datasets.
4. A broader comparative study would strengthen the paper. Acceleration of diffusion models is a highly active research area, with numerous approaches offering speed improvements from different perspectives. However, the study covers only two baselines, both of which follow a similar methodological approach, i.e., improving the reverse kernel.
5. An analysis of computational costs would provide a more comprehensive comparison with baseline methods, especially since the proposed method involves the learning of multiple kernels, as well as additional neural network training (for DC-LD).

**Questions:**

See the weaknesses above.

---

> ### Author Response · Authors · 2024-11-24
> **Responses to Questions and Weaknesses from Reviewer skMA**
>
> We sincerely thank the reviewer for the valuable suggestions. We respond to the reviewer's comments point by point as follows.
>
> >**Weakness 1**: The motivation of the proposed approach comes from the inherent discrepancy between the ground truth reverse process, expressible as a mixture of Gaussians (in Eq. 5), and the traditional reverse kernel based on uni-modal Gaussian (i.e., Eq. 6). While this is a natural approach, a limitation remains as the authors still rely on the unimodal Gaussian kernel to approximate the MoG kernel for clustered data, as noted in L244-246. The convergence result to the ground-truth reverse process with this uni-modal approximation seems also missing.
>
> Learning a component of the clustered data with a Gaussian is significantly simpler than learning from the entire dataset. In the extreme case where each component only has one sample, the reverse kernel of each component becomes a Gaussian. Intuitively, it is much more straightforward to fit a subset of data with a Gaussian when the elements within the subset are closely related, as opposed to fitting the entire dataset with a Gaussian. In fact, we have found instances where a Mixture of Gaussians (MoG) outperforms a single Gaussian in approximating the whole dataset.
>
> Let $C = \frac{\alpha_s^2 \sigma_{t|s}^2}{2 \sigma_s^2 \sigma_t^2}$. If $(\bar{y}^{l_1})^T \bar{y}^{l_2} = 0$ for all $l_1, l_2$ and $\bar{y}^l > \log a^l / C (a^l - 1)$, then we have $KL(p(x_s | x_t) || \sum_l a^l(x_t, t) \~{p}^l(x_s | x_t)) < KL(p(x_s|x_t) || \~{p}(x_s | x_t))$.
>
> Additionally, a closely clustered subset will reduce the $\sigma^*_f$ of theorem 5.3 in [1], which will accelerate the SGD convergence rate and simplify the training process.
>
> > **Weakness 2**: Although the proposed approach offers advantages in accelerating diffusion models, it necessitates specialized training, making integration into conventional diffusion frameworks—proven to be highly effective—challenging.
>
> We would like to suggest that our key contribution is not solely in accelerating diffusion models. We theoretically propose a generalization of the conventional diffusion process, extending the range of diffusion models with solid theoretical proof.
>
> The generalized diffusion model, DC-DPM, is able to more accurately compute each step in the reverse process and naturally has the advantage of accelerating the diffusion process. However, the benefits of DC-DPM go beyond acceleration. The label diffusion in DC-DPM can be regarded as a single-token-predicting auto-regressive network. This more generalized form of the diffusion model provides a novel perspective to combine auto-regressive models with diffusion models. In this combination, the auto-regressive model interacts with the diffusion model at each denoising step. In contrast, conventional diffusion models limit auto-regressive models to serving only as conditions when combined.
>
> Regarding integration into real applications, given the rapid advancements and great success in parameter-efficient fine-tuning (PEFT) to extract general priors from pre-trained models, we believe it is sensible to employ techniques like LoRA[2] or adapters[3], which have showcased strong transfer learning abilities in a wide range of tasks, to redirect a pre-trained diffusion network to our DC-DPM with minimal training effort. However, this may be beyond the scope of this paper, which focuses on diffusion model theory.

---

> ### Author Response · Authors · 2024-11-24
> **Responses to Questions and Weaknesses from Reviewer skMA (2)**
>
> > **Weakness 3**: Empirical validation on challenging, large-scale benchmarks (such as ImageNet) is lacking. The paper primarily focuses on relatively small, simpler datasets, such as CIFAR-10 and CelebA-HQ-256, raising questions about the scalability of the proposed approach when applied to more complex datasets.
>
>  We appreciate your suggestion to evaluate the effectiveness of our method on large-scale benchmark. We conducted experiments by applying our method to DDPM on ImageNet64x64[4], and the results are as follows:
>  |                    | 25    | 50    |  100   | 200  | 400  |
> |--------------------|-------|-------|--------|------|------|
> | DDPM               | 29.21 | 21.71 | 19.12 | 17.81 | 17.48 |
> | SN-DDPM [5]           | 27.58 | 20.74 | 18.04 | 16.72 | 16.37 |
> | GMS [6]               | 26.50 | 20.13 | 17.29 | 16.60 | 15.98 |
> | **DDPM+DC-DPM (Ours)** | **24.60** | **18.91** | **16.46** | **14.93** | **14.00** |
>
> Our method demonstrates superior FID performance compared to previous techniques. Due to the limited time available during the rebuttal period, we only applied our DC-DPM to DDPM and were unable to train higher-order models on the large-scale ImageNet dataset for a more thorough comparison with SN-DDPM and GMS. Nevertheless, it is significant that merely integrating our method with DDPM, which employs a simple single Gaussian kernel for each data partition, has already surpassed the FID scores of SN-DDPM and GMS. These other methods involve training higher-order diffusion models and using more sophisticated kernel designs, underscoring the effectiveness of our approach on more complex datasets. We will include all results in the final version.
>
> > **Weakness 4**: A broader comparative study would strengthen the paper. Acceleration of diffusion models is a highly active research area, with numerous approaches offering speed improvements from different perspectives. However, the study covers only two baselines, both of which follow a similar methodological approach, i.e., improving the reverse kernel.
>
> Our main contribution is to provide a novel method to prove the convergence of diffusion models and further promote the conventional diffusion model to a more generalized form. This form approximates the reverse kernel in a divide-and-conquer manner rather than using a single Gaussian distribution, thereby enabling a more precise approximation at each sampling step.
>
> The two baselines we selected for comparison are the most relevant to our methodology, as they both propose new forms of reverse kernels. Our method, however, offers a more comprehensive representation by introducing a divide-and-conquer framework to depict the reverse transition kernel. Consequently, previous improvements oriented towards reverse-kernel can be utilized to represent each division within our framework.
>
> Proposing a new form of reverse kernel is at a more granular level compared with other previous diffusion model acceleration methods. Prior acceleration strategies, such as distillation [7] and training-free methods like DPM-Solver[8] and DPM-Solver++[9], typically continue to use a single Gaussian distribution to model each step in the reverse process. In contrast, we have developed a superior representation by employing a divide-and-conquer form for each step. These methods are orthogonal to ours and can be integrated with our approach.
>
> To substantiate our claim, we have conducted additional experiments. Results in the table below demonstrate that combining our method with acceleration approaches like DPM-Solver and DPM-Solver++ further enhances the generation quality:
> |            | 10    | 25    | 30 |  50   |
> |:----------|:-----:|:-----:|:-----:|:-----:|
> |DPM Solver [8]  | 7.95  | 6.54 | 6.17 | 3.37  |
> |**+DC-DPM(Ours)**| **5.78** | **3.49** | **3.28** |**2.90**  |
> |DPM Solver++ [9] | 11.11 | 7.41 | 6.76 | 3.42  |
> |**+DC-DPM(Ours)**| **10.94** | **4.29** | **3.80** |**2.99**  |

---

> ### Author Response · Authors · 2024-11-24
> **Responses to Questions and Weaknesses from Reviewer skMA (3)**
>
> > **Weakness 5**: An analysis of computational costs would provide a more comprehensive comparison with baseline methods, especially since the proposed method involves the learning of multiple kernels, as well as additional neural network training (for DC-LD).
>
> We appreciate the importance of such a comparison, especially given the multiple kernels and additional neural network training involved in our proposed method.
>
> However, it's important to clarify that our multiple kernels are not intended to be learned from multiple networks. Instead, we propose training a conditional diffusion network to limit the associated computational costs. For instance, in our ImageNet experiments, the parameter count for the diffusion network with and without the condition is 120.26M and 115.46M, respectively. This represents a mere 4.16\% increase in parameter count, demonstrating the minimal computational overhead introduced by our method to learn multiple kernels. Experimental results validate that this efficient design is effective enough to achieve a better generation results.
>
> Regarding the part to merge the multiple kernels, we offer the DC-FC variant, which does not require any additional computational costs or extra neural network training. As for the DC-LD variant, while it does necessitate training an additional label model, numerous model compression techniques can be applied for further optimization.
>
> For instance, both the label model and the diffusion model take the current timestep and noisy samples as inputs. Label model consists of an encoder and a light-weight prediction head while the diffusion model consists of an encoder and a decoder. A practical strategy would be to initially train the label model, then repurpose its encoder as the encoder for the diffusion model. We can freeze the encoder and train the decoder only. The shared encoder can effectively reduce both inference and training costs.
>
> However, it's important to note that the primary contribution of our work is the proposal of a novel proof for the diffusion model's convergence and an extension of conventional diffusion model forms. As such, these downstream application details are beyond the scope of our current work and will be considered for future extensions.
>
> References:
>
> [1] Garrigos G, Gower R M. Handbook of convergence theorems for (stochastic) gradient methods. ArXiv 2023.
>
> [2] Hu E J, Shen Y, Wallis P, et al. Lora: Low-rank adaptation of large language models. ICLR 2022.
>
> [3] Houlsby N, Giurgiu A, Jastrzebski S, et al. Parameter-efficient transfer learning for NLP. ICML 2019.
>
> [4] Deng J, Dong W, Socher R, et al. Imagenet: A large-scale hierarchical image database. CVPR 2009.
>
> [5] Bao F, Li C, Sun J, et al. Estimating the optimal covariance with imperfect mean in diffusion probabilistic models. ICML 2022.
>
> [6] Guo H, Lu C, Bao F, et al. Gaussian mixture solvers for diffusion models. NeurIPS 2024.
>
> [7] Song Y, Dhariwal P, Chen M, et al. Consistency models. ICML 2023.
>
> [8] Lu C, Zhou Y, Bao F, et al. Dpm-solver: A fast ode solver for diffusion probabilistic model sampling in around 10 steps. NeurIPS 2022.
>
> [9] Lu C, Zhou Y, Bao F, et al. Dpm-solver++: Fast solver for guided sampling of diffusion probabilistic models. ArXiv 2022.

---

> ### Comment · Reviewer_skMA · 2024-11-25
> **Post rebuttal (1)**
>
> Thank you for the detailed rebuttal.
>
> Firstly, the reviewer expressed concern regarding the authors' sudden shift in the core contribution of the paper - from focusing on acceleration to emphasizing convergence. This significant change represents a notable departure from the original submission, and it is unclear whether such a modification is appropriate during the rebuttal period. Furthermore, this shift might give the impression that the authors are attempting to sidestep addressing certain challenging comments by redirecting the focus; see the reviewer’s comments on the authors’ responses to W2 and W5 below for instance.
>
> Acknowledging that the primary contribution is now centered on convergence, the reviewer has two key concerns:
>
> - As also highlighted by Reviewer nRuj, the convergence results rely on Assumption 1, which models the true data distribution as a sum of Dirac delta functions. The reviewer questions whether the provided theoretical results hold for smooth data densities, which are more representative of real-world scenarios compared to Dirac deltas. To the reviewer's knowledge, prominent prior works addressing convergence, such as [1], typically consider smooth data distributions, albeit under certain smoothness constraints.
>
> - The reviewer also questions the tightness of the proposed convergence results relative to existing works. This aspect does not appear to be adequately substantiated in the current manuscript.

---

> ### Comment · Reviewer_skMA · 2024-11-25
> **Post rebuttal (2)**
>
> Here are the reviewer’s comments on the point-by-point rebuttal:
>
> **W1**. Addressed.
>
> **W2**.
> > Regarding integration into real applications, given the rapid advancements and great success in parameter-efficient fine-tuning (PEFT) to extract general priors from pre-trained models, we believe it is sensible to employ techniques like LoRA[2] or adapters[3], which have showcased strong transfer learning abilities in a wide range of tasks, to redirect a pre-trained diffusion network to our DC-DPM with minimal training effort. However, this may be beyond the scope of this paper, which focuses on diffusion model theory.
>
> I cannot agree with the assertion that PEFT techniques could enable the transfer of DC-DPM with minimal training effort. DC-DPM incorporates a distinct conditioning mechanism compared to existing diffusion models, which may necessitate re-training or computationally expensive fine-tuning processes. Thus, this argument should be substantiated with appropriate experimental evidence.
>
> > We would like to suggest that our key contribution is not solely in accelerating diffusion models. We theoretically propose a generalization of the conventional diffusion process, extending the range of diffusion models with solid theoretical proof. ... Regarding integration into real applications, ... , However, this may be beyond the scope of this paper, which focuses on diffusion model theory.
>
> Additionally, some responses give the impression that the authors are attempting to avoid addressing practically relevant comments—such as this one—by shifting the focus to the theoretical aspects of their contribution. The reviewer finds this approach inappropriate.
>
> **W3**. Thanks for the results. One question is whether the baselines are implemented with unconditional or class-conditional models.
>
> **W4**. The provided results exhibit *synergistic* benefits of DC-DPM when incorporated with DPM-solvers. This is distinct from the reviewer’s point to clarify the *isolated* advantage that comes solely from of the use of DC-DPM. For instance, the authors could compare the performance of DC-DPM with that of DDGAN [2] – a notable accelerating framework that involves a specialized training (similar to DC-DPM).
>
> **W5**. Beyond the parameter count, the convergence speeds of training class-conditional models may differ from those of unconditional models, and this distinction should be further clarified. Additionally, the training budget for DC-LD is still missing.
>
> Given your point that the training budget of DC-DPM is nearly the same as that of naive conditional models, the following questions arise: (i) Are the CIFAR-10 baseline models used in the paper class-conditional or unconditional? (ii) Does the proposed training method show an FID improvement over naive class-conditional models? Addressing these questions would help clarify the practical advantages of using the proposed method over naive class-conditional training.
>
> > However, it's important to note that the primary contribution of our work is the proposal of a novel proof for the diffusion model's convergence and an extension of conventional diffusion model forms. As such, these downstream application details are beyond the scope of our current work and will be considered for future extensions.
>
> This response once again suggests that the authors may be sidestepping certain practically relevant comments - such as this one - by redirecting the focus to the updated core contribution on theoretical aspects.
>
> ----
>
> [1] Convergence of score-based generative modeling for general data distributions. NeurIPS 2022.
>
> [2] Tackling the Generative Learning Trilemma with Denoising Diffusion GANs. ICLR 2022.

---

> ### Comment · Reviewer_skMA · 2024-11-25
> **Post rebuttal (3)**
>
> Given the heightened concern during the rebuttal process, I will adjust the score accordingly.

---

> ### Author Response · Authors · 2024-11-26
> **Responses to Reviewer skMA Post rebuttal**
>
> Thank you for your response. We apologize for any concerns we may have caused and would like to provide the following clarifications:
>
> > Emphasis on theoretical contributions:
>
> The proof of convergence is complex and required significant effort to achieve. However, based on general feedback, it seems that not all reviewers have fully appreciated this aspect. To address this, we have decided to highlight its importance in both the abstract and introduction. **We have not made any substantial changes to the proposed method itself**; instead, we are providing additional guidance to aid in a more accurate understanding of the paper.
>
> > Regarding the assumption of the sum of Dirac deltas
>
> We have adopted the assumption about the initial distribution from EDM[1]. This is because it accurately represents the data distribution during training, given the finite amount of training data available in practical scenarios. Furthermore, we can consider $y _i$s as independent and identically distributed samples from any underlying continuous distribution $\tilde{p} _{data}$, and $p _{data}$ in equation (1) will weakly converge to $\tilde{p} _{data}$ [2]. This implies that the sum of Dirac deltas can approximate any continuous distribution in the weak topology. An intuitive example of this concept is the Monte Carlo integral, which approximates the integral using only a finite number of samples. Moreover, our method can be extended to any initial distribution with compact support. The only requirement is to replace the sum over $y _i$s with an integral, and to validate the conditions for interchanging this integral with other integrals and derivatives. We will provide the proofs for these in the revised submission.
>
> > The tightness of the proposed convergence results relative to existing works
>
> The optimal convergence rate of diffusion models is $\Delta t^\frac{1}{2}$[3][4].  [5] managed to attain a superior error bound by making additional assumptions on the Jacobian of the score functions. However, in our paper, we do not make such assumptions on the Jacobian of the score functions, meaning the tightest bound we can achieve is $\Delta t^\frac{1}{2}$. This aligns remarkably well with our findings, as the $\beta$ in proposition 4 can be made arbitrarily small. As a result, we hypothesize that it might be possible to eliminate this $\beta$. This is an area we intend to concentrate on in our future work.
>
>
> > Response to W2:
>
> This strategy is inspired by SN-DDPM [6] and GMS [7]. In these works, the entire diffusion UNet is kept frozen during training, and only an attached convolution layer is trained. This approach redirects a pre-trained diffusion network from predicting noise to predicting high-order moments, demonstrating its efficacy. We will further provide additional experimental evidence and the corresponding training budget to support this approach.
>
> > Difference with Conditional Generation:
>
> Our method is different from conditional sampling. Conditional sampling, with a condition $c$, only generates the distribution $p(x_0 | c)$  rather than $p(x_0)$. In contrast, the distribution our method generates is $p(x_0)$. Thus we compare with the unconditional generation models in all the experiments, which also generates the distribution $p(x_0)$.
>
>
> We will conduct additional experiments to address the other concerns you raised.
>
> ---
> References:
>
> [1] Karras T, Aittala M, Aila T, et al. Elucidating the Design Space of Diffusion-Based Generative Models. NeurIPS 2022.
>
> [2] Varadarajan V S. On the convergence of sample probability distributions. Sankhyā: The Indian Journal of Statistics (1933-1960), 1958.
>
> [3] Lee H, Lu J, Tan Y. Convergence for score-based generative modeling with polynomial complexity. NeurIPS 2022.
>
> [4] Lee H, Lu J, Tan Y. Convergence of score-based generative modeling for general data distributions. PMLR 2023.
>
> [5] Li G, Wei Y, Chen Y, et al. Towards faster non-asymptotic convergence for diffusion-based generative models. arXiv 2023.
>
> [6] Bao F, Li C, Sun J, et al. Estimating the optimal covariance with imperfect mean in diffusion probabilistic models. ICML 2022.
>
> [7] Guo H, Lu C, Bao F, et al. Gaussian mixture solvers for diffusion models. NeurIPS 2024.

---

> ### Comment · Reviewer_skMA · 2024-11-26
>
> Thanks for your additional responses. Here are my further comments:
>
>
> > Emphasis on theoretical contributions
>
> The reviewer agrees that the manuscript contains a substantial amount of theoretical content, much of which remains unchanged from its initial version. However, the practical implementation aspects of the work are still significant within the manuscript. Therefore, the reviewer believes that practical concerns should be adequately addressed to substantiate the overall contribution of the work effectively.
>
>
> > Regarding the assumption of the sum of Dirac deltas
>
> While the Dirac delta assumption is used in EDM [3], this does not necessarily justify its applicability in this work, as the contexts of its usage are fundamentally distinct. The reviewer kindly requests references to other works that rely on the Dirac delta assumption but share a similar focus on convergence aspects.
>
> The reviewer also highlights related concerns as follows.
>
> - The assumption considers discrete Dirac deltas as the ground-truth data distribution. This diverges significantly from reality, where the true distribution is typically assumed to be a continuous density [1,2]. In contrast, the Dirac delta is primarily used to represent empirical distributions (e.g., finite training data [3]). To the reviewer’s knowledge, this is why many theoretical studies rely on continuous density assumptions, as seen in [1].
>
> - Incorporating the Dirac delta assumption into the proposed theoretical results implies memorization of finite samples, which fails to account for the generalization capabilities often demonstrated by diffusion models with sufficient amounts of finite training data [2].
>
> Reviewer nRuJ has expressed similar concerns; please refer to their comments above. Hence, the reviewer recommends revising the mathematical results to adopt more reasonable assumptions.
>
>
>
> > Response to W2
>
> If the main baselines (i.e., SN-DDPM and GMS) employ PEFT and enjoy low complexities, it is essential to compare the computational costs of the proposed approach against these baselines to ensure a fair evaluation.
>
>
>
> > Difference with Conditional Generation
>
> I already understand that a naive conditional sampling (described in the authors’ explanation) is different from the proposed approach. My point is that conditional diffusion models are in fact another instance of implementing a divide-and-conquer approach for modeling data distributions, which can be applied under the same practical conditions as DC-DPM (e.g., with clustered data) while benefiting from much simpler implementations. Hence, demonstrating the superiority of DC-DPM over conditional diffusion models is critical to validate its practical relevance. However, as the authors noted, such comparisons are absent in the current version of the manuscript.
>
>
> Also, even with the use of additionally conditioning information, the benefits of DC-DPM appear limited compared to the baselines (i.e., SN-DDPM and GMS). For instance, SN-DDPM improves 49.3% of the DDPM sampler at 10-step of CIFAR-10 (Linear Schedule), reducing FID from 43.14 to 21.87. In contrast, DC-DPM (DC-FC) shows 20.1% enhancements over DDPM from 43.14 to 34.48.
>
> While there may be inconsistencies in the diffusion backbones used across the three methods (which may cause the performance gap), the current experimental results do not clearly demonstrate the practical advantages of DC-DPM over existing methods.
>
>
> ----
> [1] Convergence of score-based generative modeling for general data distributions. NeurIPS 2022.
>
> [2] Generalization in diffusion models arises from geometry-adaptive harmonic representations, ICLR 2024.
>
> [3] Elucidating the Design Space of Diffusion-Based Generative Models, NeurIPS 2022.

---

### Official Review · Reviewer_P77B · 2024-11-03

**Soundness:** 2
**Presentation:** 2
**Contribution:** 2
**Rating:** 5
**Confidence:** 3

**Summary:**

This paper focuses on the challenge of approximating the reversed transition kernel of diffusion models with a Gaussian distribution. To address this problem, they propose a divide-and-conquer strategy and introduce DC-DPM. This approach clusters the data and learns transition kernels tailored to each cluster. The training and sampling processes are facilitated by two proposed approaches: label diffusion approximation and fixed class approximation. Additionally, the paper provides a theoretical convergence analysis. They empirically demonstrate that the proposed method yields improved data quality on benchmark dataset.

**Strengths:**

* The proposed method is both intuitive and reasonable. Since the characteristics of the transition kernel may vary depending on the data, adding it as a conditioning factor is likely to be effective. This is further supported by many previous experimental results showing that conditional datasets tend to achieve better FID scores than unconditional ones.
* The theoretical background supporting the proposed method is well-founded and lends credibility to the approach.

**Weaknesses:**

* Several assumptions are introduced to support the theoretical background, but it would be beneficial to explain whether these assumptions align with practical applications or under what conditions they hold in practical scenarios.

* Although the paper states that convergence is guaranteed with arbitrary data partitioning, there may be issues related to convergence speed, so it would be beneficial to further explain for the partitioning.

* The arbitrary data partitioning may also pose a challenge for training $a_\phi$ network, as the learning difficulty may vary depending on the partitioning.

* In the explanation of Corollary 3, it is mentioned that $\epsilon_{al}$ is expected to have a smaller influence than $\epsilon_{yl}$. However, this influence could vary depending on the scales of the constants $C_2$ and $C_3$, so further analysis on this point would be valuable.

* As the current experiments focus mainly on benchmark datasets with limited datasets and network structures, further experimental validation would strengthen the credibility of the method.

  * As data partitioning is an important part of the proposed method, it would be helpful to evaluate the effectiveness of the method on datasets with diverse semantic features, such as ImageNet.

  * It would be beneficial to test the proposed method on more advanced diffusion models, such as EDM [1].

  * Demonstrating its applicability to common applications of diffusion models, such as text-to-image generation, would further strengthen the approach.

* The LD variants sometimes underperform compared to the baseline models, so analyzing these cases would provide insight into potential limitations.

[1] Karras et al., Elucidating the Design Space of Diffusion-Based Generative Models, NeurIPS 2022

**Questions:**

* Could the authors provide their thoughts on the points mentioned in Weaknesses, particularly regarding the experimental point?
* In the convergence analysis, what distinguishes the proposed method from previous works in terms of achieving the tighter bound?

---

> ### Author Response · Authors · 2024-11-24
> **Responses to Questions and Weaknesses from Reviewer P77B**
>
> We would like to express our sincere gratitude to the reviewer for providing us with detailed comments and insightful questions. We have carefully considered the reviewer's feedback and would like to address each point as follows:
>
> > **Weakness 1**: Several assumptions are introduced to support the theoretical background, but it would be beneficial to explain whether these assumptions align with practical applications or under what conditions they hold in practical scenarios.
>
> We apologize for not initially providing explanations for the assumptions. Here are the detailed explanations for each assumption introduced in the paper and we have now added them in our revised submission:
>
> + **Assumption 1**: We adopt the assumption about the initial distribution from [1], as it reflects the actual scenarios during training, given the finite quantity of training data available in real-world situations. Additionally, we can consider $y_i$s as independent and identically distributed samples from any underlying continuous distribution $\tilde{p} _{data}$ , and $p _{data}$ in equation (1) will weakly converges to $\tilde{p} _{data}$ [2]. Furthermore, our method can be extended to any initial distribution with compact support. The only requirement is to replace the sum over $y_i$s with an integral, and verify the conditions for interchanging this integral with other integrals and derivatives. The second component of this assumption is that the gathered data is bounded, which is invariably the case in practical applications.
>
>
> + **Assumptions 2 and 4**: These assumptions have been adopted by previous studies [3][4][5] and is confirmed by [6].
>
> + **Assumption 3**: In practice, $\alpha _t$ is designed to be continuous and monotonically decreasing. This assumption is naturally satisfied, unless an unusual scheduler is introduced that induces unbounded derivative at $t=0$ or $t=1$.
>
> > **Weakness 2**: Although the paper states that convergence is guaranteed with arbitrary data partitioning, there may be issues related to convergence speed, so it would be beneficial to further explain for the partitioning.
>
> In this paper, our primary focus is on demonstrating the convergence of diffusion models using a novel method. Consequently, we do not delve deeply into the topic of partition selection. However, we can examine the speed of convergence for different partitions from the viewpoint of the Stochastic Gradient Descent (SGD) method.
>
> As per Theorem 5.3 of [7], the error bound of SGD aligns with two terms. Given the learning rate $\gamma _t$ and the initial state $x^0$, the first term is represented as $\frac{||x^0 - x^{ \star}||^2} {\sum_t \gamma_t}$, where $x^{ \star}$ is the optimal point. The second term is $2\sigma^{ \star} _f \frac{\sum _t \gamma_t^2}{\sum _t \gamma_t}$, where $\sigma^{ \star} _f$ is $\mathbb{E} ||\nabla f_i(x^{ \star}) - \mathbb{E} \nabla f_i(x^ \{star})||^2$.
> In our context, the choice of partition can be used to reduce $\sigma^{ \star}_f$. More specifically, when the data points $y_i^l$ of a single label are closely clustered, $\sigma^{ \star}_f$ will be small, which in turn leads to a faster convergence speed. Intuitively, the speed of convergence is quicker when each component of the partition is more compact.
>
> However, the optimization method we employ (AdamW) is more intricate than SGD, and the speed of convergence for different partitions is not straightforward to analyze. We therefore defer a detailed discussion on this topic to future work.
>
> > **Weakness 3**: The arbitrary data partitioning may also pose a challenge for training $a_\phi$ network, as the learning difficulty may vary depending on the partitioning.
>
> It's straightforward to validate that $a^l(x_t, t) = p(x_0 = l | x_t)$. As $t$ approaches 1, $a$ becomes nearly identical to a uniform distribution. Conversely, as $t$ nears 0, $a$ increasingly relies on the partition.
>
> Additionally, we can also examine the convergence speed through the SGD method. When each partition component is tightly clustered, the $\sigma^{*} _f$ of $a _\phi$ will be small, leading to a faster convergence speed. However, in this paper, our primary focus is on introducing a new proof method. Therefore, a detailed exploration of the convergence speed of $a _\phi$ will be reserved for future work.

---

> ### Author Response · Authors · 2024-11-24
> **Responses to Questions and Weaknesses from Reviewer P77B (2)**
>
> > **Weakness 4**: In the explanation of Corollary 3, it is mentioned that $\epsilon_{al}$ is expected to have a smaller influence than $\epsilon_{yl}$. However, this influence could vary depending on the scales of the constants $C_2$ and $C_3$, so further analysis on this point would be valuable.
>
> To be honest, since our focus is primarily on achieving a convergence result, the constants $C_2$ and $C_3$ are not particularly tight. There's room for effort to be made in obtaining sufficiently accurate coefficients to estimate the convergence speed. In our experiments with ImageNet, the final MSE loss of $\varepsilon_{yl}$ and $\varepsilon_{al}$ were 0.0435 and 0.000912 respectively. This suggests that the fitting error of the label model is significantly smaller than that of the diffusion model. Indeed, the primary influencing factor is the $T$ in the coefficient of $\varepsilon_{al}$, which makes the error of $a_\phi$ highly sensitive. In future work, we plan to focus our efforts on eliminating the $T$ in the coefficient, which should result in a more stable method.
>
> > **Weakeness 5.1**: As data partitioning is an important part of the proposed method, it would be helpful to evaluate the effectiveness of the method on datasets with diverse semantic features, such as ImageNet.
>
> We appreciate your suggestion to evaluate the effectiveness of our method on datasets with diverse semantic features. We conducted experiments by applying our method to DDPM on ImageNet64x64[8], and the results are as follows:
> |                    | 25    | 50    |  100   | 200  | 400  |
> |--------------------|-------|-------|--------|------|------|
> | DDPM               | 29.21 | 21.71 | 19.12 | 17.81 | 17.48 |
> | SN-DDPM [9]       | 27.58 | 20.74 | 18.04 | 16.72 | 16.37 |
> | GMS [10]           | 26.50 | 20.13 | 17.29 | 16.60 | 15.98 |
> | **DDPM+DC-DPM (Ours)** | **24.60** | **18.91** | **16.46** | **14.93** | **14.00** |
>
> Our method achieves better FID than previous methods. Due to the time constraints of the rebuttal period, we only applied our DC-DPM to DDPM and did not have enough time to further train higher-order models on the large-scale ImageNet dataset for a more comprehensive comparison with SN-DDPM and GMS. However, it is notable that merely employing our method on DDPM, which uses the naive single Gaussian kernel to represent each data partition, has already surpassed the FID of SN-DDPM and GMS, which further train higher-order diffusion models and employ more advanced kernel designs, showcasing the effectiveness of our method on more complex datasets. We will complete all the results in the final version.
>
> > **Weakness 5.2**: It would be beneficial to test the proposed method on more advanced diffusion models, such as EDM.
>
> Thank you for your suggestion to include a comparison with EDM. We have applied our DC-DPM to EDM, and as demonstrated in the experimental results below, our method further enhances the generation quality of EDM.
>
> |            | 10    | 25    |  30  |   50   |
> |------------|-------|-------|-------|--------|
> |EDM[1]         | 49.30  | 26.65 | 24.32 |    19.57    |
> |**+DC-DPM(Ours)**| **36.86** | **21.14** | **19.21** | **14.72**|
>
> > **Weakness 5.3**: Demonstrating its applicability to common applications of diffusion models, such as text-to-image generation, would further strengthen the approach.
>
> We would like to suggest that the key contribution of this paper is the novel proof method for the convergence of diffusion models and the extension to DC-DPM. Consequently, we focused on establishing the necessary inequalities for global convergence. Before we can proceed with text-to-image training, several issues need to be addressed, such as the partition method and neural network structure design, which require extensive experimentation.
>
> Previous theoretical publications in this area, including Analytic-DPM[12], SN-DDPM[9], GMS[10], DPM-Solver[13], and DPM-Solver++[14], evaluate generation quality on image benchmarks without showcasing results on text-to-image generation tasks. Our experiments, which align with these works, include benchmarks such as CIFAR10, CelebAHQ-256, and ImageNet 64x64, covering a range of datasets from small to large scale and from image space to latent space. We believe that these experiments are solid enough to validate the effectiveness of our proposed theory.
>
> Further application to downstream tasks like text-to-image generation is beyond the scope of this paper, which focuses on diffusion theory. Therefore, we have decided to leave large-scale text-to-image work for future research.

---

> ### Author Response · Authors · 2024-11-24
> **Responses to Questions and Weaknesses from Reviewer P77B (3)**
>
> > **Weakness 6**: The LD variants sometimes underperform compared to the baseline models, so analyzing these cases would provide insight into potential limitations.
>
> According to Corollary 3, the two primary sources of error are $\varepsilon_{yl}$ and $\varepsilon_{al}$. Given that the DC-FC method consistently outperforms the baseline, we infer that the error associated with $\varepsilon_{yl}$ is smaller than the baseline. This is in line with the above convergence rate analysis with respect to SGD.
>
> Since the DC-LD method performs better when the sampling step is small, specifically 10, we conclude that the error related to $\varepsilon_{al}$ is greatly influenced by the number of sampling steps $T$ in its coefficient. When $T$ is large, this term introduces more error during the sampling process. Even though the neural network $a_\phi$ is much closer to the ground truth than $y_\theta$, the error introduced by $T$ cannot be ignored.
>
> Our primary goal for future work is to eliminate the influence of $T$ in the error bound.
>
> > **Question 1**: Could the authors provide their thoughts on the points mentioned in Weaknesses, particularly regarding the experimental point?
>
> Please refer to the answers for Weaknesses5.1-5.3 above.
>
> > **Question 2**: In the convergence analysis, what distinguishes the proposed method from previous works in terms of achieving the tighter bound?
>
> Prior research has achieved an error bound of $\sqrt{\Delta t}$, which is remarkably close to our findings, as the $\beta$ in proposition 4 can be made arbitrarily small. Consequently, we hypothesize that it's possible to eliminate this $\beta$, and we plan to focus our future work on achieving this.
>
> Unlike previous studies that focus on the error bound, our paper emphasizes a method that is independent of the Kolmogorov equations, potentially inspiring extensions of diffusion models. We also believe that $\sqrt{\Delta t}$ is the optimal bound unless additional assumptions are introduced.
>
>
> References:
>
> [1] Karras et al., Elucidating the Design Space of Diffusion-Based Generative Models, NeurIPS 2022.
>
> [2] Varadarajan V S. On the convergence of sample probability distributions. Sankhyā: The Indian Journal of Statistics 1958.
>
> [3] Lee H, Lu J, Tan Y. Convergence for score-based generative modeling with polynomial complexity. NeurIPS 2022.
>
> [4] Lee H, Lu J, Tan Y. Convergence of score-based generative modeling for general data distributions. PMLR 2023.
>
> [5] Chen S, Chewi S, Li J, et al. Sampling is as easy as learning the score: theory for diffusion models with minimal data assumptions. ICML 2023.
>
> [6] Oko, Kazusato, Shunta Akiyama, and Taiji Suzuki. Diffusion models are minimax optimal distribution estimators. ICML 2023.
>
> [7] Garrigos G, Gower R M. Handbook of convergence theorems for (stochastic) gradient methods. ArXiv 2023.
>
> [8] Deng J, Dong W, Socher R, et al. Imagenet: A large-scale hierarchical image database. CVPR 2009.
>
> [9] Bao F, Li C, Sun J, et al. Estimating the optimal covariance with imperfect mean in diffusion probabilistic models. ICML 2022.
>
> [10] Guo H, Lu C, Bao F, et al. Gaussian mixture solvers for diffusion models. NeurIPS 2024.
>
> [11] Zhang P, Yin H, Li C, et al. Formulating discrete probability flow through optimal transport. NeurIPS 2024.
>
> [12] Bao F, Li C, Zhu J, et al. Analytic-dpm: an analytic estimate of the optimal reverse variance in diffusion probabilistic models. ICLR 2022.
>
> [13] Lu C, Zhou Y, Bao F, et al. Dpm-solver: A fast ode solver for diffusion probabilistic model sampling in around 10 steps. NeurIPS 2022.
>
> [14] Lu C, Zhou Y, Bao F, et al. Dpm-solver++: Fast solver for guided sampling of diffusion probabilistic models. ArXiv 2022.

---

> > ### Comment · Reviewer_P77B · 2024-11-26
> >
> > Thank you for the authors' response. After reviewing the response and comments from other reviewers, I still have concerns and will maintain my current rating.
> >
> > In particular, my concerns regarding the experiments have increased, and I believe a more detailed explanation of the experimental setup is needed. For example, the EDM experiment provided in response to Weakness 5.2 shows a significant difference in performance compared to the results reported in the original EDM paper. In the unconditional CIFAR-10 experiment, when the number of timesteps is 18 (35 NFE), the FID performance is reported as 1.97. If the authors used a different sampler, they should specify that, and even then I doubt the performance would be that different.

---

### Official Review · Reviewer_nRuJ · 2024-11-04

**Soundness:** 2
**Presentation:** 2
**Contribution:** 2
**Rating:** 3
**Confidence:** 3

**Summary:**

This paper proposes a Divide-and-Conquer strategy to improve the traditional single Gaussian transition kernel representation in each denoising step of Diffusion Probabilistic Models (DC-DPM), enhancing generation quality particularly over a limited number of timesteps. By dividing the data into clusters, our DC-DPM learns specific kernels for each partition.

**Strengths:**

A Divide-and-Conquer strategy to improve the traditional single Gaussian transition kernel representation in each denoising step of Diffusion Probabilistic Models (DC-DPM), which was shown to improve the sample generation quality in the numerical experiments.

**Weaknesses:**

The model assumptions seem confusing and need more clarification; there is limited comparison with existing theory on diffusion models; see below for more detailed comments.

**Questions:**

It would be beneficial if the authors could discuss its potential connection with conditional sampling. For example, in the CIFAR example, the clusters are based on the image labels, and thus the $y_\theta$ becomes $y_\theta(x,t,c)$ where c is the data label, this resembles the conditional sampling in certain extent.

My major concern and question are as follows. The data distribution is assumed as eq (1) which is essentially a uniform distribution supported on training data. This is a very confusing assumption to me, and since this is the starting point of all the theoretical analysis, I believe the authors needs to elaborate more on why this assumption is reasonable. Does this imply that the learned diffusion model will simply "memorize" data and cannot generate distribution outside of the empirical support well? Furthermore, I believe that in most theoretical analyses for diffusion models, the ground truth data distribution is assumed as a non-parametric distribution (continuous distribution), rather than the eq(1) assumed here. Additionally, as a paper studying the theoretical perspectives of diffusion models, some representative papers in this field are not cited or discussed, e.g.,  “Oko, Kazusato, Shunta Akiyama, and Taiji Suzuki. Diffusion models are minimax optimal distribution estimators. ICML 2023” and many others.

---

> ### Author Response · Authors · 2024-11-24
> **Responses to Questions and Weaknesses from Reviewer nRuJ**
>
> We sincerely appreciate the insightful comments provided by the reviewer. We have carefully considered each point raised and would like to respond as follows:
>
> >  **Question 1**: It would be beneficial if the authors could discuss its potential connection with conditional sampling. For example, in the CIFAR example, the clusters are based on the image labels, and thus the $y_\theta$ becomes $y_\theta(x,t,c)$ where c is the data label, this resembles the conditional sampling in certain extent.
>
> In the process of conditional sampling, with a condition $c$, we only generate the distribution $p(x_0 | c)$, not the desired $p(x_0)$. Mixing individual conditional models to generate the distribution $p(x_0)$ is not trivial. In DC-DPM, we introduce a new technique, the Label Diffusion Approximation (LD). This differs from conditional sampling in that the class label $c$ is sampled from the output of the neural network $a_\phi$, rather than being fixed.
>
> Additionally, we provide a robust proof that our merging method is "correct". This means that with a sufficiently small $|\mathcal{D}|$, the generated distribution can be made arbitrarily close to the data distribution. This is the primary contribution of our paper.
>
> The class prediction network $a_\phi$ can be interpreted as a single token prediction made by an auto-regressive model. By substituting the class label with a more complex structure, we can readily extend DC-DPM to a method that couples auto-regressive models with diffusion models. This could potentially inspire alternative approaches for multimodal models, offering a fresh perspective beyond the recently popular methods such as MAR[1] and Transfusion[2].
>
> In conclusion, we propose a theoretically solid and potentially inspiring method that we believe holds substantial value in the field of generative models. We think that it would be advantageous for researchers in this field to know this method.
>
> > **Question 2: About Assumption (1)**:  The data distribution is assumed as eq (1) which is essentially a uniform distribution supported on training data. This is a very confusing assumption to me, and since this is the starting point of all the theoretical analysis, I believe the authors needs to elaborate more on why this assumption is reasonable. I believe that in most theoretical analyses for diffusion models, the ground truth data distribution is assumed as a non-parametric distribution (continuous distribution), rather than the eq(1) assumed here.
>
> We adopt the assumption about the initial distribution from EDM[3], as it is precisely the conditions during training, given the finite quantity of training data available in real-world situations.
>
> Additionally, we can consider $y_i$s as independent and identically distributed samples from any underlying continuous distribution $\tilde{p}_ {data}$, and $p_{data}$ in equation (1) will weakly converges to $\tilde{p}_{data}$ [4].
>
> Furthermore, our method can be extended to any initial distribution with compact support. The only requirement is to replace the sum over $y_i$s with an integral, and verify the conditions for interchanging this integral with other integrals and derivatives.
>
> We have incorporated these explanations into our revised submission.
>
> > **Question 3**: Does this imply that the learned diffusion model will simply "memorize" data and cannot generate distribution outside of the empirical support well?
>
> This is an good question that pertains to the as-yet-undiscovered mechanism underpinning the generative ability of diffusion models. For unconditional diffusion models trained on a finite number of samples, the model can only generate samples from the training dataset, provided the model and sampling methods are accurate. As demonstrated in Appendix A.2 of [5], when $t$ is sufficiently small, the score function pushes $x_t$ towards the nearest data point.
>
> We have also conducted experiments using a unconditional DDPM model trained on approximately 100 images. The model was able to generate images nearly identical to those in the training dataset. The generative capacity of the diffusion model remains a mystery at this point.
>
> We hypothesize that this generative ability may stem from two factors. Firstly, the trained model and the numerical methods for SDE and ODE may not be entirely accurate. Secondly, we utilize conditional diffusion models which amortize across various conditions. This amortization allows the model to amalgamate different concepts derived from a range of text-image pairings.
>
> However, these are merely conjectures on our part. The true cause of the generative ability is still awaiting discovery.

---

> ### Author Response · Authors · 2024-11-24
> **Responses to Questions and Weaknesses from Reviewer nRuJ (2)**
>
> > **Question 4**: Some representative papers in this field are not cited or discussed, e.g., “Oko, Kazusato, Shunta Akiyama, and Taiji Suzuki. Diffusion models are minimax optimal distribution estimators. ICML 2023” and many others.
>
> This paper provided in the question delves into the error that may arise when employing a neural network to approximate the score function. Within the realm of diffusion models, we encounter two distinct types of errors.The first is the aforementioned approximation error, and the second is the discretization error, which originates from the numerical solutions to SDEs or ODEs.
>
> In our study, we focus on the discretization error. We operate under the assumption that the neural networks approximate the true score function in the $L^2(p(x_t))$ sense, which aligns perfectly with the findings of the above-referenced paper. We appreciate your providing evidence that supports our Assumption 2 and 4 regarding neural network approximation. In our revised submission, we have referenced this paper to substantiate these assumptions.
>
> References:
>
> [1] Li T, Tian Y, Li H, et al. Autoregressive Image Generation without Vector Quantization. ArXiv 2024.
>
> [2] Zhou C, Yu L, Babu A, et al. Transfusion: Predict the next token and diffuse images with one multi-modal model. ArXiv 2024.
>
> [3] Karras T, Aittala M, Aila T, et al. Elucidating the Design Space of Diffusion-Based Generative Models. NeurIPS 2022.
>
> [4] Varadarajan V S. On the convergence of sample probability distributions. Sankhyā: The Indian Journal of Statistics (1933-1960), 1958.
>
> [5] Zhang P, Yin H, Li C, et al. Tackling the Singularities at the Endpoints of Time Intervals in Diffusion Models. NeurIPS 2023.

---

> > ### Comment · Reviewer_nRuJ · 2024-11-28
> >
> > I would like to thank the authors for the detailed response, and sorry for my late reply. I will maintain my original score, and leave my final comments below.
> >
> > About conditional sampling: I understand this paper aims to generate $p(x)$ and conditional sampling is for $p(x|c)$, but for certain cases, the generation from $p(x|c)$ can be mixed together to produce samples from $p(x)$. Therefore, I do not think this is a major contribution of this work, and I would think the main contribution should lie in the theoretical side.
> >
> > About distribution assumption on $p_{data}$ and generalizability: To avoid confusion I will just write $p_{data}$ in this paper as $\hat p_{data}$ since it is assumed to be the empirical distribution. To the best of my knowledge, EDM is a methodology paper and theoretical development is not its core component (and $p_{data}$ only appear in its appendix roughly once). I do not think learning $\hat p_{data}$ is a good idea in theoretical studies (though it is indeed the case in the training process), as the convergence from $\hat p_{data}$ to the ground truth data distribution (the "True" $p_{data}$) suffer from the curse of dimensionality when data dimension is high (which is usually the case) even if the true $p_{data}$ is smooth, which makes learning $\hat p_{data}$ impossible to be minimax optimal. In the paper (Suzuki et al, ICML 2023), it was shown that diffusion models are minimax optimal distribution estimators, and the generalization abilities of diffusion modeling for well-known function spaces were studied.
> >
> > Overall, I appreciate all the effort the authors have made to revise the paper and answer reviewers' questions, but I think the paper will benefit from clearer explanations of its contributions and a more detailed theoretical analysis and comparison with existing literature. Therefore, I will maintain my original score.

---

### Official Review · Reviewer_agy8 · 2024-11-08

**Soundness:** 3
**Presentation:** 3
**Contribution:** 3
**Rating:** 6
**Confidence:** 3

**Summary:**

The paper introduces a strategy called Divide-and-Conquer to accelerate the convergence of Diffusion Probabilistic Models. This approach begins by partitioning the data, allowing for separate learning of transition kernels for each partition. These transition kernels are then combined during both the training and sampling phases to form a unified model. The authors also provide a formal convergence proof for their proposed strategy, reinforcing its theoretical soundness. Experimental results on the CelebA and CIFAR-10 datasets demonstrate that the proposed method improves both the Fréchet Inception Distance (FID) and the visual quality of generated images, highlighting its effectiveness in practical applications.

**Strengths:**

- The paper presents a straightforward yet effective strategy to enhance the learning performance of Diffusion Probabilistic Models (DPMs).

- The authors also develop a convergence proof for the proposed method, providing a solid theoretical foundation that supports its reliability.

- Additionally, the manuscript is well-organized and clearly presented.

**Weaknesses:**

- The paper presents numerous theorems, but it is unclear which are original contributions. Proper citation of existing results is needed to clarify which findings are novel and which build on prior work.
- The evaluation is limited to two datasets, which may not comprehensively assess the proposed method’s effectiveness. Including additional datasets would improve the robustness of the benchmarking.
- Only one baseline method is compared, despite significant related work in kernel representation that could serve as relevant baselines.
- The qualitative results in Figure 3 provide generated images with minimal discussion and analysis, limiting insights into the model's qualitative performance.

**Questions:**

What is the difference between DC-DPM and GMS, which use GMM as kernels?

---

> ### Author Response · Authors · 2024-11-24
> **Responses to Questions and Weaknesses from Reviewer agy8**
>
> We would like to express our sincere appreciation to the reviewer for providing us with detailed comments and suggestions. We have carefully reviewed each comment and offer the following responses:
>
>
> > **Weakness 1:** The paper presents numerous theorems, but it is unclear which are original contributions. Proper citation of existing results is needed to clarify which findings are novel and which build on prior work.
>
> We apologize for any confusion regarding the originality of our claims. To clarify, all claims, including Propositions 1-5, Corollaries 1-3, and Lemmas 2-6, are original contributions. The exception is Lemma 1, which is a well-known result included for completeness. We have emphasized the originality of our work in the revised submission.
>
> > **Weakness 2:** The evaluation is limited to two datasets, which may not comprehensively assess the proposed method‘s effectiveness. Including additional datasets would improve the robustness of the benchmarking.
>
> We appreciate your suggestion to evaluate the effectiveness of our method on more datasets. We conducted experiments on a more challenging dataset, ImageNet64x64[1] and the results are as follows:
> |                    | 25    | 50    |  100   | 200  | 400  |
> |:-------------|:-------:|:-------:|:--------:|:------:|:------:|
> | DDPM               | 29.21 | 21.71 | 19.12 | 17.81 | 17.48 |
> | SN-DDPM [2]           | 27.58 | 20.74 | 18.04 | 16.72 | 16.37 |
> | GMS [3]              | 26.50 | 20.13 | 17.29 | 16.60 | 15.98 |
> | **DDPM+DC-DPM (Ours)** | **24.60** | **18.91** | **16.46** | **14.93** | **14.00** |
>
> Our method achieves superior FID compared to previous approaches. Due to the time constraints of the rebuttal period, we only applied our DC-DPM to DDPM and did not have sufficient time to train higher-order models on the large-scale ImageNet dataset for a more comprehensive comparison with SN-DDPM and GMS. However, it is noteworthy that simply applying our method to DDPM, which uses a basic single Gaussian kernel for each data partition, has already outperformed the FID of SN-DDPM and GMS. These other methods further train higher-order diffusion models and utilize more advanced kernel designs, highlighting the effectiveness of our approach on more complex datasets. We will provide complete results in the final version
>
> >**Weakness 3:** Only one baseline method is compared, despite significant related work in kernel representation that could serve as relevant baselines.
>
> Thank you for your suggestion to compare our method with other kernel representation methods. However, the concept of kernel representation is quite broad, so it would be helpful if you could specify the "significant related work" you're referring to. We suspect you might be referring to the kernel trick in machine learning or the kernel methods in Gaussian Processes.
>
>
>
> In the context of the kernel trick in machine learning, a kernel is always provided, making the inner product tractable with the relationship $k(x,y) = <\phi(x), \phi(y)>$. In contrast, diffusion models estimate the transition kernel from the data. Additionally, many learning methods define an objective function and train a model to minimize it, but often do not provide proof that a well-trained model can achieve this minimum. In our paper, we focus on demonstrating that, given well-trained models and a sufficiently small $\Delta t$, the generated distribution can be made arbitrarily close to the data distribution. It is important to note that the objectives of the kernel trick in machine learning differ from our method, and thus it may not be appropriate to make direct comparisons between them.
>
> Regarding Gaussian processes, the kernels are more like a prior. The goal of Gaussian process models is to find the parameters that allow the model to best fit the training data. If the underlying process is not Gaussian, the model will never truly learn the ground truth distribution. However, as we have proven, our method will converge to the ground truth distribution. Therefore, we do not compare our method with this one.
>
> We hope this clarifies our position and we would be happy to discuss any specific works you feel are relevant.

---

> ### Author Response · Authors · 2024-11-24
> **Responses to Questions and Weaknesses from Reviewer agy8 (2)**
>
> > **Weakness 4:** The qualitative results in Figure 3 provide generated images with minimal discussion and analysis, limiting insights into the model's qualitative performance.
>
> We apologize for the lack of discussion regarding Figure 3 in our original submission.
> While we have demonstrated the convergence of our proposed method with DDPM (SDE), establishing convergence with DDIM (ODE) is not a trivial matter. However, our experimental results in Table 2 indicate that our method is effective with DDIM as well. The images in Figure 3 are generated using our DC-DPM with DDIM, and they serve as supplementary qualitative examples, showcasing that our method produces correct samples and suggesting compatibility with ODE-based samplers. In our revised submission, we have restructured the section on experiments involving the ODE-based sampling method.
>
> > **Questions:** What is the difference between DC-DPM and GMS, which use GMM as kernels?
>
> Our method differs from GMS in the following three aspects:
> +  **Kernel Representation**: GMS utilizes only two Gaussian kernels with fixed weights of 1/3 and 2/3, and proposes to determine the unknown parameters for GMM by estimating the first three moments of the reverse conditional distribution. In contrast, our DC-DPM proposes a more general framework. We not only enable the use of a flexible number of kernels but also directly estimate the weights and centers of each kernel. Each kernel can be approximated with different methods and any previous method, including GMS, can be used for this purpose.
> + **Theoretical Robustness**: One of our key contributions is that we provide robust evidence demonstrating that DC-DPM can generate distribution arbitrarily close to the data distribution. However, it's important to note that GMS does not offer a similar guarantee of convergence.
> + **Potential Extension**: The weight estimation in DC-DPM, referred to as label diffusion, can be viewed as an auto-regressive model predicting a single token. This approach can be extended to more complex formulations, such as text and audio, indicating a novel coupling strategy between diffusion models and auto-regressive models. This differs from methods like MAR[4] and Transfusion[5], where auto-regressive models merely provide conditions for diffusion models, similar to current text-to-image models such as Stable Diffusion[6], DALL-E[7], and FLUX[8]. In our approach, diffusion models and auto-regressive models interact at each diffusion step. Consequently, our method could inspire the creation of a unified text-image model, a potential that GMS does not possess.
>
> References:
>
> [1] Deng J, Dong W, Socher R, et al. Imagenet: A large-scale hierarchical image database. CVPR 2009.
>
> [2] Bao F, Li C, Sun J, et al. Estimating the optimal covariance with imperfect mean in diffusion probabilistic models. ICML 2022.
>
> [3] Guo H, Lu C, Bao F, et al. Gaussian mixture solvers for diffusion models. NeurIPS 2024
>
> [4] Li T, Tian Y, Li H, et al. Autoregressive Image Generation without Vector Quantization. ArXiv 2024.
>
> [5] Zhou C, Yu L, Babu A, et al. Transfusion: Predict the next token and diffuse images with one multi-modal model. ArXiv 2024.
>
> [6] Rombach R, Blattmann A, Lorenz D, et al. High-resolution image synthesis with latent diffusion models. CVPR 2022.
>
> [7] OpenAI. Dall-E 3. Available at: https://openai.com/index/dall-e-3/.
>
> [8] Black Forest Labs. FLUX. Available at: https://github.com/black-forest-labs/flux.

---

### Author Response · Authors · 2024-11-24
**To all Reviewers, AC and PC: Contribution and summary of modifications during discussion**

We deeply thank the reviewers for their appreciation of this work and insightful feedbacks. Your suggestions greatly help us to improve and enrich our work. In the following discussions, we individually address your comments.

**Upon reviewing the abstract and the introduction, we realized that the main contribution of this paper wasn't sufficiently emphasized, which could have led the reviewers to overlook it. This paper goes beyond a technical improvement by proposing a divide-and-conquer strategy for the diffusion reverse transition kernel. More importantly, we introduce a novel method to prove the convergence of diffusion models. Unlike previous methods that require the reverse process to be derived from a Stochastic Differential Equation (SDE), our approach demonstrates that diffusion models with non-SDE reverse processes can also converge. Consequently, we have revised our abstract and introduction to better highlight these points. The revised sections are indicated in blue.**

Our method thus extends the framework of converged diffusion models and introduces a more powerful representation of the reverse process, namely DC-DPM, which employs a divide-and-conquer approach for the reversed transition kernel. According to our experimental results, DC-DPM exhibits superior generation capabilities compared to conventional diffusion models. It is important to note that the convergence of our DC-DPM cannot be proven using any of the existing SDE-based diffusion model convergence proof methods.

Additionally, DC-DPM enables a new way to integrate auto-regressive models with diffusion models. Specifically, the label diffusion model can be viewed as a single-token-predicting auto-regressive model and can be extended to generate more complex modes like text and audio.
In our approach, diffusion models and auto-regressive models interact at each diffusion denoising step, unlike previous methods such as MAR[1] and Transfusion[2], where auto-regressive models merely provide conditions for diffusion models.

We have updated our paper for adding highlights on the modifications we made to our paper. we mainly modified our manuscript as listed below:
+ The abstract and introduction are modified to demonstrate our contribution more clearly and emphasize our theoretical contribution.
+ More detailed explanation of our proposed theory are included in the method section for better clarity.
+ Experiments on a more challenging, larger-scale dataset, ImageNet64x64[3], are provided in appendix, showcasing our method can still outperform previous methods on datasets with diverse semantic features.
+ Experiments applying our DC-DPM to other representative diffusion acceleration methods, including DPM-Solver[4], DPM-Solver++[5], and quality improvement method EDM[6], are provided in the appendix to demonstrate the further improvements achieved when combined with our method.

Due to the page limit of the main paper, we move the ablation study result in the paper before to the appendix.

We hope our revised manuscript and additional experimental results can help address our reviewers' concerns and doubts. We look forward to your further replies. Thanks.

References:

[1] Li T, Tian Y, Li H, et al. Autoregressive Image Generation without Vector Quantization. ArXiv 2024.

[2] Zhou C, Yu L, Babu A, et al. Transfusion: Predict the next token and diffuse images with one multi-modal model. ArXiv 2024.

[3] Deng J, Dong W, Socher R, et al. Imagenet: A large-scale hierarchical image database. CVPR 2009.

[4] Lu C, Zhou Y, Bao F, et al. Dpm-solver: A fast ode solver for diffusion probabilistic model sampling in around 10 steps. NeurIPS 2022.

[5] Lu C, Zhou Y, Bao F, et al. Dpm-solver++: Fast solver for guided sampling of diffusion probabilistic models. ArXiv 2022.

[6] Karras T, Aittala M, Aila T, et al. Elucidating the design space of diffusion-based generative models. NeurIPS 2022.

---

### Note · Authors · 2024-12-03

I have read and agree with the venue's withdrawal policy on behalf of myself and my co-authors.